# Nicheformer: a foundation model for single-cell and spatial omics

Alejandro Tejada-Lapuerta[1,2,7], Anna C. Schaar[1,2,7], Robert Gutgesell [2,3], Giovanni Palla[2,4], Lennard Halle [2], Mariia Minaeva [2,4], Larsen Vornholz [2,4], Leander Dony [2,4,5], Francesca Drummer[2,6], Till Richter[1,2], Mojtaba Bahrami [2,4] & Fabian J. Theis [1,2,4] ✉

Tissue makeup depends on the local cellular microenvironment. Spatial single-cell genomics enables scalable and unbiased interrogation of these interactions. Here we introduce Nicheformer, a transformer-based foundation model trained on both human and mouse dissociated single-cell and targeted spatial transcriptomics data. Pretrained on SpatialCorpus-110M, a curated collection of over 57 million dissociated and 53 million spatially resolved cells across 73 tissues on cellular reconstruction, Nicheformer learns cell representations that capture spatial context. It excels in linear-probing and fine-tuning scenarios for a newly designed set of downstream tasks, in particular spatial composition prediction and spatial label prediction. Critically, we show that models trained only on dissociated data fail to recover the complexity of spatial microenvironments, underscoring the need for multiscale integration. Nicheformer enables the prediction of the spatial context of dissociated cells, allowing the transfer of rich spatial information to scRNA-seq datasets. Overall, Nicheformer sets the stage for the next generation of machine-learning models in spatial single-cell analysis.

Single-cell genomics technologies have advanced our understanding of cellular heterogeneity in tissues, organs and organisms. Large-scale data generation efforts have charted cellular atlases of specific tissues and organs, such as the lung[1] and heart[2], as well as broader cross-tissue atlases[3]. However, single-cell RNA sequencing (scRNA-seq) requires cell dissociation, losing information about the cellular microenvironment and hindering a complete understanding of molecular variation[4]. Recent advances in image-based spatial transcriptomics enable in situ scRNA-seq, profiling hundreds of genes in hundreds of thousands of cells across various tissues[4,5]. In situ spatial omics has revealed spatial components of cellular variations such as cell–cell communication[6] and

spatial gradients as well as emergent properties of tissue niches[7], for example, in the mouse and human brain[8,9] and liver[10]. We hypothesize that spatial omics data are becoming rich enough to learn a spatially aware, 'foundational' representation of cellular variation at scale.

A foundation model is a deep learning model trained on broad data that can be adapted to a wide range of downstream tasks. These models have revolutionized fields such as natural language processing[11] and computer vision[12]. Foundation models increasingly account for multimodal data, by leveraging not only one data modality, for example text, but also images, video and audio[13]. By utilizing massive datasets, powerful architectures and large compute resources,

[1]TUM School of Computation, Information & Technology, Technical University of Munich, Garching, Germany. [2]Institute of Computational Biology, Computational Health Center, Helmholtz Munich, Neuherberg, Germany. [3]Institute for Diabetes and Obesity, Helmholtz Diabetes Center, Helmholtz Munich, Neuherberg, Germany. [4]TUM School of Life Sciences Weihenstephan, Technical University of Munich, Freising, Germany. [5]Department Genes and Environment, Max Planck Institute of Psychiatry and International Max Planck Research School for Translational Psychiatry (IMPRS-TP), Munich, Germany. [6]Institute for Stroke and Dementia Research, Klinikum Der Universität München, Ludwig Maximilian University of Munich, Munich, Germany. [7]These authors contributed equally: Alejandro Tejada-Lapuerta, Anna C. Schaar. ✉e-mail: fabian.theis@helmholtz-munich.de

foundation models learn general representations of language, vision or domain-specific data like DNA[14] and protein sequences[15], outperforming classical methods. Commonly based on transformer architectures, they are pretrained on vast, unlabeled data via self-supervision, learning powerful representations by identifying patterns without human-annotated labels. These learned representations then serve as a strong base for downstream tasks, while fine-tuning on labeled data further enhances performance on specific applications.

The field of single-cell biology has taken up deep learning-based representation learning for some time, leveraging autoencoders[16,17] for analysis tasks like data integration[18], atlas mapping[19] and perturbation prediction[20]. Recently, foundation models explicitly designed for single-cell genomics have emerged[21–25]. These models differ in tokenization and learning strategies, yet most of them leverage the transformer architecture with self-attention. They rely on large datasets, usually in the order of tens of millions of cells, for pretraining. The gene and cell representations learned by these models are derived from implicitly modeling the complex interplay between gene expression patterns within a single cell via the flexible transformer architecture. Single-cell foundation models are evaluated on diverse downstream tasks, such as cell-type classification[22,23], gene regulatory network inference[21,22] or prediction of cellular responses to perturbations[21]. The diversity and complexity of these tasks thoroughly probe model performance and evaluate the robustness of the learned representation and generalization ability. Current results are promising but not entirely replicated in independent benchmarks[26–28]. Notably, these models do not account for spatial relationships of cells during training, with the exception of CellPLM[29], which, however, is trained on a limited dataset of 9 million dissociated and 2 million spatial transcriptomics cells and not fine-tuned on spatial tasks beyond gene imputation.

We propose Nicheformer, a foundation model pretrained on large-scale, single-cell and spatial transcriptomics data to enable predictions for spatially dependent tasks that are constrained by limited training data. To learn spatial cellular representation at scale, we compiled SpatialCorpus-110M, a large curated collection of single-cell and spatial transcriptomics datasets, spanning over 110 million cells, including 53.83 million cells that were measured using image-based spatial technologies, from both human and mouse from 73 different organs and tissues. By incorporating contextual information through modality, organism and assay tokens, Nicheformer is able to learn a joint representation of single-cell and spatial genomics. We designed a set of novel downstream tasks showing that both fine-tuned Nicheformer and a linear-probing model trained on the Nicheformer embedding systematically outperform existing foundation models, specifically Geneformer[22], scGPT[21] and UCE[23] pretrained on dissociated data alone, foundation models trained in spatial data, specifically CellPLM[29], and embedding models like scVI[17] and principal-component analysis (PCA) for these tasks. We demonstrate that Nicheformer accurately transfers the spatial context identified in spatial transcriptomics onto dissociated single-cell data, allowing users to enrich nonspatial scRNA-seq data with spatial context. This work paves the way for a new generation of foundation models for learning robust representations of cellular variation in tissues.

## Results

### A transformer-based foundation model for combined spatial and disassociated single-cell data

**Overview.** Nicheformer is a transformer-based model pretrained on SpatialCorpus-110M, a curated collection of over 110 million cells from dissociated and spatially resolved single-cell assays (Fig. 1a). Nicheformer generalizes prior tokenization strategies[22] by encoding sample covariates across technology modalities, enabling a unified framework for multimodal learning, opening up new possibilities for downstream tasks. We additionally enable learning multispecies embeddings with Nicheformer by defining orthologous genes across

humans and mice (Methods), which was shown to work beneficially for cross-species biological investigations and enhanced the discovery of universal gene regulatory mechanisms[30]. We evaluated Nicheformer on new downstream tasks to demonstrate its ability to transfer spatially inferred cellular variation to single-cell dissociated data (Fig. 1b).

The Nicheformer pretraining corpus comprises transcriptomics data from both humans and mice (Fig. 1a). Only expression data were used during pretraining to train the model to integrate data from dissociated and targeted spatial technologies, both of which show substantial batch effects (Fig. 1a). A limiting factor for image-based spatial transcriptomics data is the targeted feature space, measuring only hundreds to a few thousands of genes, depending on technology and panel[31]. Nicheformer is pretrained across both modalities jointly to capture cross-tissue, cross-technology and cross-disease variations. For evaluation of the downstream tasks, we focused on large-scale spatial datasets from four different solid organs profiled with three image-based technologies (Fig. 1b). We fine-tuned Nicheformer or applied linear probing, extracting embeddings from the frozen model and passing them through a task-specific linear layer for classification or regression (Methods). The embedding is obtained via forward passing a specific dataset through the pretraining model to generate a lower-dimensional representation, the so-called Nicheformer embedding. The organ-specific spatial context learned by Nicheformer can then be used to evaluate the model's ability to generalize information learned from spatial transcriptomics data, without directly accounting for the available spatial context, and transfer it to dissociated data.

**Cell representation.** We define a cell as a sequence of gene expression tokens ordered by expression level relative to the mean in SpatialCorpus-110M (Fig. 1c). As the corpus includes human and mouse data, we constructed a shared vocabulary by concatenating orthologous protein-coding genes and species-specific ones, totaling 20,310 gene tokens (Fig. 1c and Methods). Each single-cell expression vector is converted into a ranked sequence of gene tokens (Fig. 1d and Methods), a strategy shown to yield embeddings robust to batch effects while preserving gene–gene relationships[22]. We combined all technology-specific datasets and pad missing genes. Previous works[31] have demonstrably shown technology-dependent biases between spatial and dissociated transcriptomics data, with spatial data often yielding higher gene counts due to preprocessing steps[32]. To account for this, we computed technology-specific nonzero mean vectors—rather than a global one—by averaging nonzero gene expression values within each assay type. Dissociated assays are grouped as one technology, whereas spatial datasets are divided into multiplexed error-robust fluorescence in situ hybridization (MERFISH), Xenium, CosMx and in situ sequencing (ISS) technologies. Finally, we introduced contextual tokens for species, modality and technology, enabling the model to learn their distinct characteristics. As rank-based encoding is central to our approach, we confirmed that Nicheformer embeddings remain stable under perturbations, simulating incomplete gene panels (Extended Data Fig. 1a,b and Methods).

**Model design and training.** Nicheformer uses a 1,500-token context length as input to an architecture with 12 transformer encoder units with 16 attention heads per layer and a feed-forward network size of 1,024, generating a 512-dimensional embedding, resulting in a total of 49.3 million parameters. This architecture performed best compared to smaller models (Extended Data Fig. 2c) and other hyperparameter configurations (Supplementary Table 1).

We confirmed technology-dependent biases between spatial and dissociated transcriptomics data through extensive pretraining experiments across different data splits (Methods). Specifically, training on dissociated data alone (even three times the amount of spatial data) resulted in lower performance across downstream tasks (Extended Data Fig. 2a,b), indicating that dissociated data alone cannot

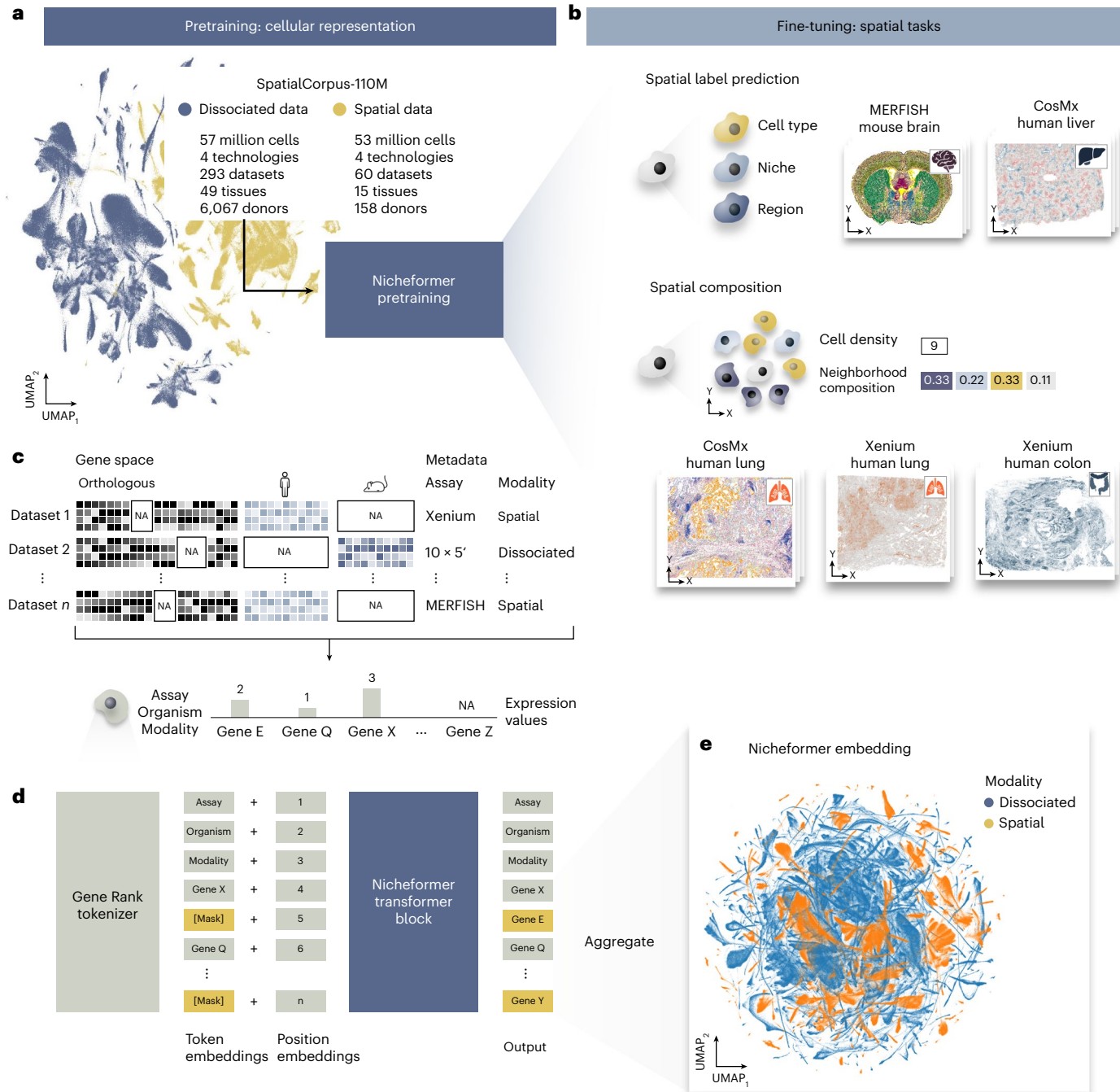

**Fig. 1 | Nicheformer, a foundation model for spatial transcriptomics.**
**a**, Nicheformer is pretrained on the SpatialCorpus-110M, a large data collection of over 110 million cells measured with dissociated and image-based spatial transcriptomics technologies. The SpatialCorpus-110M collection comprises single-cell data from *Homo Sapiens* and *Mus Musculus* across 17 distinct organs and 18 cell lines, and additional single-cell data from other anatomical systems and junctions. Shown is an exemplary uniform manifold approximation and projection (UMAP) visualization of a random 1% subset of the entire pretraining dataset (*n* = 1,108,759 cells) of the non-integrated log1p-transformed normalized SpatialCorpus-110M colored by modality. **b**, Nicheformer includes a novel set of downstream tasks, ranging from spatial cell-type, niche and region label prediction to neighborhood cell density and neighborhood composition prediction. We test our approach on large-scale, high-quality spatial transcriptomics data from the brain (mouse, MERFISH), liver

(CosMx, human), lung (CosMx, human; Xenium, human) and colon (Xenium, human). Visualized are example slices of the respective datasets colored by niche labels (brain, liver and lung) and cell density (lung and colon). **c**, The SpatialCorpus-110M is harmonized and mapped to orthologous gene names, as well as human and mouse-specific genes, to create the input for Nicheformer pretraining. We harmonized metadata information across all datasets, capturing species, modality and assay. **d**, Each cell's gene expression profile and metadata are fed into a gene-rank tokenizer to obtain a tokenized representation for each cell. The tokenized cells serve as input for the Nicheformer transformer block to predict masked tokens. Finally, the Nicheformer embedding is generated by aggregating the gene tokens (Methods). **e**, The pretrained Nicheformer embedding is visualized as UMAP colored by modality. The UMAP shows a random 5% subsample of the entire Nicheformer embedding (*n* = 4,903,086). NA, not applicable.

capture spatial variation. Similarly, we evaluated training with only human or only mouse data. Models trained on one organism performed poorly on the missing organism but outperformed those trained on the opposite organism (Extended Data Fig. 2c). Importantly, this result is not influenced by the sheer number of cells since all models are trained with the same number of cells; the only difference is the diversity of the data. These findings are statistically significant (analysis of variance, adjusted for false discovery rate (FDR); Extended Data Fig. 2a,c) and highlight the importance of data diversity in model training for optimal performance across context[33].

**Model evaluation and downstream tasks.** Current transformer-based single-cell models are used for either gene-level tasks (for example, gene regulatory networks inference, perturbation effects) or cell-level tasks (for example, cell-type annotation, batch integration)[21–23]. By incorporating dissociated and spatial scale into a single model, Nicheformer enables a new class of spatially aware tasks, where previous models primarily only focused on disassociated ones (Supplementary Table 2). These include predicting human-annotated niches, tissue regions and spatial compositions—biologically meaningful and nontrivial problems (Fig. 1b and Methods). For the spatial label prediction tasks, we also evaluated the model's uncertainty regarding the predicted labels (Methods). For spatial composition tasks, we defined a distance-based spatially homogeneous niche around each cell and asked the model to predict local density or cell-type composition. The tasks are formulated as prediction problems operating on Nicheformer's pretrained embedding (Fig. 1e), which differ from typical integrated spaces by capturing a cross-modality, cross-tissue and cross-species representation suited for downstream inference.

**Model transfer learning.** We evaluate Nicheformer in both linear-probing and fine-tuning settings. In both cases, a linear head is trained for the specific prediction task, with fine-tuning additionally updating the transformer's parameters. Linear probing—due to its simplicity—highlights the intrinsic biological signal captured by the learned Nicheformer embedding (Fig. 1e).

## SpatialCorpus-110M, a large-scale, cross-organ and cross-species pretraining dataset for single-cell and spatially resolved transcriptomics

To pretrain Nicheformer, we assembled SpatialCorpus-110M—a large harmonized corpus of single-cell and spatially resolved transcriptomics data to date. It includes 57.06 million dissociated cells and 53.8 million spatial cells across human and mouse tissues.

The dissociated portion builds upon the CellXGene CENSUS database (33.47 million cells; Methods), which we extended by an additional 180 datasets across 73 different tissues, containing 17 solid organs, 18 cell lines and various additional tissue junctions in human and mice, with harmonized ontologies and metadata (Fig. 2a). These additional dissociated datasets have been collected through the Gene Expression Omnibus (GEO)[34], sfaira[35] and the Human Cell Atlas (HCA) data explorer[36] (Supplementary Table 3 and Methods). Altogether, the dissociated collection of SpatialCorpus-110M comprises cells from over 6,000 different donors and technical or biological replicates.

For spatial transcriptomics, we curated image-based spatial datasets, specifically MERFISH[37] (Vizgen MERSCOPE), 10x Genomics Xenium, Nanostring CosMx[38] and ISS[39] data (Fig. 2b and Supplementary Table 4), sourced from publications as well as via the Vizgen data release[40] (18.8%) and the 10x Genomics data resource[41] (13.7%). It covers 15 tissues from 158 individuals or animals and over 10,600 tissue sections. Most cells originated from the brain (60.46%, $n = 32,146,779$ cells) and the lung (9.95%, $n = 3,199,548$ cells). A large proportion of the publicly available spatial omics datasets we collected are not annotated (55.23%). We included both healthy samples (64.07%) and cancer

samples (31.98%) to enable Nicheformer to learn tumor–immune microenvironment contexts.

For all datasets in the SpatialCorpus-110M, we curated metadata, such as assay, sex, organism and tissue, based on the original publications by using official ontology term identifiers (Fig. 2c and Methods). To harmonize features across species, tissues and assays, we first converted all gene symbols to ENSEMBL gene IDs using pyEnsemble[42]. Then we used BioMart[43] through the official Ensembl releases[44] to match orthologous genes between species, yielding 20,310 unique gene tokens: 16,981 orthologous, 151 mouse-specific and 3,178 human-specific genes.

Importantly, we did not integrate datasets into a unified latent space. Our goal was to preserve biological and technical variability while offering a large-scale resource for model training. Like CellXGene, SpatialCorpus-110M provides curated raw inputs, allowing researchers to choose their own normalization and integration strategies.

## Nicheformer learns sex-related differences in gene–gene dependencies in MERFISH mouse brain data

Understanding the internal mechanisms of transformer models helps uncover whether their attention patterns reflect biologically meaningful features. We investigated Nicheformer's attention matrices with two objectives: (1) to examine if its layers develop generalizable structures across tissues and modalities, and (2) to test whether attention reflects biological variation.

To assess general layer organization, we analyzed attention across all heads and layers for 2,000 cells from multiple datasets in SpatialCorpus-110M: male and female MERFISH mouse brain samples[8], the liver and lung CosMx datasets[38] used for downstream tasks (Methods) and a scRNA-seq measured brain dissociated dataset[9] (Methods). Our analysis suggests a hierarchical division across Nicheformer's layers: early layers distribute their attention more broadly, with no clear prioritization of individual tokens; middle layers exhibit a sharp attention toward specific genes (Fig. 3b), likely capturing biologically relevant relationships; and final layers consistently focus on contextual tokens (Fig. 3a and Extended Data Fig. 3a,b). This structured pattern of attention is robust across all analyzed tissues and modalities, indicating that Nicheformer learns a hierarchical representation that generalizes beyond a single dataset. We confirmed significance with a Mann–Whitney U-test comparing attention distributions (corrected with Benjamini–Hochberg FDR; Extended Data Fig. 3c,d).

At head level, some attention heads maintain consistent functional roles across tissues and modalities, such as prioritizing highly expressed genes, regardless of whether the dataset originates from brain, liver, lung or dissociated cells (Extended Data Fig. 4a). Others varied by modality, suggesting modality-specific specialization (Extended Data Fig. 4b). We also observed heads with strong self-attention patterns (visualize as strong diagonal attention scores), while some show off-diagonal patterns, likely reflecting coexpression (Extended Data Fig. 4c). These findings highlight the diverse range of attention behaviors that Nicheformer develops when processing complex biological data. These observations echo findings in large language models, where specific attention heads acquire well-defined functions, such as induction heads that detect repeated patterns in sequences[45] or successor heads that track sequential dependencies[46]. While mechanistic interpretability in biological foundation models is still in its early stages, our results suggest that Nicheformer exhibits a similar specialization, with certain heads consistently attending to biologically relevant features across datasets.

Understanding biological variation across conditions is central to single-cell analysis. We assessed whether Nicheformer captures meaningful biological variations—in this case, sex-specific patterns—in these attention mechanisms by analyzing attention patterns in male and female MERFISH mouse brain datasets from the SpatialCorpus110-M[8] (Fig. 3c–e). Both datasets share common coordinate framework

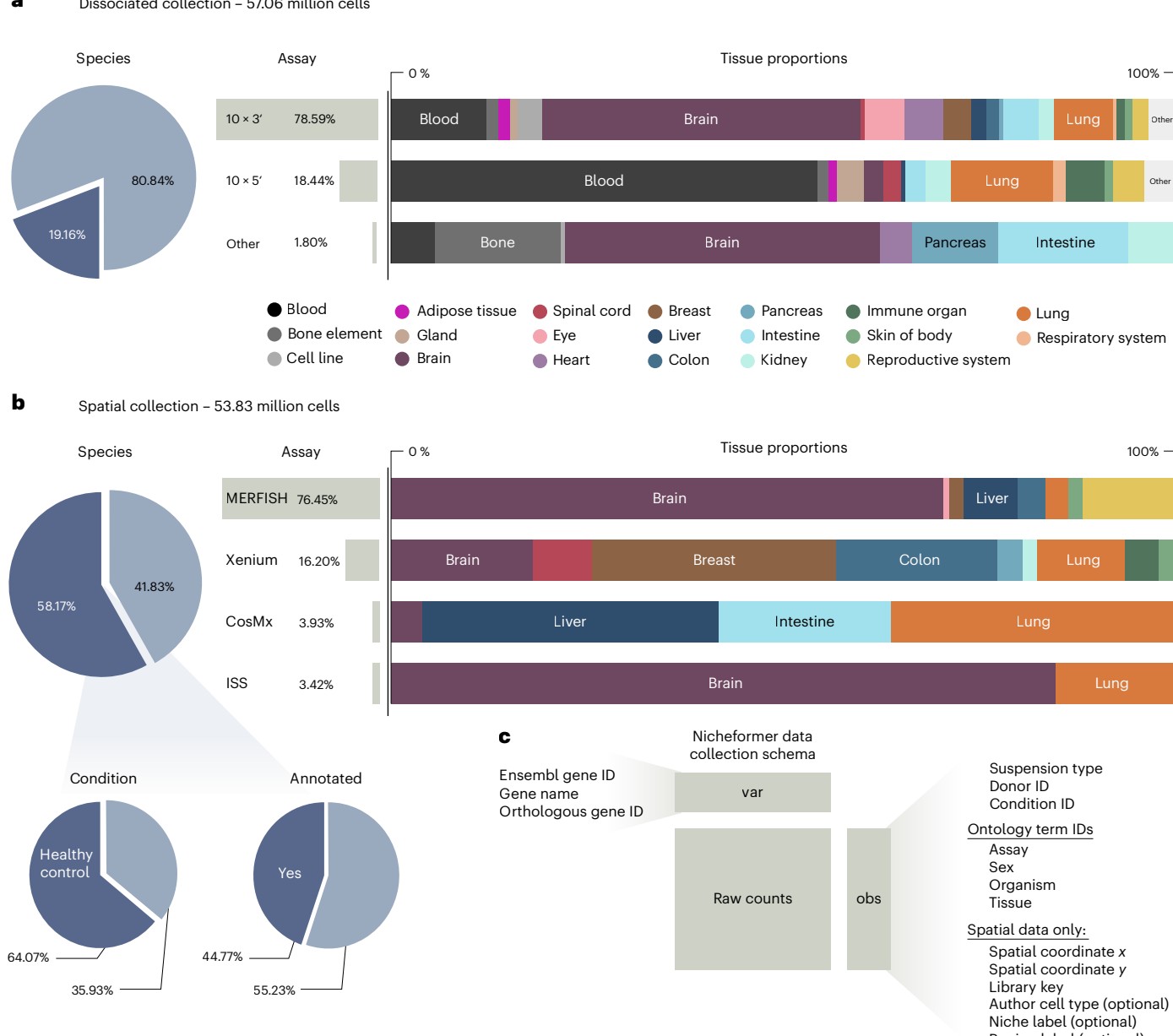

**Fig. 2 | Overview of the SpatialCorpus-110M collected for training Nicheformer. a**, The dissociated single-cell genomics data collection contains 57.06 million human and mouse cells. The collection includes cells from 17 different organs, 18 different cell lines, blood, bone elements, tissue junctions and other anatomical entities, grouped by primary solid organs to simplify visualization and analysis. **b**, The spatial transcriptomics data collection contains 53.83 million targeted spatially resolved cells obtained from humans and mice. The collection comprises four different profiling technologies across 15 different solid organs. **c**, The SpatialCollection-110M was collected with harmonized metadata defined in the Nicheformer data collection schema (Methods). Metadata were harmonized both on the gene level and on the cell level depending on modality.

(CCFv3)[47] annotations, allowing for tagged analysis of the anteroventral periventricular nucleus (AVPV), known for sex-dependent morphology and gene expression[48].

We analyzed all attention matrices from 2,000 AVPV cells per sex, focusing on ten genes previously reported as sexually dimorphic[49–51], and comparing the attention paid to the predefined set of genes against the attention paid to 100 randomly selected genes. We do the analysis both for all cells in the AVPV section and for just HY GABA cells, a small population of cells in the AVPV that modulate the firing of the different glutamatergic neurons in the AVPV that stimulate the synthesis of gonadotropins[52]. We identify key differences between the male and female cells (Fig. 3f,g). The first eight layers had the greatest average attention differences for both sexually dimorphic genes (SDGs) and

100 random genes not directly linked to sex-specific differences in the brain (Extended Data Fig. 4d,e). In contrast, layers nine and ten show high maximal attention value differences for SDGs, when performing differential testing on the attention weights between those two groups, especially for HY GABA cells (Fig. 3h,i). This suggests that specific attention heads in these layers capture subtle sex-specific cues. The contrast between the average and the maximum attention difference indicates that the sex differences are captured by a subset of the attention heads, with at least one of the 16 attention heads showing a stronger focus. This contrast between the average and the maximum difference in attention also holds for genes in the random set (Extended Data Fig. 4f,g). Furthermore, six of the ten genes with the highest attention differences between sexes (*Adgrf5*, *Nfib*, *Pou6f2*, *Rgs4*,

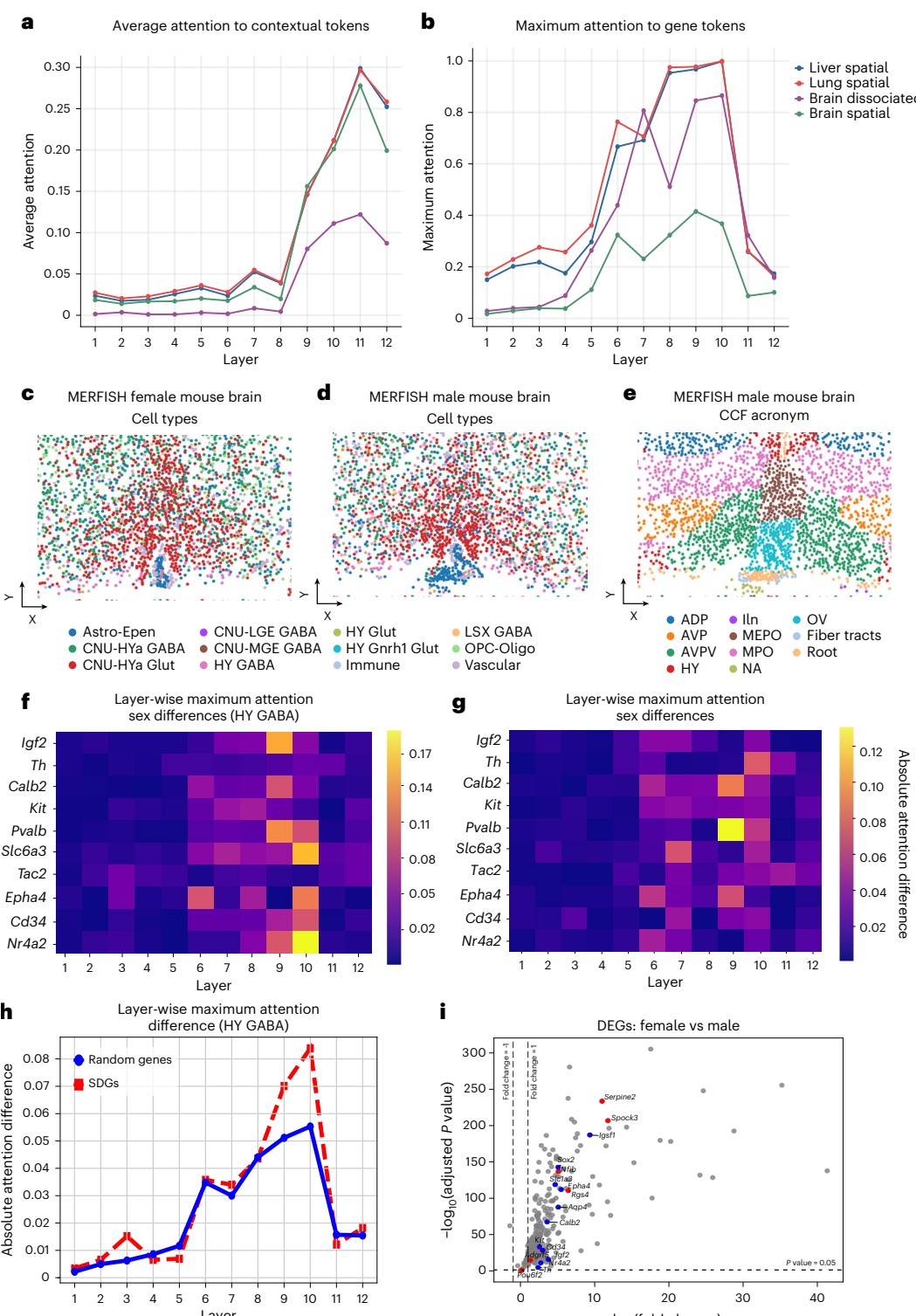

**Fig. 3 | Nicheformer identifies gene–gene dependencies between male and female MERFISH mouse brain sections. a**, Analysis of layer-wise attention patterns reveals that Nicheformer's later layers consistently pay more attention to contextual tokens across all tissues and modalities, demonstrating a clear and robust hierarchical processing pattern. **b**, Maximum layer-wise attention paid to gene tokens. For all tissues and modalities, Nicheformer's middle layers pay the most attention to the gene tokens. **c**, Single cells resolved in space on an example slice (*n* = 2,292 cells) of the MERFISH female mouse brain dataset with cell-type label superimposed. **d,e**, Single cells resolved in space on an example slice (*n* = 2,269 cells) of the MERFISH male mouse brain dataset with the cell-type label (**b**) and CCF acronym label (**c**) superimposed. ADP, anterodorsal preoptic nucleus; AVP, anteroventral preoptic nucleus; HY, hypothalamus; MB, midbrain; MEPO, median preoptic nucleus; MPO, medial preoptic nucleus; NA, nucleus accumbens;

IIn, second cranial nerve; OV, organum vasculosum laminae terminalis. **f,g**, Absolute difference of layer-wise attention scores between male and female MERFISH mouse brain sections show per transformer block of the SDGs considering just the HY GABA cells (**d**) and the entire AVPV section (**e**). **h**, Maximum layer-wise attention difference across layers between male and female HY GABA cells. The attention paid to random genes and SDGs is equal across all layers except in layers 9 and 10, where there is an increment in the maximum attention paid to the SDGs in comparison to the attention paid to the random set of genes. **i**, Volcano plot showing the differentially expressed genes (DEGs) highlighting the genes with highest attention difference between sexes (red), and highlighting the SDGs (blue). The genes found with highest attention differences are not among the most differentially expressed. *P* values were obtained from two-sided Wald tests and adjusted for multiple comparisons using the Benjamini–Hochberg procedure.

*Serpine2*, *Spock3*) have not previously been reported to have sexually dimorphic expression in the brain and some were not differentially expressed between the male and female brain section (Fig. 3i), yet they play roles in development, G-protein-coupled receptor regulation or the extracellular matrix—functions relevant to AVPV biology in which we expect to see sex differences. These effects are likely due to interaction patterns with both known dimorphic genes and others not included in the panel (for example, *Kiss1*, *Gnrh*, *Esr1*). Notably, Nicheformer's ranked tokenization and attention mechanisms enable robust differentiation without requiring matching expression depth, highlighting a key strength of the model.

## Nicheformer allows transferring spatially resolved cell-type, niche and region labels onto unseen data

Dissociated single-cell atlases excel at mapping cell-type diversity, typically defined by stable molecular states across tissues. However, cell types are defined ignoring the spatial context, which provides additional value for understanding cellular microenvironments[53]. Spatially resolved single-cell genomics allows us to augment cell-type definitions by incorporating neighborhood gene expression and histological structure, defining cell niches. These are spatially dependent, local tissue structures (for example, immune or tumor niches), often nested within broader tissue regions, which reflect higher-order spatial organization.

Transferring labels between dissociated and spatial data is challenging due to limited gene overlap[54], and modality-specific methods are not designed to learn from reference atlases at the scale of hundreds of million of cells. Nicheformer addresses this by leveraging the SpatialCorpus-110M to enable scalable annotation transfer.

We evaluated Nicheformer on a large MERFISH mouse brain dataset[8], where 17 different brain regions and 8 distinct tissue niches (Fig. 3a) are labeled (Extended Data Fig. 5a–c). We tested linear probing—linear head over the frozen Nicheformer embeddings (Extended Data Fig. 5e,f)—and fine-tuning approaches for both labels for unseen, held-out tissue sections from the MERFISH mouse brain dataset, measuring one male mouse brain (Extended Data Fig. 5a–d). Compared to embeddings from PCA and scVI (trained on either the brain dataset or subsets of SpatialCorpus-110M; Methods), and to foundation models (Geneformer, scGPT, UCE, CellPLM), Nicheformer achieved the highest macro F1 scores (Fig. 4b and Extended Data Fig. 6a,b). While PCA with a large number of components offers a good performance, practically on par with using a linear probe on top of Nicheformer's representations, or even surpassing it in the case of region prediction, it still fell short of the fine-tuned Nicheformer model (Extended Data Fig. 7a,b). The differences between Nicheformer and competitors were statistically significant as derived from *t*-tests between Nicheformer and the best-performing comparison method (Extended Data Fig. 6a,b).

We performed a similar analysis on a randomly held-out test set of the CosMx human liver dataset defining tissue niches as different zonations between the central and portal veins (Extended Data Fig. 8a–c). Again, fine-tuned Nicheformer led in terms of macro F1 score. However, linear probing underperformed compared to scVI and PCA trained on the training set of the liver dataset (Extended Data Fig. 8f). We hypothesized that this is related to the insufficient model capacity due to limitations regarding a relatively low overall abundance in the SpatialCorpus-110M (Fig. 2a,b). Extended pretraining on liver data improved performance, suggesting undertrained tissues can benefit from additional fine-tuning (Extended Data Fig. 8f). Surprisingly, we observed that in Nicheformer models trained with just ~1% data, there was no such a drop in performance. Additionally, we observed that the model trained on a smaller dissociated subset (1%) performed slightly better than one trained on a larger subset (3%), which also supports the hypothesis that 'compute per sample' is important (Supplementary Note 1).

We next assessed label transfer between spatial and dissociated data, using Nicheformer to map MERFISH-defined cell types to scRNA-seq motor cortex cells (Fig. 4c,d)[9]. We find that Nicheformer correctly selects the nine motor cortex-related cell types of the overall 33 cell types present in the MERFISH mouse brain dataset (Fig. 4e and Extended Data Fig. 8l). When calculating classification uncertainty based on the overall predicted distribution generated by the model (Methods), the predicted cell-type labels show overall a high agreement and low classification uncertainty (Fig. 4e,l) with the original cell-type annotations. Mostly, all cell types were correctly matched, independently of their abundance in the cell dissociated dataset (Fig. 4h). Some deep-layer glutamatergic neurons were misclassified as midbrain glutamatergic, possibly due to transcriptional heterogeneity and subtype imbalance in MERFISH data. For niche labels, Nicheformer correctly predicted all expected assignments with low uncertainty for non-neuronal and inhibitory neurons, but higher uncertainty for excitatory subtypes (Fig. 4f,j and Extended Data Fig. 8j). Misclassifications likely stem from overlapping spatial structures. For region labels, most cells were correctly predicted as isocortex (Fig. 4g,k and Extended Data Fig. 8k). Some spillover into adjacent regions (for example, cortical subplate (CTXsp) and olfactory areas (OLF)) may reflect tissue dissection artifacts. Region prediction was slightly worse for non-neuronal cells, likely due to their lower transcriptional diversity. For extended detailed analysis, consult Supplementary Note 2.

Altogether, this demonstrates Nicheformer's ability to learn powerful cell representations by capturing nuanced spatial information. Linear probing already surpasses existing baselines, highlighting the effectiveness of the representation. Fine-tuning further refines this representation, emphasizing the importance of task-specific adaptation for capturing subtle cellular variations. Notably, Nicheformer enables the direct transfer of spatially aware annotations from spatial to dissociated single-cell data by using a simple linear layer. This capability unlocks new possibilities for analyzing single-cell data across different modalities.

## Nicheformer predicts neighborhood compositions in spatial and dissociated single-cell data

Tissue microenvironments consist of cellular neighborhoods with a diverse composition of cell types. Differences in neighborhood composition have been shown to have an important effect on gene expression and can be associated with cell–cell communication events[6]. Furthermore, the cellular composition of neighborhoods in the tumor microenvironment may hold prognostic value, because immune cell infiltration in the spatial context is a predictor for cancer survival[55]. Here we show that we can leverage Nicheformer's multimodal cell representation to accurately relate changes in gene expression to differences in neighborhood compositions in spatial data and transfer them to dissociated transcriptomes.

We define a cell's 'computational' neighborhood as the set of cells within a fixed radius (Fig. 5a and Methods). The total number of cells composing the neighborhood defines the neighborhood density, and the proportion of cell types in the neighborhood defines the neighborhood composition. This notion is consistent with previous approaches defining a cellular neighborhood[56] and allows for an interpretable evaluation of model results. Generally, the definition of a cell neighborhood can be extended in the future to account for non-isotropic cell neighborhoods that might vary in their cell-type composition and are drivers of similar biological functions with varying sizes across a dataset.

To evaluate Nicheformer's ability to predict neighborhood composition, we focused on three datasets measuring three organs with two different technologies, namely MERFISH mouse brain, CosMx human liver and CosMx human lung. We computed neighborhood compositions at varying resolutions for each of the three datasets separately. The radii were selected to contain, on average, 10, 20, 50 or 100 neighbors (Fig. 5b and Methods). We evaluated Nicheformer both in linear-probing and fine-tuned settings for each dataset and

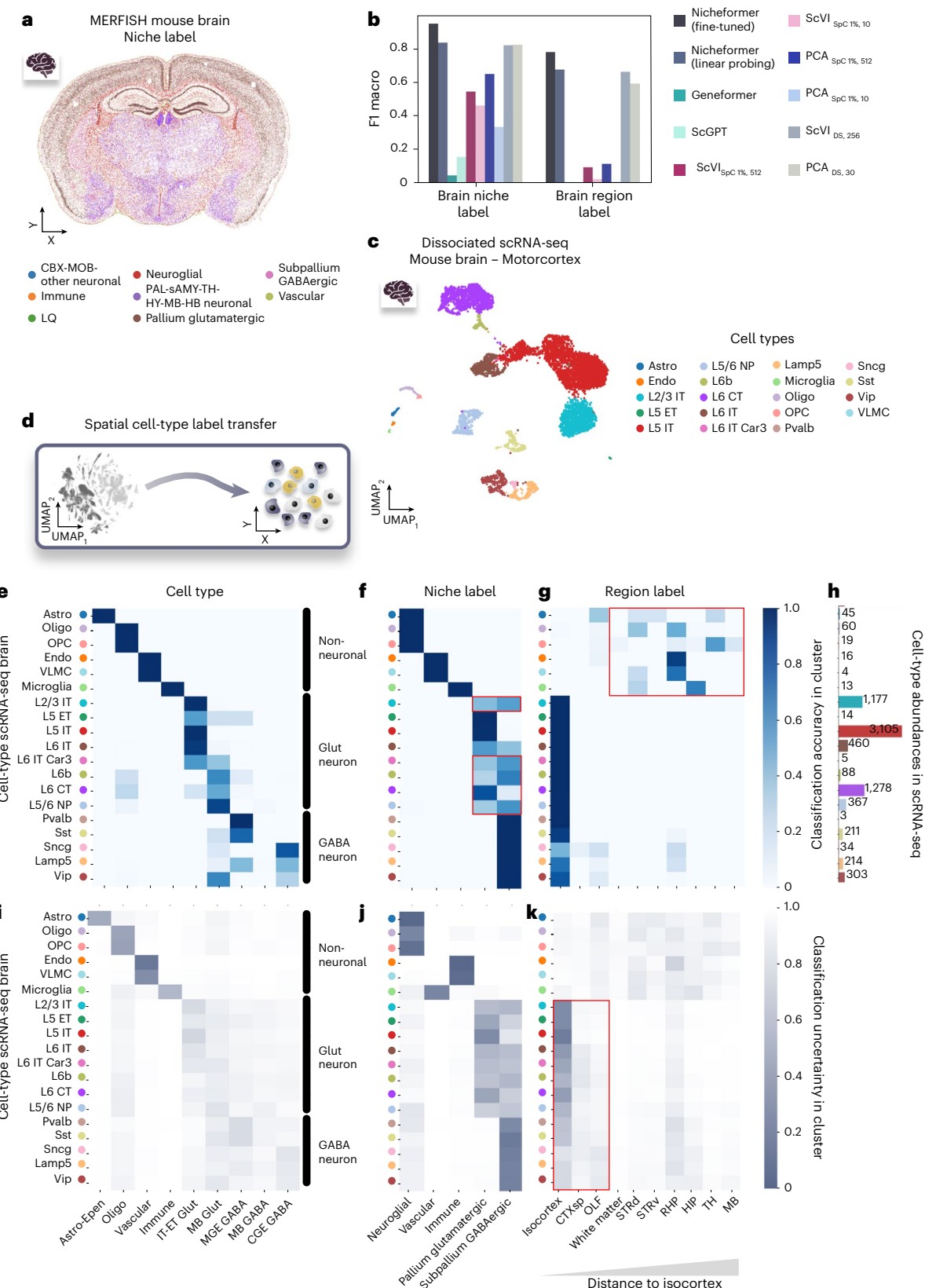

each neighborhood size individually and compared its performance to linear probing on embeddings computed with scVI, PCA, Geneformer and scGPT. We found that fine-tuned Nicheformer systematically outperformed the linear-probing models trained on Nicheformer embedding, Geneformer, scGPT, scVI and PCA, independently of the number

of principal components used, even though PCA's performance notably improves with more principal components (Extended Data Fig. 7a,c,d), for this task on all three organs in terms of mean absolute error. Likewise, for UCE and CellPLM, which we evaluated by training a linear layer on their embeddings, we also found that linear probing with

**Fig. 4 | Nicheformer accurately transfers cell-type, niche and region label to unseen spatial and dissociated data in the brain. a**, Single cells resolved in space on an example slice ($n$ = 114,396 cells) of the MERFISH mouse brain dataset with niche label superimposed. **b**, Test-set F1 macro of niche and brain region label prediction of the fine-tuned Nicheformer model, the linear-probing model and a linear-probing baseline computed based on embeddings generated with Geneformer, scGPT, scVI and PCA, respectively. For scVI and PCA, both embeddings generated from a random 1% subset of the SpatialCorpus as well as embeddings generated from the training set of the original dataset are evaluated. **c**, UMAP of dissociated scRNA-seq dataset with original author cell-type label superimposed. ET, extratelencephalic neurons; IT, intratelencephalic neurons; CT, corticothalamic neurons; NP, near-projecting neurons; OPC, oligodendrocyte precursor cells. **d**, Nicheformer can transfer spatial niche and region labels onto dissociated single-cell data. **e**, Nicheformer accurately classifies cells from the dissociated motor cortex to relevant cell types ($n$ = 9 of 33 distinct ones in the classifier) trained on the whole mouse brain MERFISH dataset. **f**,**g**, Nicheformer correctly projects dissociated single cells to niche

(**f**) and region (**g**) labels to provide spatially dependent labels. STRd, dorsal striatum; STRv, ventral striatum; RHP, retrohippocampal region; HIP, hippocampal formation; TH, thalamus. **f**, Nicheformer misclassified parts of layer 2/3 (L2/3) IT neurons as residing in the subpallium GABAergic niche (highlighted in the red box). Additionally, the deep cortical excitatory neurons L6b, L6 CT, L6 IT, and L6 IT Car3 (highlighted in the red box) should be classified as pallium glutamatergic niche instead of subpallium GABAergic by Nicheformer. **g**, Most of the non-neuronal cells (84.7% of all non-neuronal cells, $n$ = 133) were misclassified as not belonging to the isocortex or the adjacent brain regions (highlighted in the red box). **h**, Cell-type abundances in the scRNA-seq dataset measuring the primary motor cortex in the mouse. **i**–**k**, Classification uncertainty of label transfer of the dissociated scRNA-seq dataset to the MERFISH mouse brain data for cell-type label (**i**), niche label (**j**) and region label (**k**) with a value of 0 representing a high uncertainty and 1 being a lower uncertainty, that is, high certainty. **k**, Observed high uncertainty for parts of the Glut and GABA neurons for the region prediction of the isocortex, CTXsp and OLF, which are neighboring brain regions.

Nicheformer outperformed both methods across all three datasets (Extended Data Fig. 6a,c,d). Statistical tests ($t$-test) to assess the statistical significance of the results were performed, with positive results (Extended Data Fig. 6a,c,d). Notably, the linear-probing models trained on Nicheformer embeddings also outperformed all other methods, except for the fine-tuned Nicheformer (Fig. 5c). However, for bigger radius sizes in the liver dataset, the scVI models trained in a subset of SpatialCorpus-110M performed on par with fine-tuned Nicheformer. We believe this to be related to the previous classification results in the same dataset (Extended Data Fig. 8f). Interestingly, Nicheformer's performance increased with neighborhood size in the case of the brain datasets. In the liver, we observed a stronger performance trend, which might be related to transcriptional patterns of zonation and structural components in the liver[57]. For the CosMx liver dataset, we additionally evaluated whether a multitask multilayer perceptron (MLP) would allow the prediction of all neighborhood sizes jointly (Methods). We observed that a multitask MLP did not outperform a neighborhood size-specific linear-probing model or the fine-tuned Nicheformer model, indicating that downstream tasks should be evaluated separately (Extended Data Fig. 8g).

To understand the model's behavior and performance in more detail, we additionally assessed the fine-tuned Nicheformer performance for each cell type separately in the MERFISH mouse brain dataset (Fig. 5d and Methods). We computed the absolute error between predicted and true neighborhood compositions across all four neighborhood sizes and sorted the result based on the median values per cell type. We found that the most accurately predicted cell types in terms of absolute error are also within the 8 (of 33) most abundant cell types in the MERFISH mouse brain dataset. In contrast, the 4 cell types for which Nicheformer performed worse are in the 14 least abundant cell types (Fig. 5d). For example, highly abundant cell types predominantly from cortical layers (IT-ET Glut, NP-CT-L6b Glut) are structurally organized

in the brain and have a quite homogeneous neighborhood composition. Those two factors help to explain the very accurate Nicheformer predictions. Similarly, CB Glut cells are based in the cerebellum, an area with very high cell density[58] and high neighborhood homogeneity. Even though they have a lower abundance in the overall dataset, Nicheformer accurately predicted their neighborhood composition (Fig. 5d). On the other hand, Nicheformer shows a lower performance on cell types predominantly found in the midbrain or hypothalamus (MB GABA, MB, Dopa, HY Glut, Hy MM Glut). These cell types are relatively rare cell types in the given dataset and are located in more diverse and complex tissue layouts and show a greater variety of neighboring cell types[8]. This indicates that regionally diverse and less abundant cell types in the pretraining corpus are harder to predict for the Nicheformer model. The performance differences might be related to the structural properties of the brain regions as well as their varying cell-type compositions and abundance in the dataset. We further observed a relatively good performance of Nicheformer for the neighborhood composition prediction of immune cells, despite their relatively low abundance and their lack of regional specificity in the brain. Immune cells are scattered across the brain and accomplish very specific but differing tasks ranging from regulating synaptic plasticity, and immune surveillance, to preventing excitotoxicity[59]. Interestingly, the Nicheformer embedding of the immune cells in the MERFISH mouse brain data preserves the regional information of those cells and region-specific subclusters can be identified (Fig. 5e).

To assess whether our results generalize across organs and technologies, we performed a similar analysis for the CosMx human liver dataset, evaluating the overall cell-type performance in the task of predicting the neighborhood composition across resolutions (Extended Data Fig. 8h). Again, we observed that Nicheformer's performance heavily depends on the cell-type abundance in the dataset and the regional specificity of the individual cells, for example, we

**Fig. 5 | Nicheformer accurately predicts neighborhood compositions at multiple niche resolutions for the brain, liver and lung. a**, We define the neighborhood of a cell as its local neighborhood given a radius and an index cell. The neighborhood cell density is then defined by the number of cells in the neighborhood, and the neighborhood compositions are the proportions of neighboring cell types. **b**, Neighborhoods are computed at multiple resolutions resulting in different neighborhood size distributions. Each barplot shows the distribution of the number of neighbors across the brain, liver and lung datasets. We extract neighborhoods with the mean number of neighbors 10, 20, 50 and 100 for each dataset. Neighborh., neighborhood. **c**, The fine-tuned and linear-probing Nicheformer models outperform for brain and lung linear-probing models trained on Geneformer, scGPT, scVI and PCA embeddings in terms of mean absolute error across all neighborhood sizes. Still, it struggles to outperform all benchmarks in liver, where scVI models are

very competitive. This is an issue related to the previous liver performance reported in the previous section (Extended Data Figs. 2a and 8f). **d**, Left, Fine-tuned Nicheformer performance on the MERFISH mouse brain data grouped by index cell type. Shown are the absolute error values between predicted and observed neighborhood composition vectors for held-out test cells. For each box in **d**, the centerline defines the median, the height of the box is given by the interquartile range (IQR), the whiskers are given by 1.5 times the IQR, and outliers are given as points beyond the minimum or maximum whisker. Center, Index cell-type abundances in the entire MERFISH mouse brain dataset. Right, UMAPs of MERFISH mouse brain Nicheformer embedding with the selected index cell type as color superimposed. **e**, UMAP of the Nicheformer embedding of all immune cells in the MERFISH mouse brain dataset with region label as color superimposed.

saw a lower absolute error for hepatocytes compared to circulating immune cells (Extended Data Fig. 8h). Hepatocytes are predominantly found in highly structured cellular microenvironments and show strong spatial patterns in their gene expression[60], while liver-resident immune cell populations were shown to be mobilized under certain circumstances, hence their regional specificity might be lower compared to other cell types[61]. This indicates that the Nicheformer embeddings can be useful to identify and understand region-specific and niche-specific structures and differentiate cell types that show a higher regional specificity.

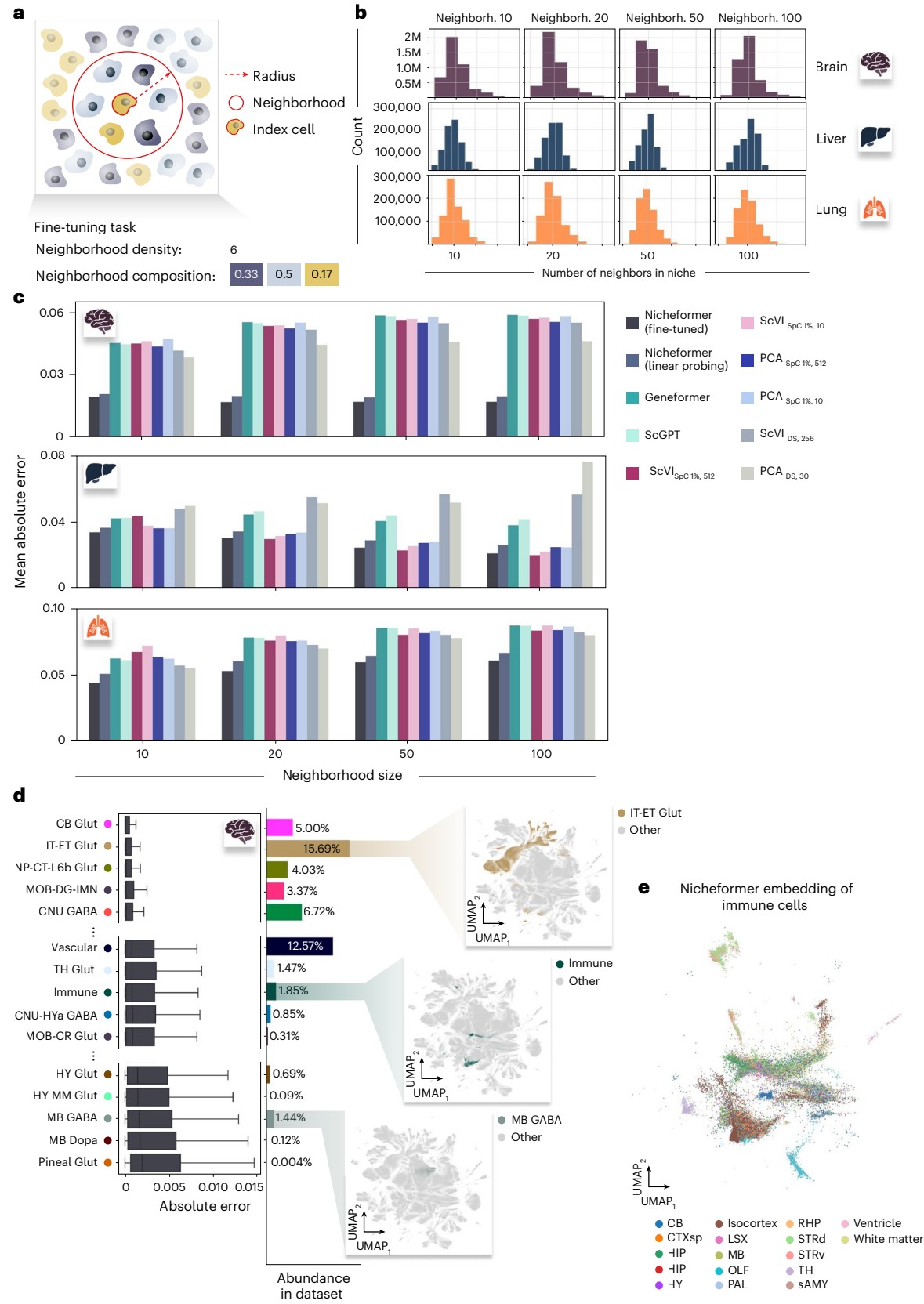

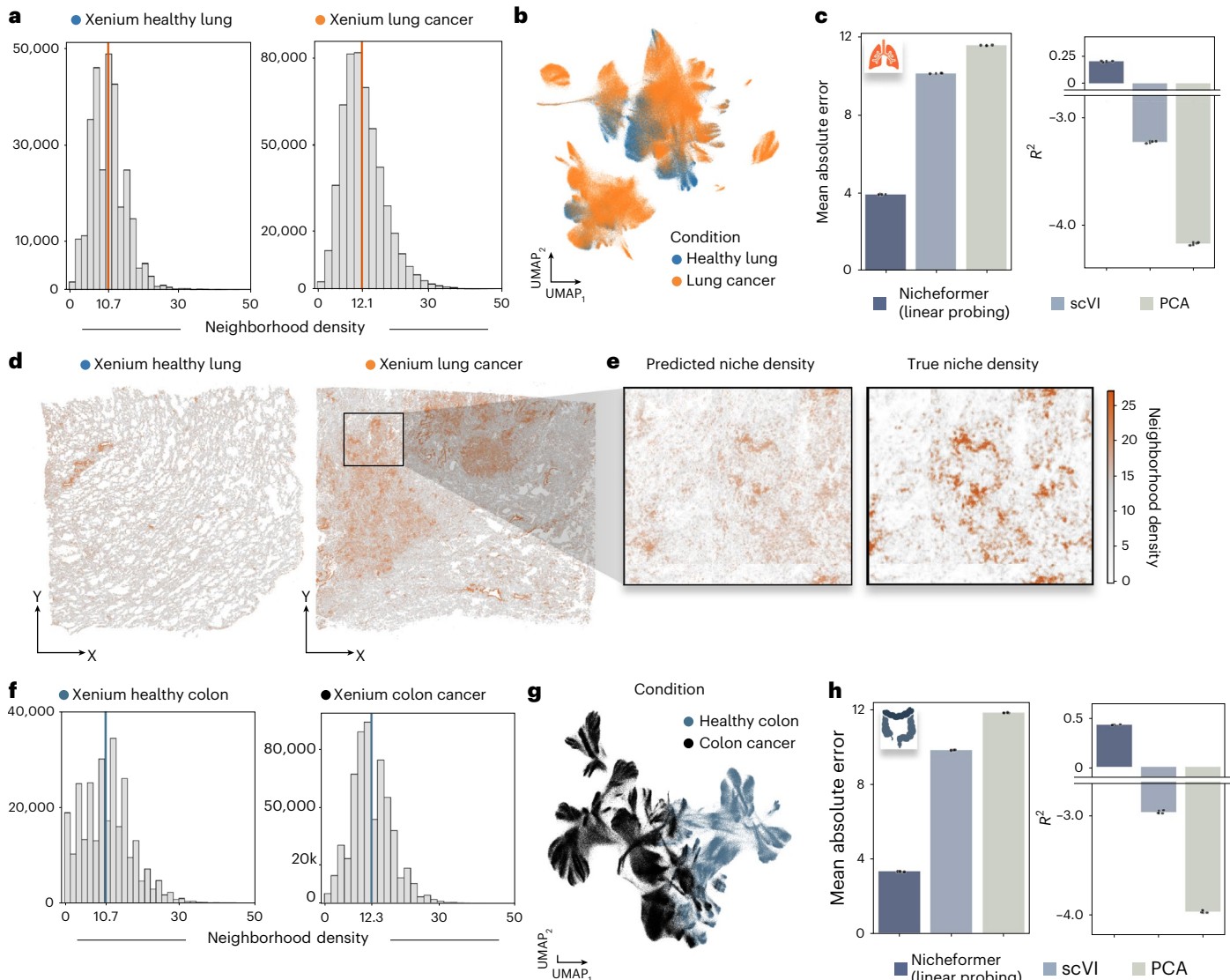

**Fig. 6 | Nicheformer accurately predicts changes in cellular neighborhood density in the lung and colon. a**, Barplot of cellular neighborhood densities split by condition for the Xenium human lung dataset. **b**, UMAP of the Nicheformer embedding of the Xenium human lung dataset colored by condition. **c**, Mean absolute error and $R^2$ for the cellular neighborhood density prediction task for a Nicheformer linear-probing model and linear-probing models trained on scVI and PCA embeddings. Data are presented as mean values with error bars showing the standard deviation, using three random seeds. **d**, Spatial allocation of cells in the Xenium human lung dataset colored by predicted cellular neighborhood density in the healthy and diseased lung. **e**, Predicted-versus-true cellular neighborhood density for a zoomed-in section of the Xenium lung cancer section. **f**, Barplot of cellular neighborhood densities split by condition for the Xenium human colon dataset. **g**, UMAP of the Nicheformer embedding of the Xenium human colon dataset colored by condition. **h**, Mean absolute error and $R^2$ for the cellular neighborhood density prediction task for a Nicheformer linear-probing model and linear-probing models trained on scVI and PCA embeddings. Data are presented as mean values with error bars showing the standard deviation, using three random seeds.

## Nicheformer infers cellular niche density in unseen data

Beyond cellular niche labels and neighborhood composition, we asked whether local cell density is encoded in a cell's expression profile. It is long known that cell density can strongly affect growth behavior in vivo and in culture; also, increased cell density is a key feature of the formation of the tumor microenvironment, which leads to the creation of a hypoxic environment and depletion of infiltrating immune cell populations[62]. For example, in colon cancer, it was shown that the immune cell density is associated with patient survival and can be used for tumor–immune patient stratification for improved anticancer therapy[63]. In non-small-cell lung cancer[64], immune cell density and neighborhood compositions were used to stratify specimens into groups associated with clinical outcomes.

We tested whether Nicheformer accurately predicts the neighborhood density in a Xenium lung dataset measuring an adult human healthy lung section and a section with invasive adenocarcinoma from a second patient[65], and in a Xenium formalin-fixed paraffin-embedded-preserved healthy and diseased colon with stage 2A adenocarcinoma from two different patients[65]. Consistent with literature observations[63,64], we observed a higher average cellular density in the cancer sections (colon, 12.3 cells; lung, 12.1 cells) compared to healthy tissue (colon, 10.7 cells; lung, 10.7 cells) when extracting cellular neighborhoods at the same radius (Fig. 6a,f and Methods).

We first computed Nicheformer embeddings for both datasets by generating a forward pass through the Nicheformer pretrained model (Fig. 6b,g). Additionally, we embedded the two datasets with scVI, and PCA (Methods). The three resulting embeddings for the datasets were then used as input for a linear-probing regression model to predict the cellular neighborhood density for each cell. The linear-probing models trained on the scVI and PCA embeddings failed to correctly

predict the mean density and performed worse than random prediction, resulting in negative $R^2$ values for both tissues. Interestingly, the linear-probing model trained on the Nicheformer embedding outperformed the other two models in terms of mean absolute error and $R^2$ (Fig. 6c,h) and was able to accurately predict a higher cellular density in the tumor regions and denser tissue structures in the Xenium lung dataset (Fig. 6d). This demonstrates that the Nicheformer embeddings are able to capture neighborhood density variation solely on transcriptome information better than the baselines. Nicheformer's ability to infer cellular neighborhood density in healthy tissue and cancer tissue can be useful to inject spatial relationship information in dissociated data to further characterize cell-state variation in systems such as the tumor microenvironment.

## Discussion

Nicheformer demonstrates the potential of multiscale foundation models for dissociated single-cell and spatial transcriptomics data. By leveraging the SpatialCorpus-110M and evaluating the model in different spatially informed downstream tasks and assessing the model's prediction uncertainty, we demonstrate that Nicheformer captures complex relationships between gene expression and spatial context. We introduce a newly designed set of downstream tasks designed explicitly for spatial data analysis, in which Nicheformer consistently outperforms baseline models, including foundational models trained only on scRNA-seq data such as GeneFormer, UCE and scGPT, and also models trained on spatial data such as CellPLM, highlighting its effectiveness in learning a cell representation that is able to predict spatial features and the need to train on multiscale and diverse datasets to capture the intricate spatial relationships present in tissue organization. These results strongly suggest that spatial context can be effectively inferred from transcriptomics data using Nicheformer. To further understand how Nicheformer processes information, we analyzed its attention mechanism, finding that different layers attend to distinct features. We identified specific attention heads that remain robust across modalities and tissues, as well as others that adapt to these variations. We also explored how Nicheformer captures biological conditions through its attention patterns. Additionally, we conducted an analysis of the performance of models pretrained on different data subsets to evaluate the impact of various modalities and organisms on its performance. Our results highlight that broad coverage in training data is essential for achieving robust performance across diverse contexts. Further, Nicheformer paves the way for transferring spatial information to large collections of dissociated single-cell data, which opens the door for more nuanced analyses of cellular function in the tissue environment in silico.

A cell integrates its spatial context, that is, its cellular neighborhood by cell interaction and communication, which is reflected in the cell's transcriptomic profile. This property has been used successfully to learn cell-type communication profiles from coexpressed receptor–ligand interactions[66], to reconstruct spatial gene expression from spatial context and anchor points using optimal transport[67,68] and to determine cell interactions beyond known receptor–ligands via graph neural networks[56]. With Nicheformer, we build upon these results and show that we can predict spatial context from a cell's gene expression profiles alone with consistent accuracy. We found that, for example, immune cell neighborhoods in the brain are most likely encoded in the gene expression profiles, making it easier for Nicheformer to understand these differences and relate them to neighborhood composition changes. Extending this analysis to additional tissues has the potential to characterize recurrent immune niches across tissues and organs.

A long-term vision in systems biology has been to create multiscale models, from molecules and cells up to tissue, organs and eventually the whole organism. Nicheformer represents a step toward creating a generalizable multiscale model for single-cell and spatial biology, bridging the gap from the single-cell to the tissue modality.

More generally, it will be necessary to operate on multimodal data to generate a true representation of the cellular state. While spatial transcriptomics captures the cellular microenvironment in tissues well, integrating additional data modalities, such as protein abundance or epigenetic modifications, will provide a more complete picture of the cellular state. The development of multimodal foundation models faces multiple challenges. One key hurdle is the lack of sufficient paired data measured across multiple or even all cellular modalities. However, with the development of new assays and sequencing technologies, we expect the number of multimodal datasets to grow, enabling the development of architectures to model them. Incorporating additional modalities will remain a challenge in the future as, for example, epigenetic modifications, protein abundance and gene expression all have unique characteristics, and effectively combining them in a way that leverages their strengths remains an ongoing research area.

While Nicheformer represents a process for learning general representations for single-cell biology, we acknowledge some limitations of this approach. Firstly, Nicheformer performance depends on the data abundance and transcriptional diversity of the cells under study. Indeed, we showed that Nicheformer's performance for predicting spatial labels and spatial compositions is impacted by cell-type and tissue-type abundance in a spatial transcriptomics dataset. With the ongoing growth in spatial transcriptomics data availability as well as improved throughput thanks to technological advances, we expect that the prediction performance will improve across evaluated tissues. Secondly, Nicheformer does not explicitly incorporate the physical location of a cell during pretraining, limiting its capability to fully leverage the available information on spatial context. We deliberately chose not to include spatial coordinates during pretraining because we wanted to learn a general representation of gene expression variation across both modalities, fully supervised by gene expression alone. Nevertheless, we anticipate that future iterations of Nicheformer will account for spatial relationships of cells by encoding spatial neighbor graphs, for example, and potentially leveraging graph transformer architectures[69] for the pretraining stage on spatial transcriptomics data. Graph transformers excel at modeling relationships between nodes in graphs, making them ideal for capturing nearest-neighbor effects on a cell's transcriptome. Thirdly, the interpretability of the Nicheformer model has not been fully explored. In future iterations, it would be interesting to inspect the learned architecture in order to understand interactions between genes within cells and niches to extract biological mechanistic knowledge, for example, by assessing how gene relationships are associated with cell state across the two modalities under consideration. Additionally, the current strategy excludes metadata tokens from the final cell representation to avoid bias from their high norm (Methods), which can impede label transfer. However, this may limit model expressivity by discarding these tokens entirely. More refined strategies, such as selective integration, could retain relevant context without allowing it to dominate the embedding. We additionally see a need to scale Nicheformer in the number of parameters, pretraining time and dataset size. Characterizing scaling laws for foundation models in genomics has the potential to identify bottlenecks in learning schemes and datasets, thus informing design and pretraining choices for the next generation of models. Finally, we want to highlight the need for more comprehensive benchmarks than the set of spatial tasks presented here, which will help judge extensions and future alternative models. The field of biological foundation models is a novel area brimming with potential. However, unlike more established AI domains, there's a crucial gap in the form of standardized benchmarks for evaluating these models. Establishing robust benchmarks is a critical next step to compare and improve performance, rigorously assess methodological progress and guide future model development to unleash the full potential of foundation models for single-cell biology.

Overall, Nicheformer demonstrates the feasibility of learning a foundational representation able to effectively transfer information from single-cell to spatial genomics and its reverse, paving the way for the next generation of foundation models trained on large heterogeneous collections of dissociated and spatial single-cell data. We describe a set of newly designed evaluations that are explicitly for probing the model's ability to encode spatial context and its transferability to a different modality that can be leveraged as a new benchmark for multimodal foundation models for single-cell and spatial genomics. We believe Nicheformer represents an important progress toward building a general and robust representation of cellular biology phenotypes advancing our understanding of the heterogeneous effects of cellular niches in development and disease. We envision Nicheformer and similar models to actively assist in experimental design through hypothesis generation and experiment selection, ultimately accelerating the pace of scientific progress by helping to choose the next set of most informative experiments. Nicheformer will thus help to guide and design spatial experiments based on scRNA-seq measurements, supporting the upcoming transition from cell to tissue atlases.

## Online content

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

## Methods

### Collection of the SpatialCorpus-110M

**Dissociated data collection.** We collected and combined dissociated single-cell and single-nucleus data from the latest patch of CellXGene[70], 50 additional curated studies available through the sfaira data zoo[35], 150 datasets acquired through the GEO data repository[34,71] and 4 datasets from the HCA data explorer[72].

For the data originating from CellXGene, we used the CZ CellXGene Discover Census[70] v.2023-07-15 and its Python API to download the latest batch of all data available on the census. The CZ CellXGene Discover Census only contains cells from human or mouse, as well as only gene expression measurements obtained via RNA-seq. We additionally only downloaded primary data that were marked with the respective identifier in the Census to ensure that cells are not represented multiple times in our collection. Subsequently, we downloaded the entire cell and gene metadata as well as the raw counts and stored them as H5AD on disk. For additional data acquisition, firstly, we selected human and mouse 10x Genomics technology datasets not present in the latest CellXGene patch from the sfaira data zoo[35] and excluded datasets without publicly available raw count matrices. We then downloaded the selected data through the sfaira interface, removed any cells with less than 200 expressed genes, streamlined the feature space of each dataset to Ensembl release 104 (GRCh38) protein-coding genes, applied sfaira metadata streamlining, and applied the Nicheformer metadata scheme. We stored the data for each study from sfaira as individual H5AD objects on disk.

Secondly, for the acquisition from the GEO data repository, we focused on GEO IDs previously included in the recent scsimilarity[25] preprint publication. After cross-checking this list with the other used data sources to avoid duplicated data, we acquired the necessary metadata from the GEO website and the corresponding publications. We downloaded the count matrices, converted the various data formats into AnnData format and combined them with the collected metadata to save them as individual H5AD objects on disk. We curated ontology term identifiers for species based on the ontology representation of the NCBI organismal taxonomy (NCBITaxon)[73], tissue based on the Uber-anatomy ontology (Uberon)[74,75], sex based on the ontology of phenotypic qualities (PATO)[76,77] and assay based on the Experimental Factor Ontology (EFO)[78]. All ontology terms were obtained through the Ontology Lookup Service (OLS)[79].

Lastly, we followed the same approach for the four HCA data explorer[36] datasets as for the GEO datasets. To make the dataset acquisition process reproducible and available to the community, we have shared scripts for downloading and standardizing all datasets. All data collection-related code can be found at https://github.com/theislab/nicheformer-data/. We additionally implemented a validator to streamline the verification process, ensuring alignment between metadata formats and the data collection schema. A detailed list and overview table of all datasets containing GEO ID, DOI, the number of cells, tissue, assay and author information can be found in Supplementary Table 3.

**Spatial data collection.** The spatial part of the SpatialCorpus-110M consists of datasets measured with image-based spatial transcriptomics technologies, namely CosMx, ISS, MERFISH and 10x Xenium. We collected 60 different datasets across 15 different solid organs. Most of the spatial data collection was collected via the Vizgen data release[40], the 10x Genomics data resource[41] and the CosMx data resource[38]. The remaining datasets were collected through the data resources stated in the original publications. Unpublished datasets were obtained before publication via the original authors. Each dataset was downloaded and stored as individual H5AD files. For each dataset, we collected expression data and associated gene-level and cell-level metadata, but high-resolution images and segmentation masks were not collected and curated. We curated ontology term identifiers for species based on the ontology representation of the NCBI organismal

taxonomy (NCBITaxon)[73], tissue based on the Uber-anatomy ontology (Uberon)[74,75], sex based on the ontology of phenotypic qualities (PATO)[76,77] and assay based on the Experimental Factor Ontology (EFO)[78]. All ontology terms were obtained through the Ontology Lookup Service (OLS)[79]. For Xenium and CosMx assays, official ontology terms are not yet defined, so we replaced them with placeholders. For datasets that did not provide Ensembl gene identifiers, we used pyEnsembl[42] with the Ensembl release 104 (GRCh38) to map gene names to Ensembl gene identifiers and subsequently BioMart[43] through the official Ensembl releases[44] for mapping mouse genes to orthologous gene identifiers. Scripts for acquiring the spatial data are also shared in our GitHub repository. We used the same validator as used for the dissociated datasets to streamline the verification process of the collected metadata. We applied no additional quality control, gene-level or cell-level filtering for the spatial omics datasets beyond the filters applied by the original authors of the publications or the filters automatically applied by the individual spatial transcriptomics technologies. A detailed list and overview table containing the GEO ID, DOI, the number of cells, tissue, assay and author information for the spatial datasets can be found in Supplementary Table 4.

**Datasets used for downstream tasks and evaluations.** Publicly available datasets used for downstream tasks and evaluations were collected in the same way as the other spatial transcriptomics datasets present in the SpatialCorpus-110M. As most of our downstream tasks require cell-type, niche and region label annotations, we focused primarily on annotated and large-scale spatial transcriptomics datasets. We provide a detailed description of those datasets below.

### MERFISH mouse brain

Yao et al.[8] measured 4.3 million cells across 59 tissue sections from one whole male mouse brain using MERFISH with a 500-gene panel. This dataset contains a hierarchical cell-type annotation structured into four nested levels of annotation. We used the 'class_label' field with 33 distinct cell types as input for the Nicheformer niche regression task (Extended Data Fig. 3c), the 'division_id' label, containing seven distinct labels (CBX-MOB-other neuronal, immune, low quality (LQ), neuroglial, PAL-sAMY-TH-HY-MB-HB neuronal, pallium glutamatergic, subpallium GABAergic, vascular) as niche labels (Extended Data Fig. 5b), and the 'clean_region_label' field, containing 17 distinct labels (CB, CTXsp, HB, HIP, HY, isocortex, LSX, MB, OLF, PAL, retrohippocampal region, dorsal striatum, ventral striatum, TH, sAMY, ventricle, white_matter) as the region label (Extended Data Fig. 5a) for the Nicheformer label prediction tasks. The tissue niches represent the cellular organization in the brain, grouping together neurons by major brain structure (pallium, subpallium, hypothalamus/extended amygdala, thalamus/epiphysis and midbrain/hindbrain), as well as major neurotransmitter type (glutamate and GABA)[8]. Non-neuronal cells are grouped into neuroglial, immune and vascular niches. The train–test split defined for this dataset is composed of a random image or tissue section hold-out across all sections in the measured entire male mouse brain (Extended Data Fig. 5a–c).

### CosMx human liver

We collected the CosMx human liver dataset from the publicly available CosMx data resource[38]. The dataset comprises cells from both a normal healthy liver measuring 332,877 cells across 301 fields of view covering one tissue section in a male 35-year-old patient, as well as cells from a hepatocellular carcinoma measuring 460,441 cells across 383 fields of view in one tissue section from a 65-year-old female patient. Both samples were measured with the 1000-plex CosMx Human Universal Cell Characterization Panel. The dataset includes both cell-type and niche labels. For the niche label prediction task, we used the healthy liver section, which provides six distinct labels defining structural zones in the liver: portal vein (zone 1a), zone 1b, zone 2a, zone 2b,

zone 3a and central vein (zone 3b; Extended Data Fig. 8b,d). We did not use the cancer liver sample for the niche label prediction task as it was primarily composed of cells annotated as a general tumor niche without further substructures provided. For the niche composition prediction task, we used both the cancer and healthy liver sections with the cell-type labels, which define 22 distinct cell types (antibody-secreting B cells, CD3[+] alpha beta T cells, central venous liver sinusoidal endothelial cells, cholangiocytes, erythroid cells, Hep, Hep 1, Hep 3, Hep 4, Hep 5, Hep 6, inflammatory macrophages, mature B cells, natural killer (NK)-like cells, non-inflammatory macrophages, periportal liver sinusoidal endothelial cells, portal endothelial cells, stellate cells, gamma delta T cells 1, tumor 1, tumor 2 and an undefined type (NotDet; Extended Data Fig. 8e). The train–test split defined for this dataset is composed of a random field of view hold-out across both tissue sections (Extended Data Fig. 8a,d).

### CosMx human lung

We collected the CosMx human lung dataset from the publicly available CosMx data resource[38]. This dataset contains samples from five different donors (301,611, 89,975, 227,110, 71,304 and 81,236 cells, respectively) across eight fields of view measured with the 1000-plex CosMx Human Universal Cell Characterization Panel. All donors have just one field of view, except for the first donor, which has three fields of view, and the third donor, which has two fields of view. The train–test split defined for this dataset is composed of a random field of view hold-out (Extended Data Fig. 9a,b). CosMx provides both cell-type and niche labels. We use the 22 distinct cell-type labels defined in this dataset for the niche composition prediction task. These labels are B cell, NK, T CD4 memory, T CD4 naive, T CD8 memory, T CD8 naive, regulatory T, endothelial, epithelial, fibroblast, myeloid dendritic cell, macrophage, mast, monocyte, neutrophil, plasmacytoid dendritic cell, plasmablast, tumor 12, tumor 13, tumor 5, tumor 6 and tumor 9 (Extended Data Fig. 9c).

### Xenium human lung

We collected the Xenium human lung dataset from the 10x Genomics data resource (https://www.10xgenomics.com/datasets/). This dataset measures two different lung sections, an adult human healthy lung (295,883 cells) and an adult human lung with invasive adenocarcinoma (531,165 cells). Both sections are measured with the 289-plex Xenium Human Lung Gene Expression Panel and an additional 100 lung cell-type-specific genes. As this dataset is not annotated, we only use it for the neighborhood density prediction task. We computed a spatial graph of cells with a radius of 25 μm² to calculate the cellular niche densities. The train–test split defined for this dataset is a random cell hold-out across all cells from both sections.

### Xenium human colon

We collected the Xenium human colon dataset from the 10x Genomics data resource (https://www.10xgenomics.com/datasets/). This dataset measures two different colon formalin-fixed paraffin-embedded-preserved tissue sections: a non-diseased colon (275,822 cells) and a cancer stage 2A adenocarcinoma (587,115 cells). Both sections are measured with the 325-plex Xenium Human Colon Gene Expression Panel and an additional 100 genes specifically selected to cover signaling and chemokine genes, and markers for stromal cells. As again this dataset is not annotated, we only use it for the neighborhood density prediction task. We computed a spatial graph of cells with a radius of 17 μm² in both sections to calculate the cellular niche densities. The train–test split defined for this dataset is a random cell hold-out across all cells from both sections.

### Dissociated dataset used for label transfer. *scRNA-seq of the primary motor cortex*.
Yao et al. generated a large-scale transcriptomic and epigenetic atlas of the mouse primary motor cortex[9]. We subsetted this large-scale dataset to cells measured with 10x v3 scRNA-seq. The subset captures 21,884 genes in 7,416 cells and annotates 19 different cell types (Astro, Endo, L5 ET, L5 IT, L6 CT, L6 IT, L6 IT Car3, L6b, L2/3 IT, L5/6 NP, Lamp5, microglia, OPC, oligo, Pvalb, Sncg, Sst, CLMC and Vip; Fig. 3c). We manually transferred cell types present in this dataset to the cell types measured in the MERFISH mouse brain dataset. We mapped Astro to Astro-Epen; Endo and VLMC to vascular; microglia to immune; oligo and OPC to oligo; L6 IT, L6 IT *Car3*, L5 IT, L2/3 IT, L5 ET to IT-ET Glut; L5/6 NP, L6b and L6 CT to NP-CT-L6b Glut; and Lamp5, Sncg, Vip Pvalb and Sst to CGE/MGE GABA, respectively.

### Nicheformer tokenization, architecture and pretraining
**Nicheformer tokenization.** The Nicheformer training corpus encompasses over 110 million cells in total, measured in more than 350 datasets using eight different sequencing technologies and two species: human and mouse. The total number of genes considered is 20,310, comprising 16,981 orthologous, 3,178 human-specific and 151 mouse-specific genes. For Nicheformer, we use a tokenization strategy similar to the one in Geneformer[22] with the difference that the cell transcripts are normalized according to the technology-specific nonzero mean to account for differences in the sequencing protocol. First, all cells are normalized so that each of them has 10,000 counts. To account for technological variations, we then compute a technology-specific gene expression nonzero mean vector, that is, the mean expression value of each gene, without considering the zero counts. We computed a single dissociated mean expression vector for the dissociated datasets because the differences between sequencing protocols in the dissociated cells are not as large as in the spatial assays. We then normalize the expression of each cell using the corresponding technology-specific mean expression vector to obtain the expression of each gene in each cell relative to the whole training corpus. Finally, the genes are ranked in descending order, from most to least expressed, excluding all non-expressed genes, creating an ordered set $T$ of genes as given by equation (1):

$$T = \left\{ \mathrm{idx}(gex_0), \mathrm{idx}(gex_1), \ldots, \mathrm{idx}(gex_n) : gex_{\mathrm{norm}_i} \geq gex_{\mathrm{norm}_{i+1}}; gex_{\mathrm{norm}_i} \neq 0 \right\} \quad (1)$$

where $\mathrm{idx}(gex_i)$ is a function that returns the index of gene $i$ in a previously defined vocabulary of genes and $gex_i$ is the gene expression of gene $i$ of a cell. To incorporate the influence of biological context on gene expression, we prepend contextual tokens for <ASSAY>, <MODALITY> and <ORGANISM> to the set $T$ to incorporate metadata information to the input data. These tokens encode metadata information, such as assay type (for example, MERFISH, CosMx and 10x 5′ v2), modality (dissociated or spatial) and organism (mouse or human). Recognizing the important impact biological context can have on gene expression, we augment the input sequences for our transformer model with modality, organism and assay tokens. This approach allows the model to explicitly learn representations that account for context-driven variations, leading to more robust and generalizable downstream analyses. Therefore, for a cell $i$, with a specific assay, organism and modality, the ordered set of tokens $T^i$ is shown in equation (2):

$$T^i = \left\{ \mathrm{assay}^i, \mathrm{organism}^i, \mathrm{modality}^i, \mathrm{idx}(gex_0^i), \mathrm{idx}(gex_1^i), \ldots, \mathrm{idx}(gex_n^i) \right\} \quad (2)$$

As a last step, the length of the set $T^i$ is truncated to $N = 1,500$. As not all cells have the same number of expressed genes, there might be sets whose total length is lower than 1,500. In those cases, <PAD> tokens are appended such that the final length is $N = 1,500$. <PAD> tokens ensure that all inputs have the same length by filling empty spaces with no semantic meaning. This is an important element when handling cells belonging to both RNA-seq and spatial assays because gene panels are usually smaller in the latter, which leads to a larger amount of <PAD> tokens in the set.

**Nicheformer architecture.** Given an initial input set $x^i \in R^{N \times D}$ composed of $N$ tokens of dimensionality $D$, Nicheformer encodes the position within the set by adding positional embeddings. Instead of modeling as sinusoidal embeddings, we use learnable embeddings for each position[80].

Nicheformer is composed of 12 stacked transformer blocks such that the output of one block is in the input of the following block. Given an input sequence $x^i \in R^{N \times D}$, according to equations (3) and (4):

$$x_0^i = x^i \tag{3}$$

$$x_{l+1}^i = \text{transformer\_block}_l(x_l^i) \quad \forall l \in [0, n-1] \tag{4}$$

Each transformer block consists of two main modules: a multihead self-attention mechanism and a feed-forward neural network. The multihead self-attention mechanism enables the model to weigh the relevance of different input elements in the input set when generating output representations. In our case, we use 16 attention heads, token dimensionality $D = 512$ and dimensionality of the hidden layer of the feed-forward network of 1,024. The <PAD> tokens are masked for the attention mechanism so that no token can pay attention to them.

**Nicheformer pretraining and performance optimization.** Nicheformer optimizes masked language modeling loss[80] during pretraining. We mask 15% of the tokens, including contextual and gene tokens but excluding <PAD> tokens, during pretraining. The model is then trained to predict the original tokens that have been masked, utilizing the unmasked tokens as context. Specifically, following the BERT schema[80], if the $i$-th token is chosen to be masked, 80% of the time it is replaced by a <MASK> token, 10% of the time by another random gene or contextual token and 10% of the time it remains unchanged. Mathematically, the masked language modeling loss is described as given by equation (5):

$$L_{\text{MLM}} = E_{x \sim X} E_M \sum_{i \in M} \left[ -\log p(x_i | x_{[1,n] \setminus M}) \right] \tag{5}$$

where $M$ is the set of masked tokens, $X$ is the entire dataset, $x$ is a cell of the dataset and $x_i$ is gene $i$ of the cell $x$.

Nicheformer was pretrained for approximately 10 days using three compute nodes, each with four Nvidia A100 40GB GPUs (total 12 GPUs). We train the model using bfloat16 mixed precision. We use the AdamW optimizer[81] with $\beta_1 = 0.9$ and $\beta_2 = 0.999$, weight decay of 0.1 and dropout of 0.0. The batch size is nine and the gradients are accumulated during ten batches before running the backward pass. The minimum learning rate is $1 \times 10^{-5}$, which increases until $1 \times 10^{-3}$ with a linear warmup of 100,000 steps. After the warmup, a cosine decay regime[82] is applied. Gradient clipping is set to 1.0 during the first epoch and then decreased to 0.5. All weights are initialized using Xavier initialization[83] with default parameters, while the bias terms are initialized to 0. Checkpoints were taken every 10,000 steps.

**Downstream tasks**

**Spatial cell-type, niche and region label prediction.** For the spatial cell-type, niche and region label classification task, we use the respective labels defined in the individual datasets (see 'Datasets used for downstream tasks and evaluations'). We extracted the unique labels for each class, transferred them to 64-bit signed integer values and one-hot encoded them as a matrix with $n$ different classes, with $n$ being the number of cell types, niches or regions. We then used for linear probing a linear layer optimized with a cross-entropy loss. We trained on the training set of the respective dataset for one epoch at a learning rate of $1 \times 10^{-3}$ and with a batch size of 256. The performance metrics reported are calculated on a held-out test set. We selected the model-assigned class label by calculating the argmax over the output

vector of the linear layer. Classification uncertainties reported in this work are the output of the linear layer rescaled to [0,1] such that the sum equals 1 using a Softmax function. We use no techniques to address class imbalances for two reasons. First, to evaluate the robustness of the representations learnt by Nicheformer. Secondly, it has been shown that using class imbalance techniques can even affect performance in cases such as cell-type classification[84].

**Neighborhood composition.** For the neighborhood composition regression tasks, we first define a spatial graph of cells by building an adjacency matrix based on the Euclidean distance in the two-dimensional coordinate space provided by the individual datasets. The adjacency matrix of spatial cells is a block-diagonal matrix $A \in R^{nxn}$, with $n$ equal to the number of cells present in the dataset calculated based on the spatial proximity of cells where connectivities can only occur within a field of view. We use a binary adjacency matrix with $a_{ij} = 1$ if $d(x_i, x_j) \leq \delta_r$ where $d(\cdot, \cdot)$ describes the Euclidean distance between nodes $i, j \in n$ and $\delta_r$ is the maximal distance between cells, and $a_{ij} = 0$ otherwise. We do not include self-connectivities for the adjacency matrix to not confound the signal. We additionally define the matrix of observed cell types $X_l \in \{0,1\}^{nxl}$ as a one-hot encoding of the $l$ distinct cell types present in the dataset. The neighborhood composition for a given radius is then given as equation (6):

$$N_r = \text{softmax}(A \times X_l) \in [0,1]^{nxl}. \tag{6}$$

The resulting matrix reflects for each cell captured in the dataset a vector giving the proportions of cell types present in the neighborhood of the cell. For the neighborhood prediction task, we used for linear probing a linear layer followed by a Softmax function to rescale the prediction to lie in the range [0,1] and sum to 1. We used the mean square error loss for optimizing this linear layer, trained on the training set of the respective dataset for one epoch at a learning rate of $1 \times 10^{-3}$ and with a batch size of 256. The performance metrics reported are calculated on a held-out test set.

**Neighborhood cell density prediction.** For the cellular niche density, we again use the adjacency matrix of spatial cells $A \in R^{nxn}$ calculated based on the Euclidean distance in the two-dimensional coordinate space. The cellular neighborhood density is then simply given by the row-wise sum of all connectivities in the adjacency matrix (equation (7)),

$$D_r = \sum_j (A_{ij}) \in N^{nx1} \tag{7}$$

for all cells present in the dataset with $r$ as a given radius, $i$ is the index cell for which we want to calculate the density, and $j$ is the total number of potential neighboring cells present in the dataset. For the density prediction task, we used for linear probing a linear layer with input being the respective embedding of a cell (Nicheformer, scVI or PCA) and output a scalar. We used the mean square error loss for optimizing this linear layer, trained on the training set of the respective dataset for one epoch at a learning rate of $1 \times 10^{-3}$ and with a batch size of 256. The performance metrics reported are calculated on a held-out test set.

**Nicheformer evaluation, linear probing and fine-tuning**

Nicheformer can be fine-tuned or used for linear probing. In both settings, we only train on the previously defined training set of the respective datasets used for downstream tasks (see 'Datasets used for downstream tasks and evaluation'). We use in both scenarios all Nicheformer gene tokens extracted from the last layer and average them to get a cell representation. Importantly, the contextual tokens are not used in the aggregation. While we observed no difference between using them and not using them in the downstream tasks focused on one modality, for example density prediction and niche

classification, we observed that transferring labels between spatial and dissociated datasets did not work at all when using the contextual tokens in the aggregation. Further investigation revealed that the output norm of the contextual token of modality was always the highest one, independently of the tissue (Extended Data Fig. 9d,e), hence playing a big role in the cell representation and biasing it toward the respective modality. This phenomenon has been reported in vision transformers[85], where some features that contain background information show higher norms as a consequence of the model using them to allocate internal computations. Literature[85] proposes the use of registers that are discarded in the computation of the final representation. While excluding contextual tokens mitigates modality bias, it may also discard useful information; future work could explore selective integration strategies to retain relevant context.

In linear probing, the previously computed parameter weights of the Nicheformer pretraining model are frozen, that is, not updated further, and are subsequently used as input to a downstream task. The cell's representation is then fed into a linear layer specific to each downstream task, which represents either a classification task in the case of the niche and region label prediction or a regression for predicting the neighborhood composition and cellular density. For the neighborhood composition task, we additionally fitted an MLP that uses the Nicheformer embedding as input and predicts the varying neighborhood composition vectors in a dataset. The MLP is optimized using the average mean squared error across all neighborhood sizes considered. Fine-tuning generally describes using a pretrained model, and training it to a specific downstream task of choice. We speak of a fine-tuned Nicheformer version when we allow the model to change the previously learned parameter space and the weights are updated for a specific task. Importantly, each downstream task can also be optimized with respect to a new set of metrics. All runs are trained for a single epoch with a maximum learning rate of $1 \times 10^{-4}$ and a cosine decay scheduler reaching $1 \times 10^{-5}$ at the end. The batch size is nine with gradients accumulated for ten batches (Supplementary Table 5). We highlight the respective tasks and metrics used to compute them in 'Downstream tasks'.

## Nicheformer cell embedding stability analysis

We evaluated the robustness of Nicheformer's gene-rank-based cell embeddings to perturbations that mimic real-world scenarios such as incomplete gene panels or measurement noise, common in spatial transcriptomics. As the model operates on sequences of gene tokens ordered by expression rank, we assessed how alterations to this sequence affect embedding stability.

We selected one dissociated brain dataset and one spatial brain dataset from SpatialCorpus-110M, tokenized the cells, and applied controlled perturbations before passing them through the pretrained Nicheformer model. Perturbations included (i) randomly shuffling 10%, 20%, 50% or 100% of the gene rankings in each cell's token sequence (Extended Data Fig. 1a) and (ii) randomly dropping 10%, 20%, 50% or 80% of the genes from the sequence (Extended Data Fig. 1b). We then embedded the perturbed cells and evaluated the similarity between perturbed and original embeddings using integration metrics from scIB[18].

To quantify embedding stability, we used the silhouette score, leveraging cell-type annotations to define ground-truth clusters. We observed that Nicheformer embeddings remained stable up to a 20% perturbation in both rank shuffling and gene dropout scenarios, indicating robustness to input noise and incomplete gene measurements (Extended Data Fig. 1). These results support the suitability of rank-based encoding for learning generalizable cell representations under varying input conditions.

## Nicheformer modalities and organisms split performance analysis

To analyze the need to train a model on a diverse train dataset, we conducted controlled experiments in which we pretrained Nicheformer

models and tested them in different downstream tasks and tissues. Specifically, we pretrained Nicheformer models of 49.3 million parameters using the same compute budget—3 days in an entire node containing four A100 GPUs. Due to the large compute needed to retrain Nicheformer models using the entire SpatialCorpus-110M, we subset it for the experiments, so each model is pretraining in 1% of that dataset (~1.1 million cells).

In particular, we pretrained models in the following data splits: 1.1 million randomly sampled spatial cells, 1.1 million randomly sampled dissociated cells and 3.3 million randomly sampled dissociated cells (to assess whether a large number of dissociated cells can account for the lack of spatial information). Additionally, we also pretrained a model in 1.1 million dissociated cells sampled in such a way that there is the same number of cells from blood, colon, intestine, lung, liver and brain, to assess the effect of the tissue variability of the dataset. To assess the importance of multispecies datasets, we also pretrained models on 1.1 million spatial cells sampled only from humans and 1.1 million spatial cells sampled only from mice.

We evaluated the pretrained models on the following downstream tasks: niche prediction in the human liver and lung CosMX datasets, and cell-type classification and niche regression in the mouse brain MERFISH dataset. In all cases, the models were evaluated in the linear-probing scenario running three seeds. All results were statistically assessed using analysis of variance, with $P$ values adjusted for multiple comparisons using the Benjamini–Hochberg procedure (FDR).

## Nicheformer attention analysis

We conducted an attention analysis to explore the attention patterns in Nicheformer and how it differentiates between male and female cells by focusing on sex-specific gene variations. We sample 2,000 CD8 and 2,000 CD4 cells from the lung; 2,000 healthy and 2,000 cancer cells from the liver; 2,000 male and 2,000 female cells from the MERFISH mouse brain datasets and 2,000 random cells from the primary motor cortex scRNA-seq dataset to ensure sufficient diversity. In all cases, except in the MERFISH mouse brain dataset, we study the attention paid to the top 50 most expressed genes on average. For the MERFISH mouse brain cells, we use two gene sets: a prior-knowledge set of SDGs, known for exhibiting sex differences, and a randomly sampled control set of 97 genes. We feed all cells into the model and extract attention matrices from all 16 attention heads across the 12 transformer blocks. Then, to assess general trends in attention distribution, we average the attention scores to obtain an attention score per layer. In addition to this, we extract the maximum attention value for each gene per layer, isolating the highest level of focus from any single attention head. Evaluating both average and maximum attention, allows us to discern whether certain genes consistently receive attention across multiple heads or are sharply focused on by individual heads. Specifically, we compare the attention scores according to equation (8):

$$A_{ij} = \mathrm{softmax}\left(\frac{Q_i K_j^T}{\sqrt{d}}\right) \qquad (8)$$

where $A_{ij}$ represents the attention that token $i$ pays to token $j$. As we have 16 attention layers, we denote $A_{ij}^h$ the attention that token $i$ pays to token $j$ in the layer $h$.

In Nicheformer, with 12 layers, the attention matrices for each layer and head are represented as $A_{ij}^{(l,h)}$, where $l \in \{1, 2, \dots, 12\}$ represents the layer, and $h \in \{1, 2, \dots, 16\}$ denotes the head. To assess how much attention each token pays to a token $m$, we focus on extracting the attention scores $A_{im}^{(l,h)}$, which capture the attention that each token $i$ allocates to the $m$ in layer $l$ and head $h$.

For each observation, we compute both the maximum and average attention that any token $i$ pays to the token $m$ across all heads in each layer. This is done by first calculating the maximum and average attention for each layer as given by equations (9) and (10):

$$\max \text{Attention}_l = \max_{i,t} A_{i,m}^{(l,h)} \tag{9}$$

$$\text{averageAttention}_l = \frac{1}{I}\frac{1}{H}\sum_{h=1}^{H}\sum_{i=1}^{I} A_{i,m}^{(l,h)} \tag{10}$$

where $i$ refers to all other tokens in the sequence and $H$ is the number of heads (16). These values give us the highest attention score and the average attention score that the token $m$ receives from other tokens for each layer, respectively, considering all heads. By averaging these maximum and average attention values across multiple observations, we can assess how attention is distributed across layers, identifying the layers where the token $m$ receives the most focus and how consistently it receives attention across tokens and heads.

### Ortholog genes analysis

We conducted an attention analysis to study deeper the role of ortholog genes in Nicheformer and assess whether there were major differences between using or not using them and how they are related. To do so, we trained small Nicheformer models in a reduced gene space with and without using orthologs. Specifically, we used a gene vocabulary of 9,026 genes, which when mapping orthologs is reduced to 7,407 (Extended Data Fig. 9f). We compared the performance of both models with three different downstream tasks: niche prediction in the CosMX human lung and liver dataset and niche regression in the MERFISH mouse brain dataset. We found that there were differences in the performance in the latter only (Extended Data Fig. 9g).

Likewise, we studied, for the model without the ortholog mapping, whether genes with known cross-organism equivalents are more similar to their ortholog equivalent than to any other random gene. To analyze that, we extracted the gene embeddings after the pretraining and analyzed their cosine similarity. The results indicated that genes are less similar to their ortholog than to random genes, which can be explained by the fact that they are never seen together in any cell and that they might have different functions (Extended Data Fig. 9h).

### Benchmarking against competing methods

**Comparisons against Geneformer, scGPT, UCE and CellPLM.** To get the Geneformer embeddings, we used the release v.0.0.1 of the official Geneformer repository on Hugging Face and extracted the embeddings using the pretrained weights of the larger 12-layer variant provided at the time. We used the second to last layers to get a more general representation as recommended by the repository. We also used mean pooling as the only available option provided to aggregate the output gene embeddings into a single-cell embedding.

For the comparison against scGPT, we first created scGPT embeddings using scGPT 0.2.1, pretrained on the whole human as recommended in the original publication. The embeddings were generated for three datasets, the MERFISH mouse brain, the CosMx human lung and the CosMx human liver. For the MERFISH mouse dataset, we first mapped the mouse genes to human genes using BioMart[43] through the official Ensembl releases[44]. The fraction of overlapping genes compared to the gene context used in scGPT was for the MERFISH mouse brain dataset of 471/483 genes, for the CosMx human liver dataset of 997/999 genes and for the CosMx human lung dataset of 958/960 genes.

To get UCE embeddings, we used the latest version from the original repository and followed the tutorials to obtain the cell embeddings. The fraction of overlapping genes compared to the gene context used in scGPT was for the MERFISH mouse brain dataset of 472/483 genes, for the CosMx human liver dataset of 990/999 genes and for the CosMx human lung dataset of 954/960 genes.

For the comparison against CellPLM, we used the latest official version of the repository. For the MERFISH mouse dataset, we first mapped the mouse genes to human genes using BioMart[43] through the official

Ensembl releases[44]. The fraction of overlapping genes compared to the gene context used in scGPT was for the MERFISH mouse brain dataset of 473/483 genes, for the CosMx human liver dataset of 997/999 genes and for the CosMx human lung dataset of 958/960 genes. The cell embeddings were obtained by following the notebook tutorials.

The resulting Geneformer, scGPT, UCE and CellPLM embeddings then served as input to a linear layer specific to each downstream task (Supplementary Table 5).

**Baseline comparisons to scVI and PCA embeddings.** We compared the performance of the fine-tuned Nicheformer model and the linear-probing scenario to embeddings generated with scVI[17] and PCA. We generated scVI and PCA embeddings on just the downstream datasets themselves and additionally on an informed 1% subset of all datasets present in the SpatialCorpus-110M. We used this subset to train two different scVI models as specified in Supplementary Table 5 to generate latent representations with 512 and 10 dimensions, respectively. The two models were then used to obtain latent representations for the datasets that were used for downstream task evaluations. The PCA embeddings were generated in a similar way using the implementation available in sklearn v.1.4.1 to obtain PCA embeddings of dimensions 512 and 10, respectively.

We split the fine-tuning datasets (MERFISH mouse brain, CosMx human liver, CosMx human lung, Xenium human lung, Xenium human colon) into a training and test set, using the same random splits as applied for the Nicheformer fine-tuning. scVI and PCA were computed on each fine-tuning dataset individually. We used scvi-tools v.1.1.2 with a negative binomial distribution gene likelihood on the raw gene expression counts and trained scVI on the training set with a batch size of 256 for 10 epochs and used two hidden layers for the encoder and decoder neural networks. The resulting embedding was chosen to have a latent dimension of 256. After training, we returned the latent representation for each cell in both the training set and the test set.

For generating PCA embeddings for each dataset, we used the implementation available in sklearn v.1.4.1. We first normalized the respective raw gene expression counts for each dataset so that each cell has a total number of counts equal to the median of the total counts for all cells with scanpy v.1.10.1. Next, we used scanpy to log1p-transform the data matrix to ensure the data are centered before using it as input to the PCA implementation. We used the sklearn implementation and evaluate the cumulative explained variance ratio in the training dataset (Extended Data Fig. 10). Finally, we evaluated the model for a diverse set of principal components to have a fair comparison (Extended Data Fig. 7). All other parameters are the defaults provided by the sklearn implementation. We fit the PCA on the training set and afterwards applied the dimensionality reduction to both the training set and test set. The resulting lower-dimensional representations, X_scvi and X_pca, then serve as input to a linear layer specific to each downstream task (Supplementary Table 5).

### Reporting summary

Further information on research design is available in the Nature Portfolio Reporting Summary linked to this article.

## Data availability

The Allen brain atlas consortium generated the Allen Institute brain atlas mouse p20, Allen Institute brain atlas mouse p28 and Allen Institute brain atlas mouse female datasets (Supplementary Table 4), which were kindly provided to us before publication. As these spatial datasets are currently unpublished, they are not yet publicly available. We will make them accessible to readers upon their official release by the Allen Institute. All other datasets used in this study are publicly available. The single-cell RNA-seq data can be accessed through the Gene Expression Omnibus (GEO) under the following accession numbers: GSE117824 (ref. 86), GSE118068 (ref. 87), GSE119940 (ref. 88),

GSE124952 (ref. 89), GSE126060 (ref. 90), GSE128423 (ref. 91), GSE128761 (ref. 92), GSE128987 (ref. 93), GSE129826 (ref. 94), GSE130593 (ref. 95), GSE130822 (ref. 96), GSE130879 (ref. 97), GSE130888 (ref. 98), GSE131339 (ref. 99), GSE131996 (ref. 100), GSE132355 (ref. 101), GSE133531 (ref. 102), GSE134571 (ref. 103), GSE135310 (ref. 104), GSE135326 (ref. 105), GSE135356 (ref. 106), GSE135414 (ref. 107), GSE136394 (ref. 108), GSE136441 (ref. 109), GSE137026 (ref. 110), GSE139168 (ref. 111), GSE140510 (ref. 112), GSE140628 (ref. 113), GSE141471 (ref. 114), GSE141526 (ref. 115), GSE141552 (ref. 116), GSE141784 (ref. 117), GSE142143 (ref. 118), GSE142797 (ref. 119), GSE143293 (ref. 120), GSE145216 (ref. 121), GSE145251 (ref. 122), GSE145326 (ref. 123), GSE145689 (ref. 124), GSE145866 (ref. 125), GSE146122 (ref. 126), GSE146138 (ref. 127), GSE146194 (ref. 128), GSE146298 (ref. 129), GSE146512 (ref. 130), GSE148339 (ref. 131), GSE148978 (ref. 132), GSE149040 (ref. 133), GSE149201 (ref. 134), GSE149356 (ref. 135), GSE149931 (ref. 136), GSE150708 (ref. 137), GSE150871 (ref. 138), GSE150995 (ref. 139), GSE151186 (ref. 140), GSE152325 (ref. 141), GSE152573 (ref. 142), GSE152988 (ref. 143), GSE152999 (ref. 144), GSE153099 (ref. 145), GSE153117 (ref. 146), GSE153274 (ref. 147), GSE153288 (ref. 148), GSE153762 (ref. 149), GSE153770 (ref. 150), GSE153802, GSE154196 (ref. 151), GSE154359 (ref. 152), GSE154386 (ref. 153), GSE154567 (ref. 154), GSE154579 (ref. 155), GSE154932 (ref. 156), GSE155226 (ref. 157), GSE155340 (ref. 158), GSE155788 (ref. 159), GSE155850 (ref. 160), GSE156136 (ref. 161), GSE156183 (ref. 162), GSE156245 (ref. 163), GSE156285 (ref. 164), GSE156920 (ref. 165), GSE157244 (ref. 166), GSE157292 (ref. 167), GSE157362 (ref. 168), GSE157525 (ref. 169), GSE157771 (ref. 170), GSE157773, GSE157977 (ref. 171), GSE158038 (ref. 172), GSE158192 (ref. 173), GSE158356_mouse (ref. 174), GSE158450 (ref. 175), GSE159354 (ref. 176), GSE159519 (ref. 177), GSE159977 (ref. 178), GSE160061 (ref. 179), GSE160097 (ref. 180), GSE160098 (ref. 181), GSE160664 (ref. 182), GSE160729 (ref. 183), GSE160772 (ref. 184), GSE161066 (ref. 185), GSE161227 (ref. 186), GSE161230, GSE161363 (ref. 187), GSE161685 (ref. 188), GSE161937 (ref. 189), GSE162073 (ref. 190), GSE162807 (ref. 191), GSE163018 (ref. 10), GSE163278 (ref. 192), GSE163650 (ref. 193), GSE163668 (ref. 194), GSE163701 (ref. 195), GSE163830, GSE163919, GSE164044 (ref. 196), GSE164573 (ref. 197), GSE165551 (ref. 198), GSE165554 (ref. 198), GSE166218 (ref. 199), GSE166262 (ref. 200), GSE166525 (ref. 201), GSE166797 (ref. 202), GSE166992 (ref. 203), GSE167595 (ref. 204), GSE167992 (ref. 205), GSE168732 (ref. 206), GSE168758 (ref. 207), GSE169718 (ref. 208), GSE172127 (ref. 10), GSE200218 (ref. 209), GSE225278 (ref. 210), GSE114687 (ref. 211), GSE117176 (ref. 212), GSE117770 (ref. 213), GSE120508 (ref. 214), GSE122342 (ref. 215), GSE122960 (ref. 216), GSE123722 (ref. 217), GSE124691 (ref. 218), GSE128855 (ref. 219), GSE129519 (ref. 220), GSE130238 (ref. 221), GSE131685 (ref. 222), GSE132672 (ref. 223), GSE135893 (ref. 224), GSE136001 (ref. 225) and GSE136103 (ref. 226). All datasets are available for download at https://huggingface.co/datasets/theislab/SpatialCorpus-110M. More information about the dissociated data collection and spatial data collection of the SpatialCorpus-110M can be found in Supplementary Tables 3 and 4, respectively. Source data are provided with this paper. Source data are provided with this paper.

## Code availability

All models described here are implemented in a Python package available at https://github.com/theislab/nicheformer/. It contains tutorial notebooks on how to use the model for downstream tasks, including learning probing and fine-tuning scenarios. It also includes a tutorial on continuing the pretraining in new datasets. Downloading and preprocessing scripts for all public datasets used in pretraining and fine-tuning the models are available at the 'data' directory of https://github.com/theislab/nicheformer. Additionally, all public datasets can be downloaded directly from HuggingFace at https://huggingface.co/datasets/theislab/SpatialCorpus-110M.

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

## Acknowledgements

We thank L. Zappia, L. Hetzel, A. Palma, S. Jimenéz, F. Fischer, J. Engelmann, A. Szałata, L. Heumos and M. Kuijs for valuable discussions and feedback on this project. We thank lamin.ai, specifically A. Wolf, L. Heumos, S. Sun and S. Rybakov for helpful discussions on data curation, data management and model training. Additionally, we thank H. Zeng, M. Kunst and the Allen Brain Atlas consortium for providing us early access to their MERFISH whole mouse brain atlas and the additional unpublished MERFISH mouse brain datasets. Additionally, we thank M. Nilsson and S. M. Salas for providing us early access to their unpublished Xenium and ISS datasets. This work was co-funded by the European Union (ERC, DeepCell - 101054957) and supported by the Chan Zuckerberg Initiative Foundation (CZIF; grant CZIF2022-007488 (Human Cell Atlas Data Ecosystem)), by the Wellcome Leap ΔTissue Program and through the BRAIN Initiative Cell Atlas Network (BICAN). G.P. and L.D. acknowledge funding by the Joachim Herz Foundation. L.D. was additionally supported by the BMBF-funded de.NBI Cloud within the German Network for Bioinformatics Infrastructure (de.NBI) (031A532B, 031A533A, 031A533B, 031A534A, 031A535A, 031A537A, 031A537B, 031A537C, 031A537D, 031A538A).

## Author contributions

F.J.T. conceived the study with the help of A.C.S., and A.T.-L.; A.C.S. and A.T.-L. contributed equally and have the right to list their name first in their curriculum vitae; F.J.T. supervised the project; A.T.-L. and A.C.S. performed the analysis and wrote the code; A.T.-L. led the data engineering, model design, implementation and pretraining in discussion with G.P.; L.H. and A.C.S. led the data curation effects for the dissociated data collection; A.C.S. led the data curation efforts for the spatial data collection; M.M. and L.D. supported the data curation efforts for the dissociated data collection; F.D. and R.G. supported the data curation efforts for the spatial data collection; M.B. and T.R. helped with the benchmarking; R.G. and L.V. helped to interpret the brain and liver results; A.C.S. designed and created all main figures; A.C.S., A.T.-L., G.P., R.G. and F.J.T. wrote the manuscript. All authors read, corrected and approved the manuscript.

## Funding

## Competing interests

F.J.T. consults for Immunai, CytoReason, Cellarity, BioTuring and Genbio.AI and Valinor Industries and has an ownership interest in Dermagnostix GmbH and Cellarity. As of September 2024, A.C.S. is an employee of Bioptimus. As of September 2024, G.P. is an employee of the Chan Zuckerberg Initiative. All results and analysis presented in this work were conducted before the two respective employment statuses. The other authors declare no competing interests.

## Additional information

**Extended data** is available for this paper at https://doi.org/10.1038/s41592-025-02814-z.

**Correspondence and requests for materials** should be addressed to Fabian J. Theis.

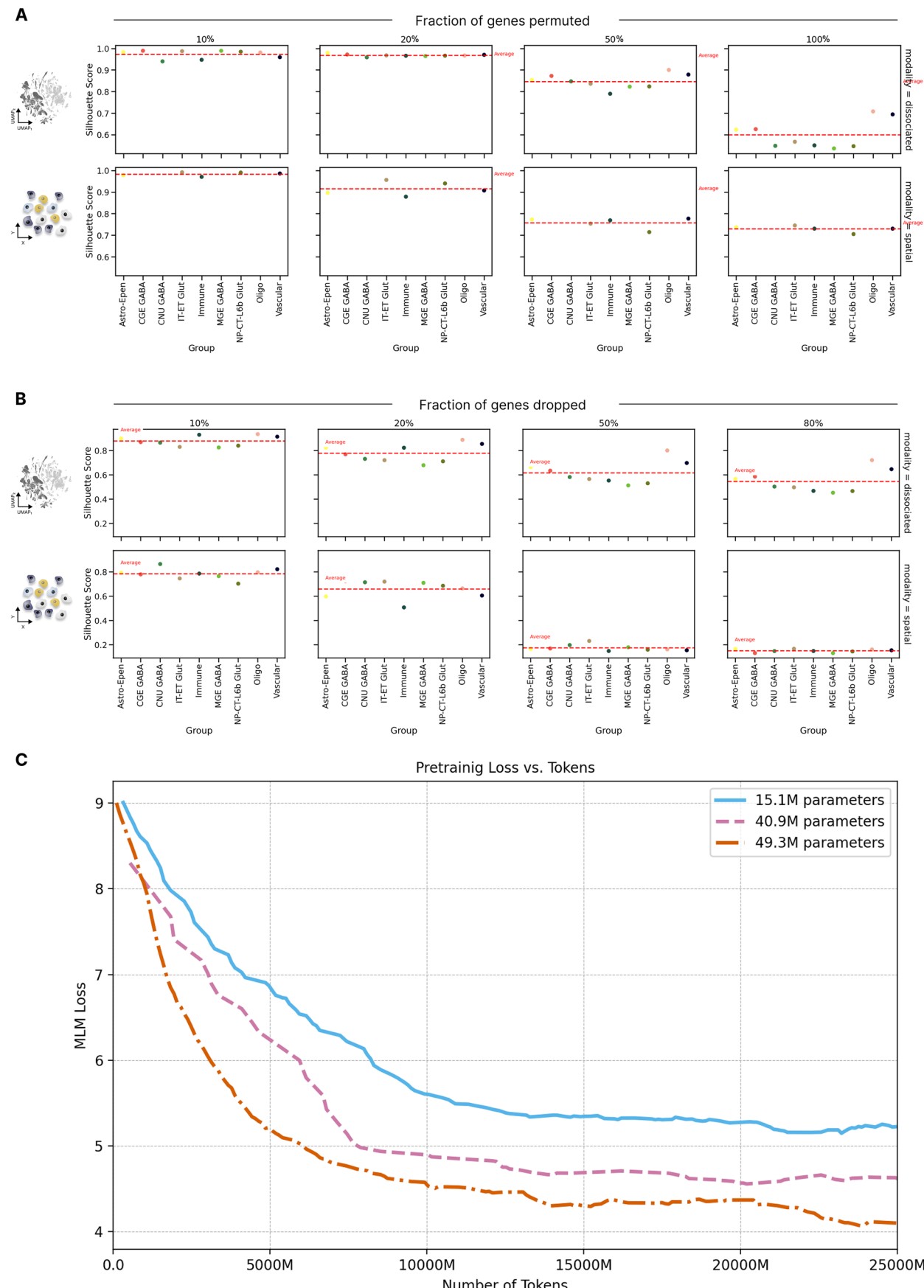

**Extended Data Fig. 1 | See next page for caption.**

**Extended Data Fig. 1 | Nicheformer's cell representations are robust to input noise and MLM loss as a function of the total number of tokens seen by the model. A)** We compute Nicheformer cell representations for a dissociated and spatial brain dataset and use author cell type annotations as ground truth. We randomly permute 10%, 20%, 50% and 100% of the genes in the input sequence and obtain cell representations. Then, we compute the silhouette score to evaluate how perturbed are the cell representations. **B)** We repeat the same experiment but instead of permuting genes, we drop them off the input sequence (which contains only non-zero genes). In particular, we drop 10%, 20%, 50% and 80% of the genes in the input sequence. In this case, the deterioration of the cell embeddings happens faster than when permuting genes. Cell representations of spatial cells deteriorate faster than dissociated cells (<0.2 silhouette score against >0.6 silhouette score for 50% dropout level). We hypothesise that this happens due to the shorter gene panels, that is in large gene panels, Nicheformer can leverage more information from the longer context length to correct disturbances in the data. **C)** Shown are the loss curves of three different models with varying parameter size, 15.1 million parameters, 40.9 million parameters and 49.3 million parameters, respectively. The larger the model, the lower is the pretraining loss. All the losses are a moving average with a window of 10. All the models were evaluated in the same training set with fixed random seed.

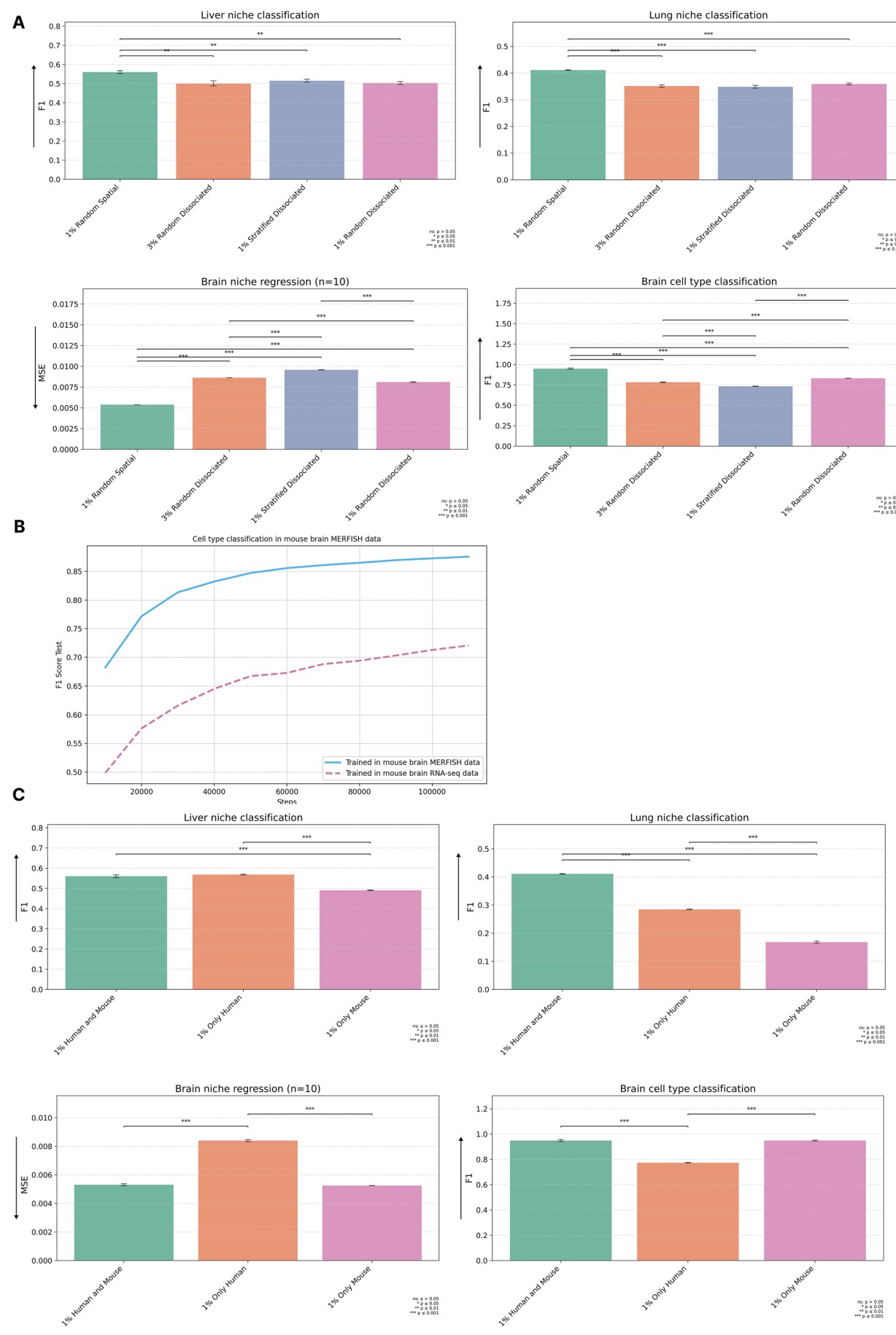

**Extended Data Fig. 2 | See next page for caption.**

**Extended Data Fig. 2 | Downstream performance across different tissues of Nicheformer models trained on different subsets of the data splitting by modality. A)** Shown are the F1 scores for niche classification in the CosMx human liver (top left) and lung (top right) datasets, cell type classification in MERFISH mouse brain (bottom right) and the MSE for niche regression in MERFISH mouse brain (bottom left) obtained by different models trained on different data subsets. The results demonstrate a clear advantage of training on spatial data compared to dissociated data. A model trained on just 1% of spatial data significantly outperforms models trained on the same or even three times the amount of dissociated data, reinforcing the fundamental difference between these modalities. This suggests that no amount of dissociated data can fully compensate for the spatial context when evaluated on spatial tasks. Additionally, computational efficiency plays a crucial role: the model trained on a smaller dissociated subset (1%) performs better than one trained on a larger subset (3%) because both were trained for the same duration, leading to more updates per sample in the smaller dataset. Furthermore, stratified training offers advantages only in specific cases, such as the liver, which can be explained by the distribution of tissue types in the random subset - since they are overly present in SpatialCorpus-110M. For example, brain cells are more abundant in the random

subset than in the stratified one, potentially influencing performance. The results are found statistically significant even after adjusting for FDR. **B)** Shown are the F1 score curves of two different models trained on different modalities: spatial and dissociated respectively. Both models have the same number of parameters and have been training for the same amount of time. The task is performed by linear probing. The model trained on MERFISH data notably outperforms the model trained on RNA-seq, highlighting a significant distribution shift between technologies. **C)** Shown are the F1 scores for niche classification in the CosMx human liver (top left) and lung (top right) datasets, cell type classification in MERFISH mouse brain (bottom right) and the MSE for niche regression in MERFISH mouse brain (bottom right) obtained by different models trained on different data subsets. As in the previous data split test, a broad coverage train distribution is necessary to achieve good performance across a variety of scenarios. In this case, models trained uniquely in mouse data underperform in downstream tasks based on human data (top row); and models trained on only human data underperform in downstream tasks based on mouse data (bottom row). A model trained on a combination of mouse and human data performs on pair in both cases. Results were found statistically significant even after FDR correction.

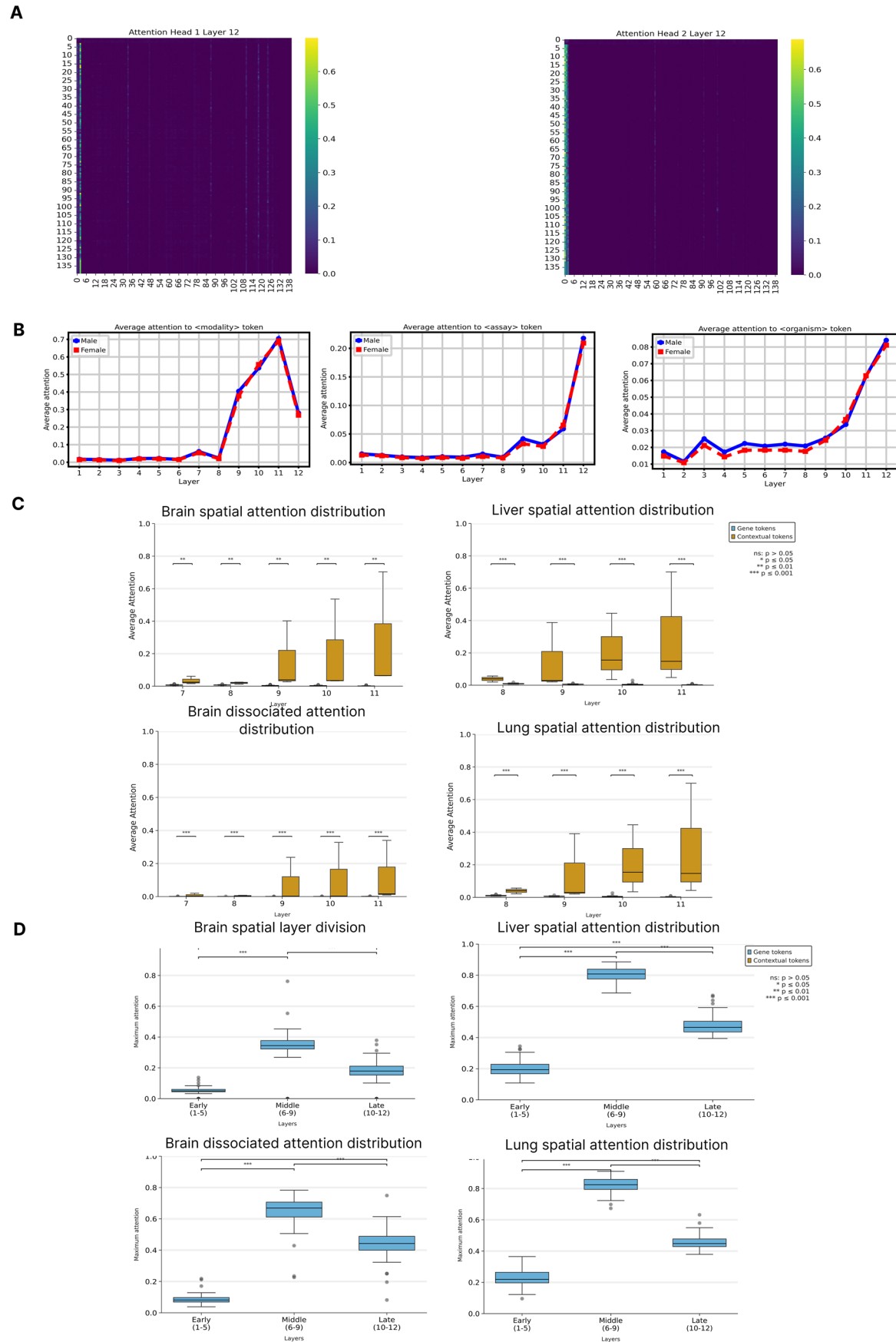

**Extended Data Fig. 3 | See next page for caption.**

**Extended Data Fig. 3 | Analysis of Nicheformer attention to contextual and gene tokens. A)** Shown are different attention matrices extracted from the last transformer block of Nicheformer. They present a similar pattern in which almost all attention is paid to the metadata tokens. **B)** Average attention paid, per layer, to the metadata tokens. It can be observed a clear trend: the last layers of the model pay, by a large margin, the most attention to the metadata tokens. The analysis is done in both male and female brain mouse datasets to showcase that the pattern is consistent. **C)** Shown are box plots representing the distribution of attention paid to contextual tokens (orange) and gene tokens (blue) in the latest Nicheformer's layers. The p-values are the result of performing Mann-Whitney

U tests to assess whether there is a significant difference between the distribution of attention paid to contextual and gene tokens. To control the false discovery rate (FDR), we applied the Benjamini-Hochberg procedure to adjust the p-values. **D)** Shown are box plots representing the distribution of attention paid to gene tokens in 3 groups of layers: early (from layer 1 to layer 5), middle (layer 6 to layer 9) and late (from layer 10 to layer 12). The p-values are the result of performing Mann-Whitney U tests to assess whether there is a significant difference between the distribution of attention paid to contextual and gene tokens. To control the false discovery rate (FDR), we applied the Benjamini-Hochberg procedure to adjust the p-values.

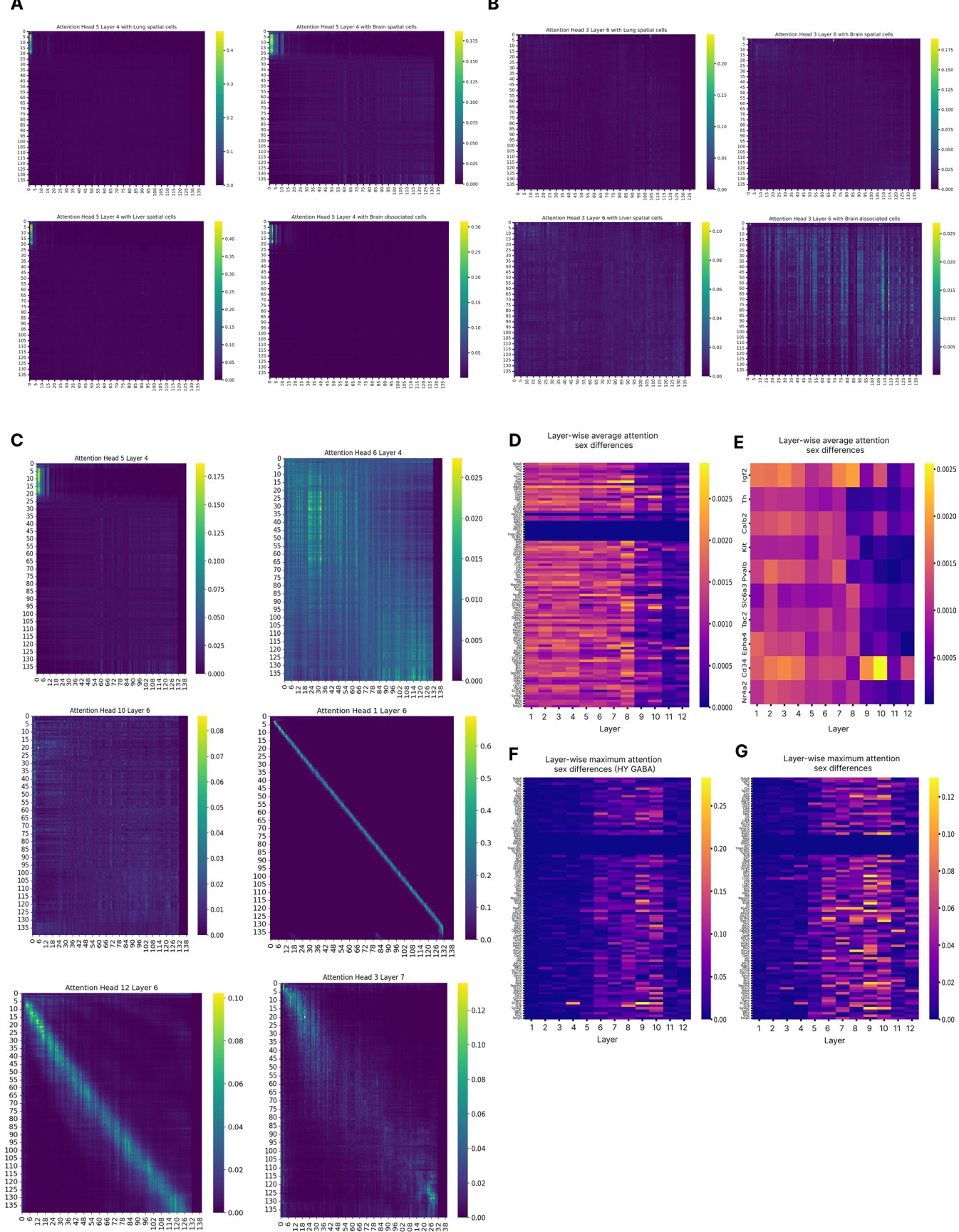

**Extended Data Fig. 4 | See next page for caption.**

**Extended Data Fig. 4 | Analysis of Nicheformer attention heads and layer-wise attention gender difference.** Shown are the attention matrices obtained from the head 5 of the Nicheformer layer 4 when processing lung spatial cells (top left), brain spatial cells (top right), liver spatial cells (bottom left) and brain dissociated cells (bottom right). It can be seen that this attention head uniquely focuses on the most expressed genes, independently of the tissue or modality of the cell. **B)** Shown are the attention matrices obtained from the head 3 of the Nicheformer layer 6 when processing lung spatial cells (top left), brain spatial cells (top right), liver spatial cells (bottom left) and brain dissociated cells (bottom right). It can be seen that the attention pattern of this attention head changes when processing dissociated cells or spatial cells. **C)** Shown are different attention matrices obtained when feeding Nicheformer with cells from the AVPV section. Different heads showcase different patterns, which reveal diverse attention behaviours, including metadata token focus (Head 5, Layer 4), selective gene interactions (Head 6, Layer 4), diffuse attention across genes (Head 10, Layer 6), strong self-attention (Head 1, Layer 6), combined self and global attention (Head 12, Layer 6), and concentrated attention on key genes (Head 3, Layer 7). **D)** The first layers of Nicheformer show the highest attention differences between cell and female cells, even though this is very small. **E)** The same pattern holds for the SDN genes. **F)** Nicheformer's middle layers show the maximum attention score differences between the male and the female cells for the HY GABA cells within the AVPV section. **G)** The same pattern occurs when examining the maximum differences for all cells in the AVPV section. The contrast of the average attention difference plotted here and the maximum attention differences (Fig. 3d-f) suggests that the sex differences are captured by a subset of the attention heads. The average attention difference is computed averaging all attention heads, whereas the maximum attention difference attends to the maximum difference reported in any head.

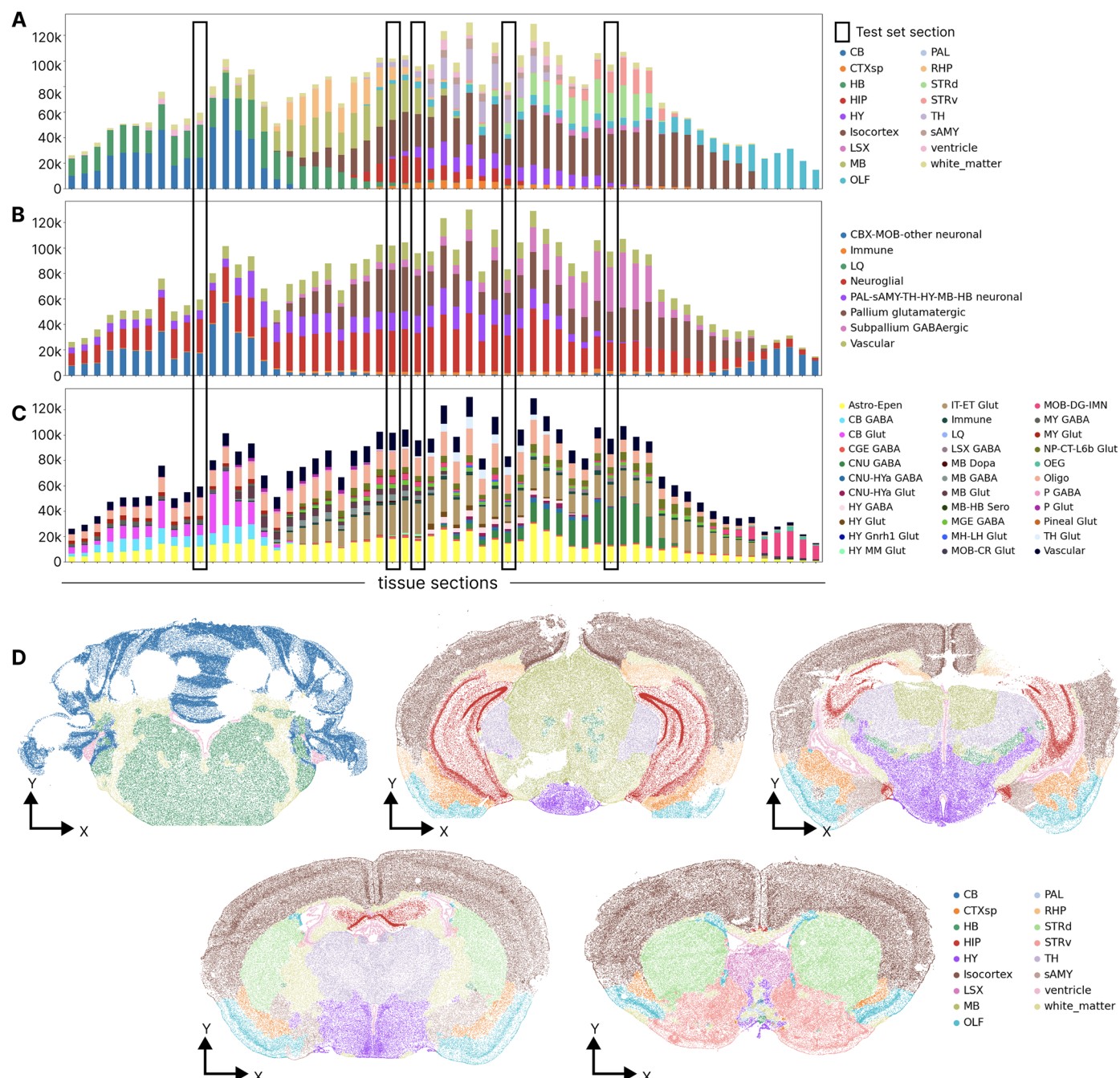

**Extended Data Fig. 5 | Nicheformer fine-tuning datasets - MERFISH mouse brain. A-C)** Region (**A**), niche (**B**), and cell type (**C**) label distribution across all tissue sections in the MERFISH mouse brain data with the test set highlighted. **D)** Spatial allocation of cells in the five test tissue sections of the MERFISH mouse brain **E)** UMAP visualization of the Nicheformer embedding of the MERFISH mouse brain dataset colored by region label. **F)** Exemplary brain slice of the MERFISH mouse brain dataset colored by region label.

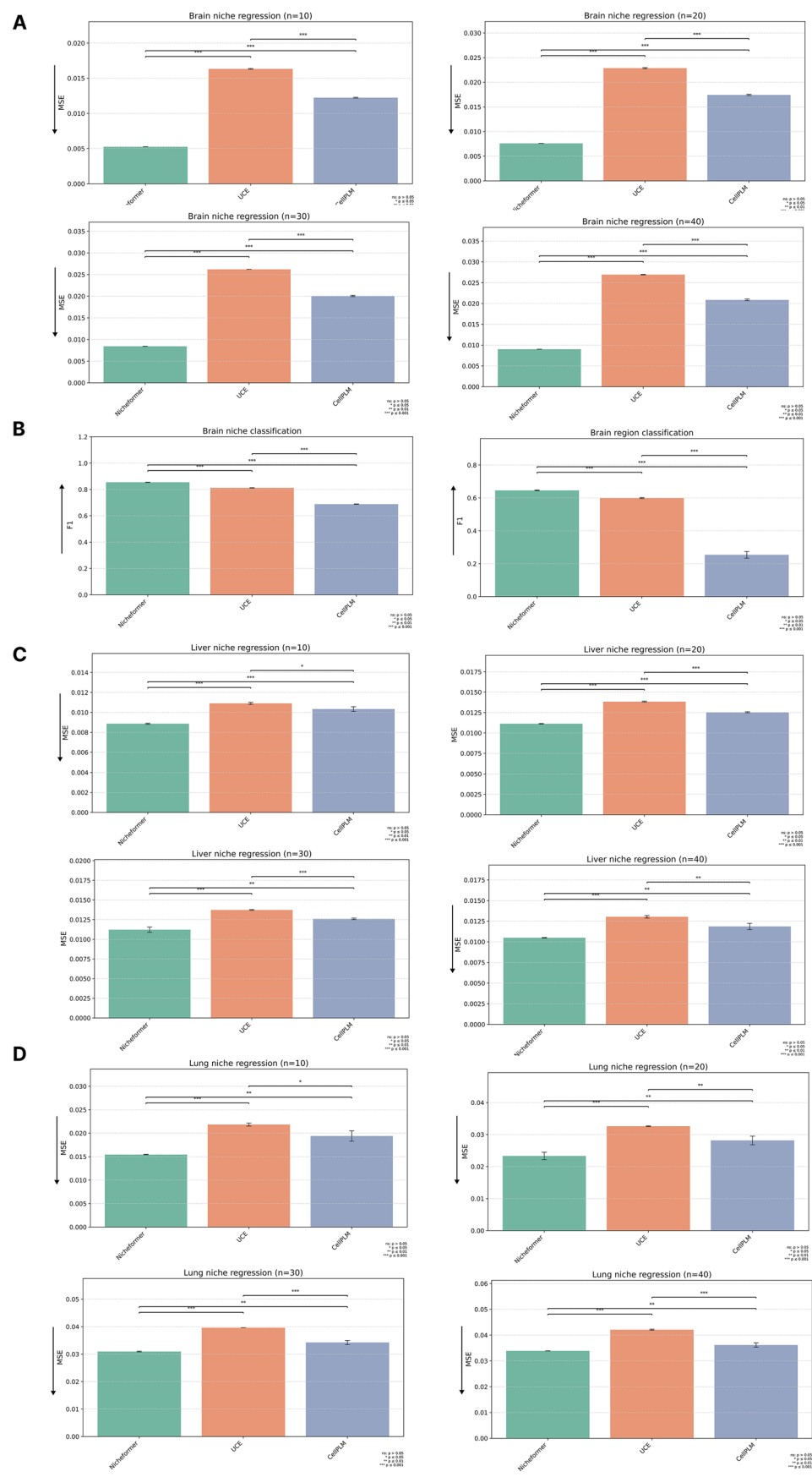

**Extended Data Fig. 6 | See next page for caption.**

**Extended Data Fig. 6 | Comparison between Nicheformer, UCE and CellPLM in the MERFISH mouse brain, CosMX human liver and CosMX human lung datasets. A)** Downstream task metrics (MSE) for models trained in the MERFISH mouse brain dataset using linear probing on Nicheformer, UCE and CellPLM embeddings. The downstream tasks evaluated are niche regression for 4 different radius sizes. In all cases, Nicheformer outperforms both CellPLM and UCE, being the differences statistically significant. **B)** F1 Score for region and niche prediction in the MERFISH mouse brain dataset. Likewise, Nicheformer outperforms CellPLM and UCE and the differences are statistically significant. The arrows indicate which direction is the optimal one. For F1 Score, the higher the better; for MSE, the lower the better. **C)** Downstream task metrics (MSE) for models trained in the CosMX human liver dataset using linear probing on Nicheformer, UCE and CellPLM embeddings. The downstream tasks evaluated are niche regression for 4 different radius sizes. In all cases, Nicheformer outperforms both CellPLM and UCE, being the differences statistically significant. **D)** Downstream task metrics (MSE) for models trained in the CosMX human liver dataset using linear probing on Nicheformer, UCE and CellPLM embeddings. The downstream tasks evaluated are niche regression for 4 different radius sizes. In all cases, Nicheformer outperforms both CellPLM and UCE, being the differences statistically significant.

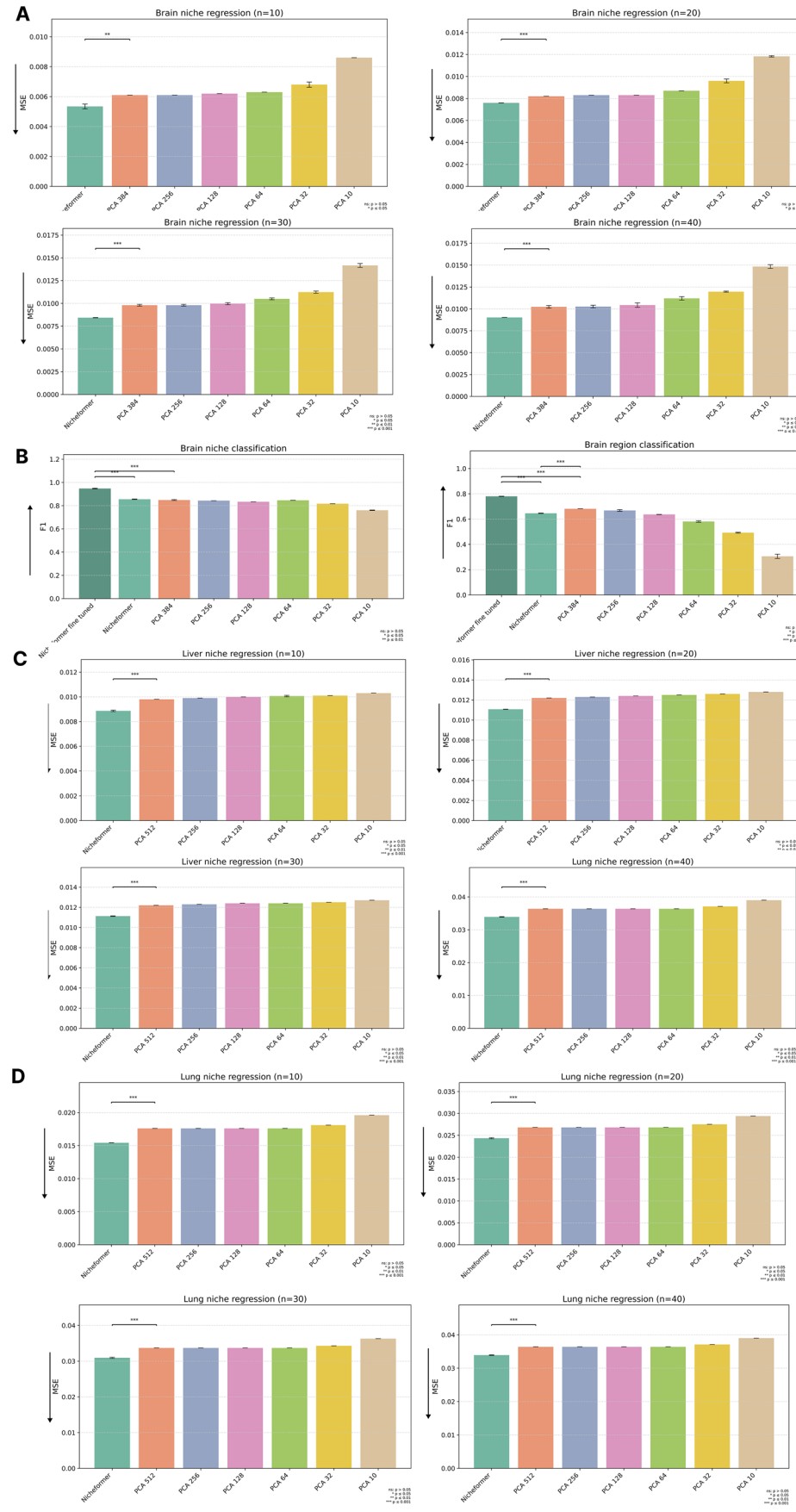

**Extended Data Fig. 7 | See next page for caption.**

**Extended Data Fig. 7 | Additional comparisons between Nicheformer and PCA for the MERFISH mouse brain, CosMX human liver and CosMX human lung datasets. A)** Downstream task metrics (MSE) for models trained in the MERFISH mouse brain using linear probing on Nicheformer and PCA embeddings with increasingly more principal components. The downstream tasks evaluated are niche regression for 4 different radius sizes. In all cases, Nicheformer outperforms PCA, even though the PCA substantially improves with the more principal components employed. Differences are found statistically significant between the best PCA performing model and Nicheformer. **B)** F1 Score for region and niche prediction. Interestingly, PCA ends up outperforming Nicheformer in the case of linear probing for the region classification and performing as good as Nicheformer for the niche classification. However, fine tuning Nicheformer is still better. **C)** Downstream task metrics (MSE) for models trained in the CosMX human liver dataset using linear probing on Nicheformer and PCA embeddings with increasingly more principal components. The downstream tasks evaluated are niche regression for 4 different radius sizes. In all cases, Nicheformer outperforms PCA, even though the PCA substantially improves with the more principal components employed. Differences are found statistically significant between the best PCA performing model and Nicheformer. **D)** Downstream task metrics (MSE) for models trained in the CosMX human lung dataset using linear probing on Nicheformer and PCA embeddings with increasingly more principal components. The downstream tasks evaluated are niche regression for 4 different radius sizes. In all cases, Nicheformer outperforms PCA, even though the PCA substantially improves with the more principal components employed. Differences are found statistically significant between the best PCA performing model and Nicheformer.

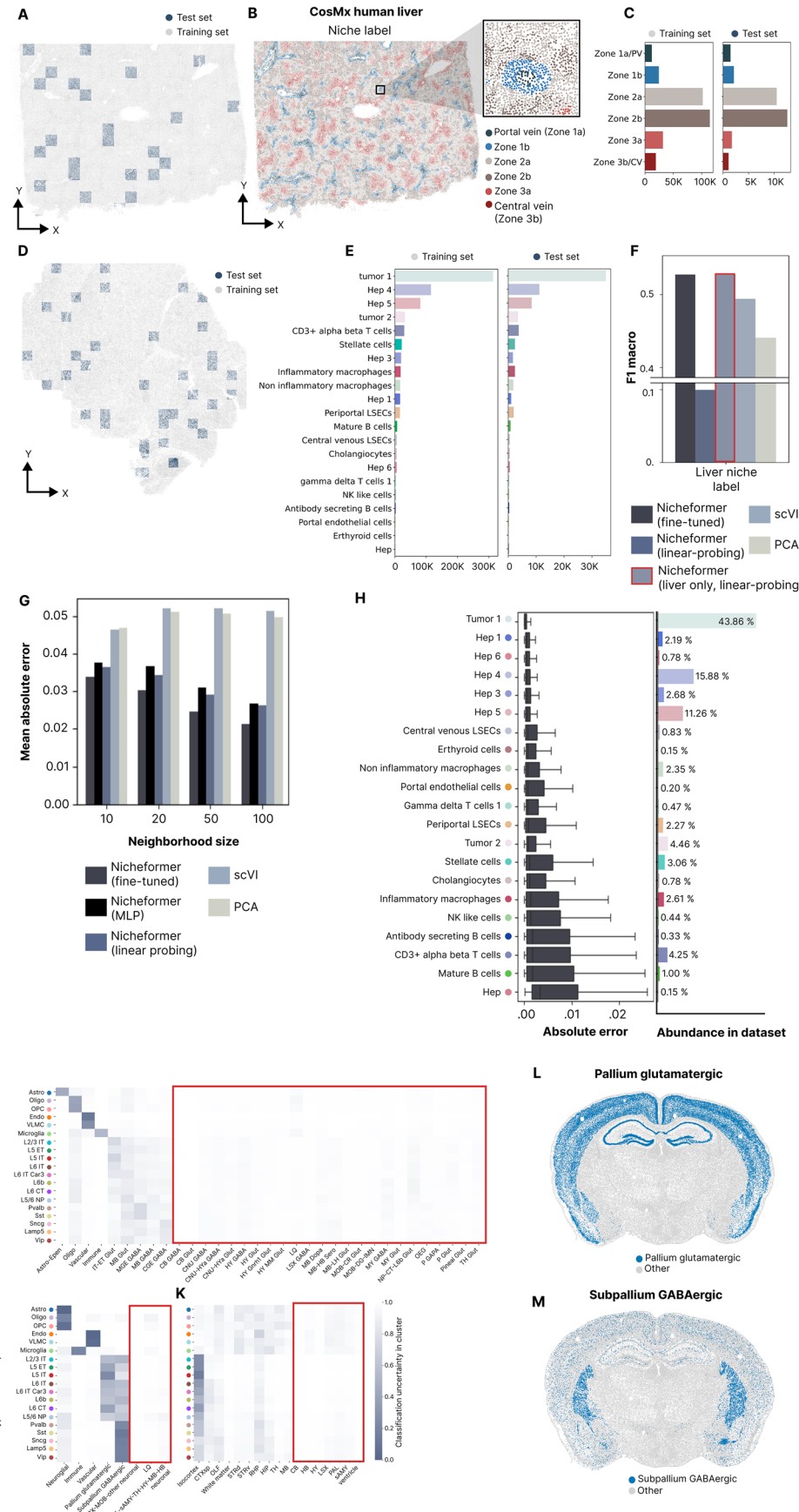

**Extended Data Fig. 8 | See next page for caption.**

**Extended Data Fig. 8 | Nicheformer fine-tuning datasets - CosMx human liver and spatial to dissociated label transfer. A-B)** Spatial allocation of cells in the healthy CosMx liver section colored by training and test split used for training Nicheformer (**A**) and niche label (**B**). **C)** Niche label distribution in the training and test set for the healthy CosMx liver dataset. **D)** Spatial allocation of cells in the cancer CosMx liver section colored by training and test split used for training Nicheformer in the cancer CosMx liver section. **E)** Distribution of cell type labels in the healthy and cancer CosMx liver data in both training and test set. **F)** Test-set F1-macro of niche label prediction of the fine-tuned Nicheformer model, the linear probing model, the linear probing model evaluated on a Nicheformer model longer trained in the liver training-set, and a linear probing baseline computed based on embeddings generated with scVI and PCA, respectively. **G)** The fine-tuned, a multi-task MLP on top of the Nicheformer embedding and the linear probing Nicheformer models outperform zero-shot models trained on scVI and PCA embeddings in terms of mean absolute error across all neighborhood sizes and all three organs, the brain, liver, and lung. **H)** Left: Fine-tuned Nicheformer performance on the CosMx human liver data grouped by index cell type. Shown are the absolute error values between predicted and observed niche composition vectors for held-out test cells. For each box in (H), the centerline defines the median, the height of the box is given by the interquartile range (IQR), the whiskers are given by 1.5 × IQR and outliers are given as points beyond the minimum or maximum whisker. Right: Index cell type abundances in the entire CosMx human liver dataset. **I-M) Nicheformer label transfer classification uncertainty from spatial to dissociated assays in the MERFISH mouse brain dataset. I-K)** Cell type (**I**), niche (**J**), and region (**K**) predicted label uncertainty across all cell types in the scRNA-seq mouse brain data. Nicheformer assigns lower uncertainty to plausible labels given the nature of the dataset and high uncertainty to labels not present in the primary motor cortex. The highlighted boxes show cell types, niches and regions one would not expect to find in the primary motor cortex. Nicheformer correctly shows a high uncertainty in those. **L-M)** Spatial allocation of cells in an exemplary section of the MERFISH mouse brain dataset colored by the pallium glutamatergic niche label (**L**) and the subpallium GABAergic niche label (**M**), respectively.

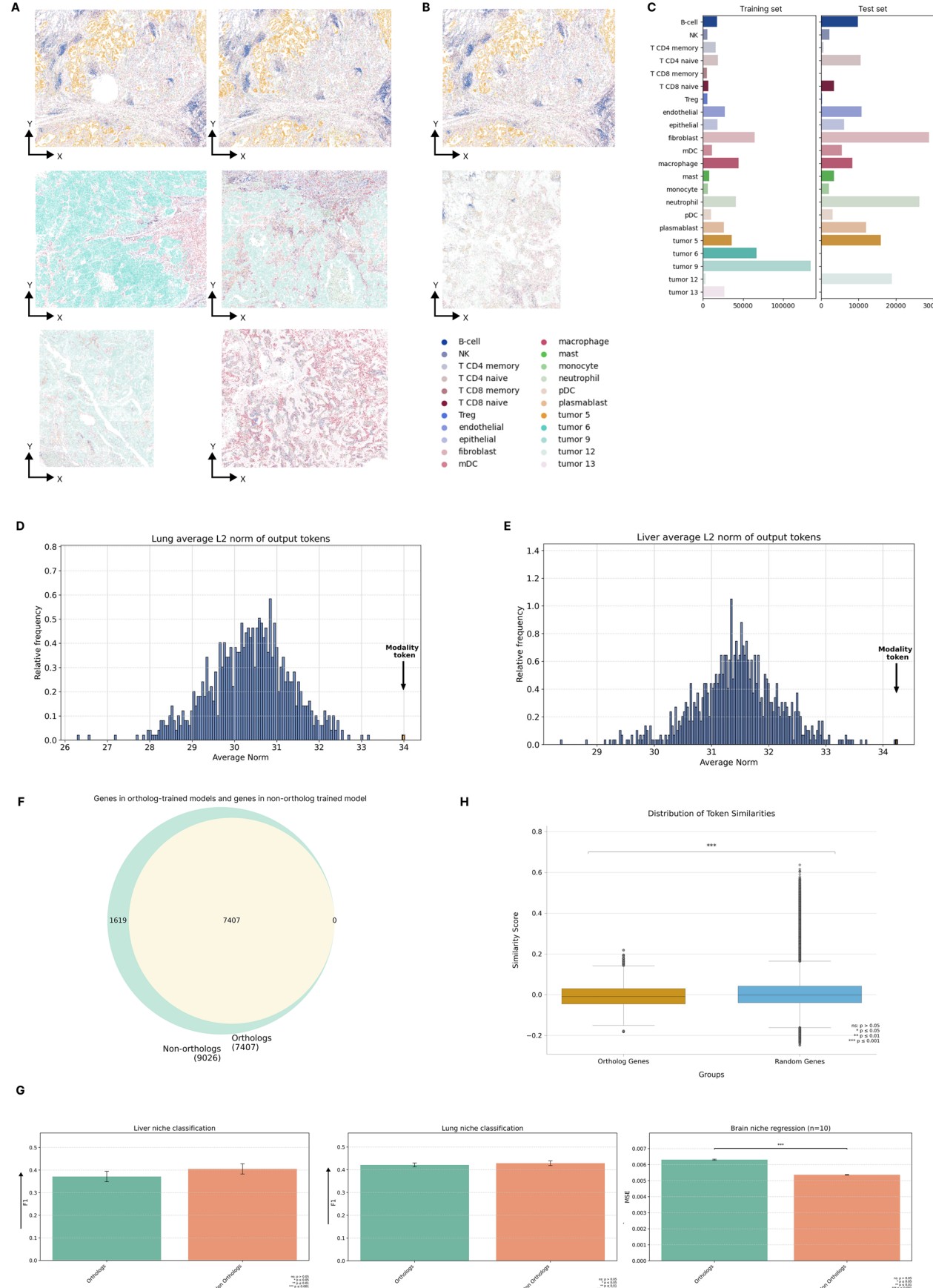

**Extended Data Fig. 9 | See next page for caption.**

**Extended Data Fig. 9 | Nicheformer fine-tuning datasets - CosMx human lung; output token norm analysis and orthologs comparison. A-B)** Spatial allocation of cells in the training set (**A**) and test set (**B**) tissue sections colored by cell type. **C)** Distribution of cell type labels in the training and test set in the CosMx human lung dataset. **D-C) Histogram of output token L2 norms for CosMx human lung and liver cells. D-C)** The histograms display the distribution of the average L2 norm of output tokens for lung (**D**) and liver (**E**) cells. The modality token, marked by an arrow, exhibits a notably higher norm compared to other tokens. These norms reflect the representation magnitudes in the model's output space. Including contextual tokens in cell representation aggregation led to poor label transfer performance. This is because aggregation is performed via mean pooling, where tokens with higher norms disproportionately influence the result. Additionally, contextual tokens appear in all cells, whereas the other tokens shown here are present only in specific subsets. As a result, while contextual tokens contribute to all cells, non-contextual tokens contribute only to the cells in which they appear. **F-H) Orthologs versus non orthologs comparison. F)** Venn diagram showing the number of genes of the non orthologs-trained model (9026) and the orthologs-trained model (7407). The 1619 genes of difference are genes that have a corresponding ortholog but we choose not to use the mapping. **G)** Niche regression in the MERFISH mouse brain dataset is the only downstream task - among the tested ones - in which there is a statistical significant difference (t-test) between both models. No statistical significance was found in the case of niche prediction for the CosMX human datasets. **H)** Boxplots showing the distribution of similarities between tokens measured as cosine similarity. We use the official Ensembl releases to map ortholog genes and assess if they are more similar between them than to random genes and we find that they are actually less similar.

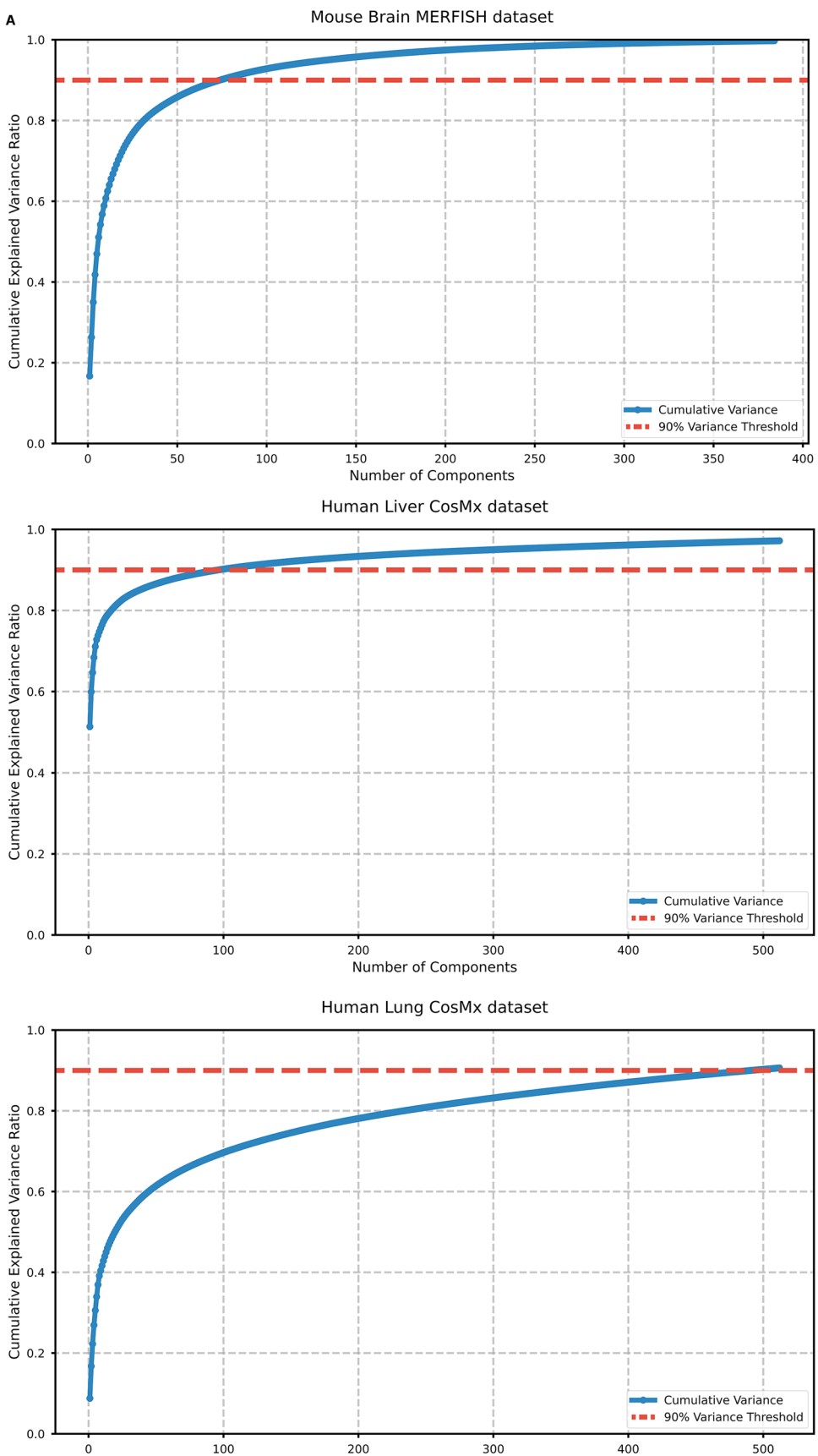

**Extended Data Fig. 10 | Cumulative explained variance ratio for the MERFISH brain mouse, the CosMx liver human and the CosMx lung human.** Shown are the cumulative explained variance ratios obtained after performing PCA. for the MERFISH brain mouse (top), CosMx human liver (middle) and CosMx human lung (bottom) datasets. Notice that this accounts for the explained variance in the train set, not in the test set (the PCA is computed in the train set and the test data transformer using the principal components obtained). The red line indicates the 90% of explained variance.

| | Corresponding author(s): | Fabian Theis |
|---|---|---|
| | Last updated by author(s): | 5/5/2025 |

# Reporting Summary

## Statistics

For all statistical analyses, confirm that the following items are present in the figure legend, table legend, main text, or Methods section.

| n/a | Confirmed | |
|---|---|---|
| ☐ | ☒ | The exact sample size (*n*) for each experimental group/condition, given as a discrete number and unit of measurement |
| ☐ | ☒ | A statement on whether measurements were taken from distinct samples or whether the same sample was measured repeatedly |
| ☐ | ☒ | The statistical test(s) used AND whether they are one- or two-sided<br>*Only common tests should be described solely by name; describe more complex techniques in the Methods section.* |
| ☐ | ☒ | A description of all covariates tested |
| ☐ | ☒ | A description of any assumptions or corrections, such as tests of normality and adjustment for multiple comparisons |
| ☐ | ☒ | A full description of the statistical parameters including central tendency (e.g. means) or other basic estimates (e.g. regression coefficient) AND variation (e.g. standard deviation) or associated estimates of uncertainty (e.g. confidence intervals) |
| ☐ | ☒ | For null hypothesis testing, the test statistic (e.g. *F*, *t*, *r*) with confidence intervals, effect sizes, degrees of freedom and *P* value noted<br>*Give P values as exact values whenever suitable.* |
| ☒ | ☐ | For Bayesian analysis, information on the choice of priors and Markov chain Monte Carlo settings |
| ☒ | ☐ | For hierarchical and complex designs, identification of the appropriate level for tests and full reporting of outcomes |
| ☒ | ☐ | Estimates of effect sizes (e.g. Cohen's *d*, Pearson's *r*), indicating how they were calculated |

*Our web collection on statistics for biologists contains articles on many of the points above.*

## Software and code

Policy information about availability of computer code

| Data collection | *Provide a description of all commercial, open source and custom code used to collect the data in this study, specifying the version used OR state that no software was used.* |
|---|---|
| Data analysis | Python 3.10, Pytorch 2.3, scikit-learn 1.6.1, scGPT 0.2.1, GeneFormer official HF, scvi-tools 1.2.0 |

For manuscripts utilizing custom algorithms or software that are central to the research but not yet described in published literature, software must be made available to editors and reviewers. We strongly encourage code deposition in a community repository (e.g. GitHub). See the Nature Portfolio guidelines for submitting code & software for further information.

## Data

Policy information about availability of data

All manuscripts must include a data availability statement. This statement should provide the following information, where applicable:
- Accession codes, unique identifiers, or web links for publicly available datasets
- A description of any restrictions on data availability
- For clinical datasets or third party data, please ensure that the statement adheres to our policy

The Allen brain atlas consortium generated the Allen Institute brain atlas mouse p20, Allen Institute brain atlas mouse p28, and Allen Institute brain atlas mouse female datasets (Suppl. Table 4), which have kindly been provided to us prior to publication. The Xenium human spinal cord, the ISS human brain GBM, the ISS human discover healthy lung, the ISS mouse EAE MS, and the Xenium mouse brain datasets have been generated by Mat Nillson lab and have been kindly provided

to us prior to publication.

All datasets used in this study are publicly available. The single-cell RNA sequencing data can be accessed through the Gene Expression Omnibus (GEO) under the following accession numbers: GSE117824 (DOI: 10.1038/s41586-019-1367-0), GSE118068 (DOI: 10.1038/s41586-019-1158-7), GSE119940 (DOI: 10.1038/s41590-019-0403-4), GSE124952 (DOI: 10.1038/s41467-019-12054-3), GSE126060 (DOI: 10.1038/s41467-019-14172-4), GSE128423 (DOI: 10.1016/j.cell.2019.04.040), GSE128761 (DOI: 10.1016/j.stem.2020.08.001), GSE128987 (DOI: 10.1038/s41556-020-00619-0), GSE129826 (DOI: 10.1016/j.celrep.2019.10.073), GSE130593 (DOI: 10.1242/dev.183251), GSE130822 (DOI: 10.1016/j.stem.2019.12.011), GSE130879 (DOI: 10.1016/j.immuni.2019.12.002), GSE130888 (DOI: 10.1126/scitranslmed.aav5341), GSE131339 (DOI: 10.1038/s41467-019-13465-y), GSE131996 (DOI: 10.1016/j.immuni.2019.06.009), GSE132355 (DOI: 10.1038/s41467-020-18231-z), GSE133531 (DOI: 10.1038/s41588-019-0531-7), GSE134571 (DOI: 10.1038/s41586-019-1535-2), GSE135310 (DOI: 10.1161/CIRCRESAHA.120.317200), GSE135326 (DOI: 10.1038/s41586-019-1644-y), GSE135356 (DOI: 10.1038/s41467-020-18957-w), GSE135414 (DOI: 10.1016/j.celrep.2020.01.075), GSE136394 (DOI: 10.1158/2326-6066.CIR19-0299), GSE136441 (DOI: 10.1073/pnas.2005570117), GSE137026 (DOI: 10.1084/jem.20220126), GSE139168 (DOI: 10.1126/sciadv.aba9950), GSE140510 (DOI: 10.1038/s41591-019-0695-9), GSE140628 (DOI: 10.1158/2159-8290.CD19-0958), GSE141471 (DOI: 10.1038/s41467-021-27899-w), GSE141526 (DOI: 10.1126/sciadv.abm7981), GSE141552 (DOI: 10.1093/hmg/ddaa038), GSE141784 (DOI: 10.1084/jem.20192362), GSE142143 (DOI: 10.1038/s41419-022-04693-0), GSE142797 (DOI: 10.1101/2020.04.27.063503), GSE143293 (DOI: 10.1038/s41586-020-3017-y), GSE145216 (DOI: 10.1016/j.cell.2020.03.004), GSE145251 (DOI: 10.1016/j.stem.2022.03.001), GSE145326 (DOI: 10.1172/JCI130323), GSE145689 (DOI: 10.1681/ASN.2020070930), GSE145866 (DOI: 10.1016/j.celrep.2020.107952), GSE146122 (DOI: 10.1016/j.cell.2020.03.015), GSE146138 (DOI: 10.1053/j.gastro.2020.09.011), GSE146194 (DOI: 10.1016/j.cell.2022.05.013), GSE146298 (DOI: 10.1016/j.celrep.2020.03.059), GSE146512 (DOI: 10.1016/j.celrep.2020.108027), GSE148339 (DOI: 10.1016/j.jcmgh.2020.07.012), GSE148978 (DOI: 10.1016/j.immuni.2020.10.024), GSE149040 (DOI: 10.1038/s41467-021-21704-4), GSE149201 (DOI: 10.1158/2159-8290.CD-20-0461), GSE149356 (DOI: 10.1126/sciimmunol.abf0125), GSE149931 (DOI: 10.1038/s41587-020-00763-w), GSE150708 (DOI: 10.21203/rs.3.rs-62758/v1), GSE150871 (DOI: 10.1038/s41586-020-2795-6), GSE150995 (DOI: 10.1172/JCI136142), GSE151186 (DOI: 10.1016/j.stem.2020.10.003), GSE152325 (DOI: 10.1038/s41556-020-00617-2), GSE152573 (DOI: 10.1182/blood.2020007747), GSE152988 (DOI: 10.1038/s41593-021-00862-0), GSE152999 (DOI: 10.1038/s41467-020-20351-5), GSE153099 (DOI: 10.1016/j.stemcr.2020.12.018), GSE153117 (DOI: 10.1038/s41467-021-26069-2), GSE153274 (DOI: 10.1038/s41467-021-22021-6), GSE153288 (DOI: 10.1038/s41467-020-17544-3), GSE153762 (DOI: 10.7554/eLife.60223), GSE153770 (DOI: 10.1016/j.celrep.2020.108004), GSE153802, GSE154196 (DOI: 10.15252/embj.2020106423), GSE154359 (DOI: 10.1016/j.stem.2020.08.015), GSE154386 (DOI: 10.1371/journal.ppat.1009240), GSE154567 (DOI: 10.1016/j.celrep.2020.108590), GSE154579 (DOI: 10.1038/s41467-020-19234-6), GSE154932 (DOI: 10.1088/1478-3975/abb09c), GSE155226 (DOI: 10.1126/scitranslmed.abf7872), GSE155340 (DOI: 10.1126/sciimmunol.abb5168), GSE155788 (DOI: 10.7554/eLife.61413), GSE155850 (DOI: 10.1172/jci.insight.139932), GSE156136 (DOI: 10.1016/j.cell.2020.09.062), GSE156183 (DOI: 10.1073/pnas.2017742118), GSE156245 (DOI: 10.1038/s41586-021-03283-y), GSE156285 (DOI: 10.1126/sciimmunol.abc6259), GSE156920 (DOI: 10.1038/s41467-021-23320-8), GSE157244 (DOI: 10.1161/JAHA.120.019019), GSE157292 (DOI: 10.1172/jci.insight.141321), GSE157362 (DOI: 10.1242/dev.197111), GSE157525 (DOI: 10.1101/gad.339978.120), GSE157771 (DOI: 10.1038/s41593-020-00745-w), GSE157773, GSE157977 (DOI: 10.1126/science.aaz6063), GSE158038 (DOI: 10.1038/s41467-021-22210-3), GSE158192 (DOI: 10.1038/s41467-021-22817-6), GSE158356_mouse (DOI: 10.26508/lsa.202000935), GSE158450 (DOI: 10.1038/s41467-020-20343-5), GSE159354 (DOI: 10.1016/j.xcrm.2020.100140), GSE159519 (DOI: 10.1016/j.cell.2020.10.030), GSE159977 (DOI: 10.1038/s41586-021-03362-0), GSE160061 (DOI: 10.1111/cpr.12933), GSE160097 (DOI: 10.1002/eji.202048797), GSE160098 (DOI: 10.1038/s41467-023-38647-7), GSE160664 (DOI: 10.1164/rccm.202008-3198OC), GSE160729 (DOI: 10.1016/j.cmet.2020.12.004), GSE160772 (DOI: 10.1096/fj.202002123R), GSE161066 (DOI: 10.3389/fphys.2021.637924), GSE161227 (DOI: 10.1084/jem.20212479), GSE161230, GSE161363 (DOI: 10.1126/science.abc1944), GSE161685 (DOI: 10.1172/jci.insight.144294), GSE161937 (DOI: 10.1073/pnas.1915389116), GSE162073 (DOI: 10.1084/jem.20200844), GSE162807 (DOI: 10.1038/s41467-022-28473-8), GSE163018 (DOI: 10.1038/s41421-021-00266-1), GSE163278 (DOI: 10.1172/jci.insight.127807), GSE163650 (DOI: 10.1371/journal.pone.0244743), GSE163668 (DOI: 10.1038/s41586-021-03234-7), GSE163701 (DOI: 10.1038/s41698-021-00160-9), GSE163830, GSE163919, GSE164044 (DOI: 10.1016/j.neuron.2019.08.002), GSE164573 (DOI: 10.1038/s41467-021-22842-5), GSE165551 (DOI: 10.7554/eLife.67436), GSE165554 (DOI: 10.7554/eLife.67436), GSE166218 (DOI: 10.1093/neuonc/noac138), GSE166262 (DOI: 10.1038/s41588-021-00818-x), GSE166525 (DOI: 10.1186/s13046-023-02686-1), GSE166797 (DOI: 10.1073/pnas.2023070118), GSE166992 (DOI: 10.1016/j.celrep.2021.108863), GSE167595 (DOI: 10.1158/1940-6207.CAPR-21-0378), GSE167992 (DOI: 10.1016/j.stem.2021.04.003), GSE168732 (DOI: 10.1038/s41467-021-25771-5), GSE168758 (DOI: 10.1016/j.jhep.2021.03.029), GSE169718 (DOI: 10.1016/j.devcel.2021.12.012), GSE172127 (DOI: 10.1038/s41421-021-00266-1), GSE200218 (DOI: 10.1016/j.cell.2022.06.007), GSE225278 (DOI: 10.1038/s41467-023-38704-1), GSE114687 (DOI: 10.1038/s41588-019-0489-5), GSE117176 (DOI: 10.1172/jci.insight.126453), GSE117770 (DOI: 10.1016/j.cmet.2019.01.021), GSE120508 (DOI: 10.1038/s41422-018-0099-2), GSE122342 (DOI: 10.1016/j.stem.2018.12.015), GSE122960 (DOI: 10.1164/rccm.201712-24100C), GSE123722 (DOI: 10.1016/j.cell.2020.11.017), GSE124691 (DOI: 10.1016/j.celrep.2019.10.131), GSE128855 (DOI: 10.1038/s41593-019-0393-4), GSE129519 (DOI: 10.1038/s41586-019-1289-x), GSE130238 (DOI: 10.1016/j.stem.2019.08.002), GSE131685 (DOI: 10.1038/s41597-019-0351-8), GSE132672 (DOI: 10.1038/s41586-020-1962-0), GSE135893 (DOI: 10.1126/sciadv.aba1972), GSE136001 (DOI: 10.1038/s41467-021-21407-W), and GSE136103 (DOI: 10.1038/s41586-019-1631-3).

The Allen brain atlas consortium generated the Allen Institute brain atlas mouse p20, Allen Institute brain atlas mouse p28, and Allen Institute brain atlas mouse female datasets (Suppl. Table 4), which have kindly been provided to us prior to publication. The Xenium human spinal cord, the ISS human brain GBM, the ISS human discover healthy lung, the ISS mouse EAE MS, and the Xenium mouse brain datasets have been generated by Mat Nillson lab and have been kindly provided to us prior to publication.

All datasets used in this study are publicly available. The single-cell RNA sequencing data can be accessed through the Gene Expression Omnibus (GEO) under the following accession numbers: GSE117824 (DOI: 10.1038/s41586-019-1367-0), GSE118068 (DOI: 10.1038/s41586-019-1158-7), GSE119940 (DOI: 10.1038/s41590-019-0403-4), GSE124952 (DOI: 10.1038/s41467-019-12054-3), GSE126060 (DOI: 10.1038/s41467-019-14172-4), GSE128423 (DOI: 10.1016/j.cell.2019.04.040), GSE128761 (DOI: 10.1016/j.stem.2020.08.001), GSE128987 (DOI: 10.1038/s41556-020-00619-0), GSE129826 (DOI: 10.1016/j.celrep.2019.10.073), GSE130593 (DOI: 10.1242/dev.183251), GSE130822 (DOI: 10.1016/j.stem.2019.12.011), GSE130879 (DOI: 10.1016/j.immuni.2019.12.002), GSE130888 (DOI: 10.1126/scitranslmed.aav5341), GSE131339 (DOI: 10.1038/s41467-019-13465-y), GSE131996 (DOI: 10.1016/j.immuni.2019.06.009), GSE132355 (DOI: 10.1038/s41467-020-18231-z), GSE133531 (DOI: 10.1038/s41588-019-0531-7), GSE134571 (DOI: 10.1038/s41586-019-1535-2), GSE135310 (DOI: 10.1161/CIRCRESAHA.120.317200), GSE135326 (DOI: 10.1038/s41586-019-1644-y), GSE135356 (DOI: 10.1038/s41467-020-18957-w), GSE135414 (DOI: 10.1016/j.celrep.2020.01.075), GSE136394 (DOI: 10.1158/2326-6066.CIR19-0299), GSE136441 (DOI: 10.1073/pnas.2005570117), GSE137026 (DOI: 10.1084/jem.20220126), GSE139168 (DOI: 10.1126/sciadv.aba9950), GSE140510 (DOI: 10.1038/s41591-019-0695-9), GSE140628 (DOI: 10.1158/2159-8290.CD19-0958), GSE141471 (DOI: 10.1038/s41467-021-27899-w), GSE141526 (DOI: 10.1126/sciadv.abm7981), GSE141552 (DOI: 10.1093/hmg/ddaa038), GSE141784 (DOI: 10.1084/jem.20192362), GSE142143 (DOI: 10.1038/s41419-022-04693-0), GSE142797 (DOI: 10.1101/2020.04.27.063503), GSE143293 (DOI: 10.1038/s41586-020-3017-y), GSE145216 (DOI: 10.1016/j.cell.2020.03.004), GSE145251 (DOI: 10.1016/j.stem.2022.03.001), GSE145326 (DOI: 10.1172/JCI130323), GSE145689 (DOI: 10.1681/ASN.2020070930), GSE145866 (DOI: 10.1016/j.celrep.2020.107952), GSE146122 (DOI: 10.1016/j.cell.2020.03.015), GSE146138 (DOI: 10.1053/j.gastro.2020.09.011), GSE146194 (DOI: 10.1016/j.cell.2022.05.013), GSE146298 (DOI: 10.1016/j.celrep.2020.03.059), GSE146512 (DOI: 10.1016/j.celrep.2020.108027), GSE148339 (DOI: 10.1016/j.jcmgh.2020.07.012), GSE148978 (DOI: 10.1016/j.immuni.2020.10.024), GSE149040 (DOI: 10.1038/s41467-021-21704-4), GSE149201 (DOI: 10.1158/2159-8290.CD-20-0461), GSE149356 (DOI: 10.1126/sciimmunol.abf0125), GSE149931 (DOI: 10.1038/s41587-020-00763-w), GSE150708 (DOI: 10.21203/rs.3.rs-62758/v1), GSE150871 (DOI: 10.1038/s41586-020-2795-6), GSE150995 (DOI: 10.1172/JCI136142), GSE151186 (DOI: 10.1016/j.stem.2020.10.003), GSE152325 (DOI: 10.1038/s41556-020-00617-2), GSE152573 (DOI: 10.1182/blood.2020007747), GSE152988 (DOI: 10.1038/s41593-021-00862-0), GSE152999 (DOI: 10.1038/s41467-020-20351-5), GSE153099 (DOI: 10.1016/j.stemcr.2020.12.018), GSE153117 (DOI: 10.1038/s41467-021-26069-2), GSE153274 (DOI: 10.1038/s41467-021-22021-6), GSE153288 (DOI: 10.1038/s41467-020-17544-3), GSE153762 (DOI: 10.7554/eLife.60223), GSE153770 (DOI: 10.1016/j.celrep.2020.108004), GSE153802, GSE154196 (DOI: 10.15252/embj.2020106423), GSE154359 (DOI: 10.1016/j.stem.2020.08.015), GSE154386 (DOI: 10.1371/journal.ppat.1009240), GSE154567 (DOI: 10.1016/j.celrep.2020.108590), GSE154579 (DOI: 10.1038/s41467-020-19234-6), GSE154932 (DOI: 10.1088/1478-3975/abb09c), GSE155226 (DOI: 10.1126/scitranslmed.abf7872), GSE155340 (DOI: 10.1126/sciimmunol.abb5168), GSE155788 (DOI: 10.7554/eLife.61413), GSE155850 (DOI: 10.1172/jci.insight.139932), GSE156136 (DOI: 10.1016/j.cell.2020.09.062), GSE156183 (DOI: 10.1073/pnas.2017742118), GSE156245 (DOI: 10.1038/s41586-021-03283-y),

GSE156285 (DOI: 10.1126/sciimmunol.abc6259), GSE156920 (DOI: 10.1038/s41467-021-23320-8), GSE157244 (DOI: 10.1161/JAHA.120.019019), GSE157292 (DOI: 10.1172/jci.insight.141321), GSE157362 (DOI: 10.1242/dev.197111), GSE157525 (DOI: 10.1101/gad.339978.120), GSE157771 (DOI: 10.1038/s41593-020-00745-w), GSE157773, GSE157977 (DOI: 10.1126/science.aaz6063), GSE158038 (DOI: 10.1038/s41467-021-22210-3), GSE158192 (DOI: 10.1038/s41467-021-22817-6), GSE158356_mouse (DOI: 10.26508/lsa.202000935), GSE158450 (DOI: 10.1038/s41467-020-20343-5), GSE159354 (DOI: 10.1016/j.xcrm.2020.100140), GSE159519 (DOI: 10.1016/j.cell.2020.10.030), GSE159977 (DOI: 10.1038/s41586-021-03362-0), GSE160061 (DOI: 10.1111/cpr.12933), GSE160097 (DOI: 10.1002/eji.202048797), GSE160098 (DOI: 10.1038/s41467-023-38647-7), GSE160664 (DOI: 10.1164/rccm.202008-31980C), GSE160729 (DOI: 10.1016/j.cmet.2020.12.004), GSE160772 (DOI: 10.1096/fj.202002123R), GSE161066 (DOI: 10.3389/fphys.2021.637924), GSE161227 (DOI: 10.1084/jem.20212479), GSE161230, GSE161363 (DOI: 10.1126/science.abc1944), GSE161685 (DOI: 10.1172/jci.insight.144294), GSE161937 (DOI: 10.1073/pnas.1915389116), GSE162073 (DOI: 10.1084/jem.20200844), GSE162807 (DOI: 10.1038/s41467-022-28473-8), GSE163018 (DOI: 10.1038/s41421-021-00266-1), GSE163278 (DOI: 10.1172/jci.insight.127807), GSE163650 (DOI: 10.1371/journal.pone.0244743), GSE163668 (DOI: 10.1038/s41586-021-03234-7), GSE163701 (DOI: 10.1038/s41698-021-00160-9), GSE163830, GSE163919, GSE164044 (DOI: 10.1016/j.neuron.2019.08.002), GSE164573 (DOI: 10.1038/s41467-021-22842-5), GSE165551 (DOI: 10.7554/eLife.67436), GSE165554 (DOI: 10.7554/eLife.67436), GSE166218 (DOI: 10.1093/neuonc/noac138), GSE166262 (DOI: 10.1038/s41588-021-00818-x), GSE166525 (DOI: 10.1186/s13046-023-02686-1), GSE166797 (DOI: 10.1073/pnas.2023070118), GSE166992 (DOI: 10.1016/j.celrep.2021.108863), GSE167595 (DOI: 10.1158/1940-6207.CAPR-21-0378), GSE167992 (DOI: 10.1016/j.stem.2021.04.003), GSE168732 (DOI: 10.1038/s41467-021-25771-5), GSE168758 (DOI: 10.1016/j.jhep.2021.03.029), GSE169718 (DOI: 10.1016/j.devcel.2021.12.012), GSE172127 (DOI: 10.1038/s41421-021-00266-1), GSE200218 (DOI: 10.1016/j.cell.2022.06.007), GSE225278 (DOI: 10.1038/s41467-023-38704-1), GSE114687 (DOI: 10.1038/s41588-019-0489-5), GSE117176 (DOI: 10.1172/jci.insight.126453), GSE117770 (DOI: 10.1016/j.cmet.2019.01.021), GSE120508 (DOI: 10.1038/s41422-018-0099-2), GSE122342 (DOI: 10.1016/j.stem.2018.12.015), GSE122960 (DOI: 10.1164/rccm.201712-24100C), GSE123722 (DOI: 10.1016/j.cell.2020.11.017), GSE124691 (DOI: 10.1016/j.celrep.2019.10.131), GSE128855 (DOI: 10.1038/s41593-019-0393-4), GSE129519 (DOI: 10.1038/s41586-019-1289-x), GSE130238 (DOI: 10.1016/j.stem.2019.08.002), GSE131685 (DOI: 10.1038/s41597-019-0351-8), GSE132672 (DOI: 10.1038/s41586-020-1962-0), GSE135893 (DOI: 10.1126/sciadv.aba1972), GSE136001 (DOI: 10.1038/s41467-021-21407-W), and GSE136103 (DOI: 10.1038/s41586-019-1631-3).

## Human research participants

Policy information about studies involving human research participants and Sex and Gender in Research.

| Reporting on sex and gender | N/A |
| --- | --- |
| Population characteristics | N/A |
| Recruitment | N/A |
| Ethics oversight | N/A |

Note that full information on the approval of the study protocol must also be provided in the manuscript.

# Field-specific reporting

Please select the one below that is the best fit for your research. If you are not sure, read the appropriate sections before making your selection.

☒ Life sciences ☐ Behavioural & social sciences ☐ Ecological, evolutionary & environmental sciences

For a reference copy of the document with all sections, see nature.com/documents/nr-reporting-summary-flat.pdf

# Life sciences study design

All studies must disclose on these points even when the disclosure is negative.

| Sample size | No power analysis was performed to select sample size. However, we followed standard computational practices using at least 3 random seeds for comparing models. For the statistical analysis performed in the attention analysis section, we sampled at least 2,000 cells from each tissue and condition to have a sufficiently large sample size of attention scores to obtain robust results. For the rest of spatial predictive tasks (e.g. niche composition prediction), entire spatial slices were held-out and the performance of the model evaluated in all the cells of those spatial slices. For the label transfer tasks, we employed a dissociated data with more than 7,000 cells to ensure a sufficient sample size. |
| --- | --- |
| Data exclusions | No data was excluded from analyses. |
| Replication | Replication of the results were done, running different random seeds for each analysis, including splits. Furthermore, since both data, model weights and code are provided, all results can be reproduced using the official github repo. |
| Randomization | Randomization was done through random seeding. Also, covariates were controlled to ensure no data leakage. For instance, in the case of the spatial tasks (e.g. niche classification), it was controlled that no test data was leakage into training and validation sets. Furthermore, the splitting was done at random, sampling spatial slices randomly. Likewise for the attention analyses, all cells were randomly sampled. |
| Blinding | Blinding not relevant in the study. Train, validation, test splits were done at random. For attention analyses, all cells were also randomly sampled. |

# Reporting for specific materials, systems and methods

We require information from authors about some types of materials, experimental systems and methods used in many studies. Here, indicate whether each material, system or method listed is relevant to your study. If you are not sure if a list item applies to your research, read the appropriate section before selecting a response.

## Materials & experimental systems

| n/a | Involved in the study |
|-----|------------------------|
| ☒ | Antibodies |
| ☒ | Eukaryotic cell lines |
| ☒ | Palaeontology and archaeology |
| ☒ | Animals and other organisms |
| ☒ | Clinical data |
| ☒ | Dual use research of concern |

## Methods

| n/a | Involved in the study |
|-----|------------------------|
| ☒ | ChIP-seq |
| ☒ | Flow cytometry |
| ☒ | MRI-based neuroimaging |

