## [Peer Review File · Nature Methods]

Nicheformer: a foundation model for single-cell and spatial omics

Corresponding Author: Professor Fabian Theis

Version 0:

Decision Letter:

26th Nov 2024

Dear Professor Theis,

Your Article, "Nicheformer: a foundation model for single-cell and spatial omics", has now been seen by 2 reviewers. As you will see from their comments below, although the reviewers find your work of potential interest, they have raised a number of concerns. We are interested in the possibility of publishing your paper in Nature Methods, but would like to consider your response to these concerns before we reach a final decision on publication.

We therefore invite you to revise your manuscript to address these concerns. We think web-lab validation is not a prerequisite for the revision.

Link Redacted

We hope to receive your revised paper within 2 months. If you cannot send it within this time, please let us know. In this event, we will still be happy to reconsider your paper at a later date so long as nothing similar has been accepted for publication at Nature Methods or published elsewhere.

OPEN SCIENCE REQUIREMENTS

REPORTING SUMMARY AND EDITORIAL POLICY CHECKLISTS

Reporting summary: <https://www.nature.com/documents/nr-reporting-summary.zip>
Editorial policy checklist: <https://www.nature.com/documents/nr-editorial-policy-checklist.zip>

DATA AVAILABILITY

All novel DNA and RNA sequencing data, protein sequences, genetic polymorphisms, linked genotype and phenotype data, gene expression data, macromolecular structures, and proteomics data must be deposited in a publicly accessible database, and accession codes and associated hyperlinks must be provided in the "Data Availability" section.

CODE AVAILABILITY

Please include a "Code Availability" subsection in the Online Methods which details how your custom code is made available. Only in rare cases (where code is not central to the main conclusions of the paper) is the statement "available upon request" allowed (and reasons should be specified).

For more information on our code sharing policy and requirements, please see: <https://www.nature.com/nature-research/editorial-policies/reporting-standards#availability-of-computer-code>

MATERIALS AVAILABILITY

ORCID

Nature Methods is committed to improving transparency in authorship. As part of our efforts in this direction, we are now requesting that all authors identified as 'corresponding author' on published papers create and link their Open Researcher and

Contributor Identifier (ORCID) with their account on the Manuscript Tracking System (MTS), prior to acceptance. This applies to primary research papers only. ORCID helps the scientific community achieve unambiguous attribution of all scholarly contributions. You can create and link your ORCID from the home page of the MTS by clicking on 'Modify my Springer Nature account'. For more information please visit www.springernature.com/orcid.

Sincerely,

Lin Tang, PhD
Senior Editor
Nature Methods

Reviewers' Comments:

Reviewer #1 (Remarks to the Author):

Comment on "Nicheformer: a foundation model for single-cell and spatial omics, by Schaar et al.

Summary:

The authors present Nicheformer, a transformer-based foundation model designed for spatial single-cell and spatial transcriptomics data. Nicheformer bridges the gap between dissociated and spatial transcriptomics by leveraging a large dataset, a SpatialCorpus-110M, which includes over 110 million cells (57 million dissociated and 53 million spatially resolved cells) from humans and mice across 73 tissues. A brief summary of results: Nicheformer can

1) perform spatial composition prediction and spatial label prediction.

2) existing foundation models trained on dissociated single-cell data alone – the authors compare with scGPT, Geneformer, scVI (and PCA) – concluding that those models are not capable of recapitulating the spatial complexity of cells in their microenvironments

3) transfer of spatial information to scRNA-seq datasets.

The study is carefully executed in its technical design. The aim of developing a foundation model incorporating both single-cell RNAseq data and spatial transcriptomics data is a natural and appreciated "next step" in the genomics community. The manuscript is well written, the introduction is clear, and setting up the context and problem is appropriate. However, there are a number of details missing, the GitHub needs work, and there are conceptual and practical issues with the authors logic of their benchmark (comparative) analysis.

Furthermore, it is difficult to be thrilled about the results and conclusions. What I mean is, for example, result (2) above -- that, indeed, we would expect a model incorporating data modalities A and B to perform "better" than a model using only A. Now, surprisingly, that information from B can be transferred / leveraged to A (i.e., result (3) above) is to be expected. Since Nicheformer works with spatial transcriptomics data (B), the observation (1) above is not revolutionary.

This assessment may sound more negative than intended, but overall, while the manuscript is a natural next conceptual step towards a foundation model, the results read and give the impression of an incremental achievement rather than a significant step forward. Let me provide more specific critiques and suggestions below.

Major points:

1. The Comparative Analysis. The authors benchmark nicheformer against scGPT, Geneformer, scVI. Now, there are a couple of principal concerns.

(a) Why is it meaningful to compare against models that are trained only on scRNAseq data? What do we learn from this? Models using modalities A + B outperforms models using modality A. It is not surprising. Can the authors elaborate on the specific insights we obtain from such an analysis? Can the authors motivate or justify the reasons and potential power of such an analysis.

(b) More technically, Geneformer is trained on 30 million cells, whereas Nicheformer uses 57 million (dissociated) cells. Is the performance difference due to (i) the extra 27 million cells, (ii) the existence of spatial transcriptomics, or (iii) just that the sheer number of cells (110 million) is larger?

(c) Let's assume that such a comparison is indeed warranted. Then why not do the benchmarking with additional scRNAseq-based models, where some are suggested to outperform Geneformer, just to name one possibility? Candidate models to

assess against Nicheformer include CellFM1 (trained on 100 million cells, outperforms scGPT and Geneformer on zero-shot tasks), scFoundation2 (50 million cells, 100 tissue types), GeneCompass3 (120 million human and mouse cells), UCE – Universal Cell Embeddings4 (multiple species and tissues, outperforms scGPT and Geneformer on clustering, batch effects, ...)

(d) A “proper” comparison would be to benchmark against a model using both scRNAseq and spatial transcriptomics data. While the authors are (almost) the first to develop such a model, CELLPLM is another such model. The authors note, “Notably, to date, none of these models account for spatial relationships of cells during training, with the exception of CellPLM, which, however, is trained on a limited dataset of 9 million dissociated and 2 million spatial transcriptomics cells and not fine-tuned on spatial tasks beyond gene imputation” [third paragraph introduction]. The authors reference – the bioarxiv 2023 – while the CellPLM has been accepted at ICLR 2024 (<https://openreview.net/forum?id=BKXvPDekud>) (i) even though cellplm (2023 version) is “small,” a comparative analysis would be informative for readers. Which differences between the two are due to scaling versus data modalities. This may also shed some light on the comparative analysis w.r.t. scRNAseq (only) models. (ii) Here, in the ICLR contribution, while pretrained on 9 and 2 million cells – as stated above – CellPLM is tested on zero-shot clustering, scRNA-seq denoising, spatial transcriptomic imputation, cell type annotation, and perturbation prediction. Thus, not only gene imputation, as stated by the authors. Hence, a comparative analysis using the latest CellPLM version is needed.

2. The GitHub, code, and reproducibility. I asked one of the senior computer scientists on the team to take a look. He is very experienced in constructing and checking all our codes in research and GitHub pages related to our publications. There were a number of issues. In summary:

- (a) Downstream tasks were missing. Only tokenization was demonstrated.
- (b) There is no guide for utilizing in-house/new datasets, making the process challenging to follow and reuse. For example, instructions on updating configuration files could be clearer.
- (c) Detailed explanations of commands, such as those for get embeddings, training, and finetuning are missing.

Hence, The original results are difficult to replicate, and the tool is challenging to reuse with new datasets. So, while he was able to install the code, the absence/lack of examples, documentation, or guidance made it impossible to replicate the results and even more so when trying to use it with new datasets.

Please update accordingly and provide the necessary annotation. Considering the use of GitHub by a less experienced user, you may need to add more than what is mentioned here.

3. Is it possible to provide any guidelines w.r.t.

(a) How to balance the amount of data, and number of cells – when considering data modalities – how much-dissociated scRNAseq is needed versus spatial transcriptomics data. Would more spatial data compensate “better” when having less dissociated data – or – could a vast amount of dissociated data make up most of the signal, whereas the spatial makes a minor (smaller) contribution? See supp Fig 3. Would more scRNAseq data reduce the difference between the curves? Different considerations for different downstream tasks ?

(b) How about guidelines for balancing mouse and human data? In the specific analysis presented in Nicheformer – to what extent are the mouse data important? Is it just a number game, more data, or does the mouse data bring something qualitatively specific?

4. The authors use orthologous genes, which makes sense as a first approximation. Yet, there are some potential inherent challenges. These include the fact that the function of the orthologue genes in mice and humans may be different, and there might be expression differences between such mouse and human orthologue genes. Could the “spatial context” between two orthologous genes be different? Did the authors investigate whether (i) those potential challenges turned up in their data and analysis, (ii) and, if so, what (size) effects they had on different downstream tasks?

5. Tissues: It appears that Nicheformer is doing its best in brain tissue, but there is less of a clear difference with liver tissue (Fig 5c). Why is this? The liver data comes from the mouse, right (Fig 2b). Considering that the liver data is spatial, one would expect that this would be very beneficial for nicheformer compared to the pure scRNAseq models, but it appears not to be that boost one would expect to see – under the assumption that spatial data really makes a difference.

6. Hyperparameters. I appreciate the clarity in presenting this – often overlooked – topic (Supp Table 1 &5). I think this part still would benefit from a deeper analysis. In essence, we have – the number of transformer blocks/stacked layers – and – the number of parallel attention mechanisms (heads) in each layer – and – the gene tokens, which would (?) depend on the modality of the dataset – spatial versus dissociated.

(a) did the authors use the same hyperparameters for all data (spatial and dissociated). I do recognize that the author tried different hyperparameters (good!!!) and settled for specific values (Table 1). Were these the same for both mice and humans, dissociated and spatial.

(b) Here, I would suspect that due to differences between dissociated versus spatial data, there would be (should be) differences. For example, in spatial, there is less of a number of genes – note that the authors use image-based spatial transcriptomics data – whereas the dissociated data have more comprehensive expression profiles. Wouldn't you then need more tokens for the dissociated data and vice versa? The size of the data sets would be biased towards more or less the

number of attention heads. Since the heads in different layers attend to different “things” (dissociated versus spatial aspects), would you then not expect different constraints with the number of heads in different layers? Fewer attention heads for spatial data? Would a particular final choice of hyperparameters tilt Nicheformer to take dissociated or spatial data into “more” account or give it a larger relative weight?

(c) same as (b) – but instead of contrasting dissociated versus spatial, rather – was there any difference w.r.t. best hyperparameters for mouse vs human?

7. The attention analysis. I find this part very interesting, but it does not really come through in the ms. The early layers find gene-gene interactions, whereas the spatial comes in later layers and metadata in the last layers.

(a) These observations – as far as I can tell from the ms – are made in the brain data, which is the premier showcase for nicheformer w.r.t. performance. Do these observations generalize to other tissues? Is this a generic scheme that is reflected here in the brain tissue?

(b) Why do we see this – a division of labor – between the layers? Is this an effect of finely tuned hyperparameters where we have large enough data of different modalities, allowing the authors to have a sufficient number of layers and attention heads to disentangle such a fine-graining of the representations? Smaller data sets, less spatial, and so forth... would (potentially?) mask this scheme.

(c) Do the authors observe this in the lung and liver tissues?

A sample of technical details that are missing:

- The model pads missing genes with <pAD> tokens and sets all sequences to a fixed length of 1,500 tokens. However, spatial technologies often measure far fewer genes than scRNA-seq. This is particularly true for image-based spatial technologies – which the authors rely on here. The paper doesn't explain how the model deals with these substantial differences in sequence lengths or the potential loss of information due to truncation. While the results claim the model is robust to incomplete gene panels, the methods lack details on how missing data affects gene ranking and model performance. High proportions of <pAD> tokens in spatial data might also impact the attention mechanisms in the model.
- There is minimal information on how scVI and PCA embeddings were generated (e.g., choice of latent dimensions, hyperparameters). Without consistent configurations, differences in performance may arise from suboptimal baseline settings rather than a genuine superiority of Nicheformer. Detailed baseline configurations are essential for a fair comparison.
- The paper does not mention any techniques to address class imbalance.
- The train-test split varies between datasets (e.g., random field of view hold-out vs. random cell hold-out). Different splitting strategies may introduce biases, such as including correlated cells in both sets or not accounting for donor variability.
- Contextual tokens for <aSSAY>, <MODALITY>, and <ORGANISM> are prepended to input sequences but are excluded during embedding aggregation for downstream tasks. Excluding contextual tokens during aggregation may remove important modality-specific information that the model has learned to incorporate during pretraining. Reassess the exclusion of contextual tokens during embedding aggregation or justify their omission.

REFERENCES

1. Zeng, Y. et al. CellFM: a large-scale foundation model pre-trained on transcriptomics of 100 million human cells. Preprint at <https://doi.org/10.1101/2024.06.04.597369> (2024).
2. Hao, M. et al. Large-scale foundation model on single-cell transcriptomics. *Nat Methods* (2024) doi:10.1038/s41592-024-02305-7.
3. Yang, X. et al. GeneCompass: Deciphering Universal Gene Regulatory Mechanisms with Knowledge-Informed Cross-Species Foundation Model The X-Compass Consortium. (2023) doi:10.1101/2023.09.26.559542.
4. Rosen, Y. et al. Universal Cell Embeddings: A Foundation Model for Cell Biology-Oct-6-2024. (2024) doi:10.1101/2023.11.28.568918.

Reviewer #1 (Remarks on code availability):

see point 2 in the reply to authors

Reviewer #2 (Remarks to the Author):

Similar to many existing studies on single-cell foundation model building, the authors proposed to build for their own foundation model. The topic is trendy but well-trying by others as evidenced by the competing methods shown by the authors themselves as well as simple Google searches. The proposed model is mostly based on transformer architecture. The results are mostly numeric for performance comparisons with downstream tasks in spatial transcriptomes. I have few comments:

1. One of the most controversial concerns on such a foundation topic is whether it is necessary to have deep learning foundation model as there is already a study arguing that PCA was already enough [1]. The authors are clearly aware of it,

based on the extensive PCA comparisons in the study. However, the current comparisons with PCA are still not enough. The most concerning part to me is that the PCA comparison is based on 30 components as stated by the authors: "We use the sklearn implementation with the number of components being set to 30. ". Other number of PCA components should be examined for fairness as it is targeted for foundation modelling but not single-task model.

2. Furthermore, how much variance the PCA has explained? It is very important for PCA tuning to such an extension that the majority of data variance can be explained. Indeed, by looking at Figure 4b, it is very clear that PCA could already be enough and it could be embarrassing to get published just for the sake of complicated AI in biology here with the exponentially increased model complexity in both time and space.

3. The word "Corpus" is mostly about text data in the NLP community. I have no idea why the authors adopted such a word for spatial transcriptomics.

4. Another challenging aspect here is the so-called "foundation model". Isn't it just a feature extraction or dimensionality reduction model? At the end of the day, it is still a model adopted by the related researchers only in a niche area. How would it be broadly adopted by biologists easily? For now, it is mostly released with source code on GitHub. I cannot see any web interfaces or easy-to-use functionalities. It is quite hard to justify for its broad impacts with the word "foundation model".

5. As the submitting journal is "Nature Methods", wet-lab validation is missing here. It is not clear if the model can reveal truly novel insights based on existing well-known data crawled from the web, although training and testing are separated.

6. Different transcriptomic sequencing technologies / platforms have different data biases. It is not clear if the existing data normalization steps are sufficient and integrated into a single "corpus".

7. Nonetheless, I do appreciate the use of orthologs in such a modelling. It is largely missed by the existing models. However, p-values and FDRs are missing in the current study.

[1] Bendidi, I., Whitfield, S., Kenyon-Dean, K., Yedder, H. B., Mesbahi, Y. E., Noutahi, E., & Denton, A. K. (2024). Benchmarking Transcriptomics Foundation Models for Perturbation Analysis: one PCA still rules them all. arXiv preprint arXiv:2410.13956.

Version 1:

Decision Letter:

Our ref: NMETH-A58401A

25th Mar 2025

Dear Dr. Theis,

Thank you for submitting your revised manuscript "Nicheformer: a foundation model for single-cell and spatial omics" (NMETH-A58401A). It has now been seen by the original referees and their comments are below. The reviewers find that the paper has improved in revision, and therefore we'll be happy in principle to publish it in Nature Methods, pending minor revisions to satisfy the referees' final requests and to comply with our editorial and formatting guidelines.

TRANSPARENT PEER REVIEW

Nature Methods offers a transparent peer review option for new original research manuscripts submitted from 17th February 2021. We encourage increased transparency in peer review by publishing the reviewer comments, author rebuttal letters and editorial decision letters if the authors agree. Such peer review material is made available as a supplementary peer review file.

Please state in the cover letter 'I wish to participate in transparent peer review' if you want to opt in, or 'I do not wish to participate in transparent peer review' if you don't. Failure to state your preference will result in delays in accepting your manuscript for publication.

ORCID

Sincerely,

Reviewer #1 (Remarks to the Author):

Comments on the revised manuscript "Nicheformer: a foundation model for single-cell and spatial omics, by Schaar et al.

The authors addressed the main concerns, including the
Lack of comparative analysis. Here, the authors performed extensive benchmarking using different subsets of data (including species) to evaluate how different modalities (dissociated vs. spatial) impact model performance. This analysis clarified the importance of spatial data, which could not be compensated for by the information in the dissociated cells.
Robustness of attention mechanisms. The additional analysis was carefully executed, and it was established that the observed structured hierarchy in attention layers was generalized to all the other tissues beyond the brain.

Additional benchmarking against the other models (Universal Cell Embeddings (UCE) and CellPLM, along with PCA, scVI, scGPT, and Geneformer) as requested. The results confirmed Nicheformer's superior performance.

The GitHub Repository was updated, and we confirmed that it was easier to install and use, including tutorials for tokenization, model training, and downstream tasks.

The authors responded and performed a proper analysis addressing the points. Furthermore, I appreciate that the authors could deepen their understanding of the attention analysis, which now has a more substantial backing in the revised manuscript.

I believe the authors are to be commended for an excellent revision, and I have no further requests.

My only remaining suggestions are:

1. The authors note that the <MODALITY> and <ORGANISM> tokens have high norms in the last layer of the transformer model, which could overshadow the gene expression signals if included in the final cell embedding. To address this, they excluded these contextual tokens when computing the embedding. However, a potential downside of this "all or nothing" approach is by completely excluding the <MODALITY> and <ORGANISM> tokens from the final embedding, the model may lose valuable information for tasks where modality or organism context is relevant. They may consider this for future research or to briefly discuss it in the manuscript as a possible extension of their approach.

2. We checked their GitHub and noticed improvements, particularly in the following aspects: (a) Improvement in the installation process, making setup easier and faster. (b) The addition of exemplary notebooks covering various tasks provides better guidance for users and enables them to run and adapt the notebooks to their data. Another point we noted is that there have been 17 open issues from different users (since last year), highlighting different problems or doubts related to the tool. Most of them remain unresolved, and addressing them would improve the user experience and the tool's performance. (issues: <https://github.com/theislab/nicheformer/issues>)

Reviewer #1 (Remarks on code availability):

My only remaining suggestions are:

2. We checked their GitHub and noticed improvements, particularly in the following aspects: (a) Improvement in the installation process, making setup easier and faster. (b) The addition of exemplary notebooks covering various tasks provides better guidance for users and enables them to run and adapt the notebooks to their data. Another point we noted is that there have been 17 open issues from different users (since last year), highlighting different problems or doubts related to the tool. Most of them remain unresolved, and addressing them would improve the user experience and the tool's performance. (issues: <https://github.com/theislab/nicheformer/issues>)

Reviewer #2 (Remarks to the Author):

Since the handling editor still gave a revision chance for the authors with only 2 reviews without any wet-lab verification on Nature Methods, I do not have any further comment although the authors are so confident in their final acceptance without due respects on reviewers.

Version 2:

Decision Letter:

11th Aug 2025

Dear Professor Theis,

I am pleased to inform you that your Article, "Nicheformer: a foundation model for single-cell and spatial omics", has now been accepted for publication in Nature Methods. The received and accepted dates will be 24th Oct 2024 and 11th Aug 2025. This note is intended to let you know what to expect from us over the next month or so, and to let you know where to address any further questions.

Over the next few weeks, your paper will be copyedited to ensure that it conforms to Nature Methods style. Once your paper is typeset, you will receive an email with a link to choose the appropriate publishing options for your paper and our Author Services team will be in touch regarding any additional information that may be required. It is extremely important that you let us know now whether you will be difficult to contact over the next month. If this is the case, we ask that you send us the contact information (email, phone and fax) of someone who will be able to check the proofs and deal with any last-minute problems.

Authors may need to take specific actions to achieve compliance with funder and institutional open access mandates.

If your research is supported by a funder that requires immediate open access (e.g. according to [Plan S principles](https://www.springernature.com/gp/open-science/plan-s-compliance) or the [NIH public access policy](https://www.springernature.com/gp/open-science/us-federal-agency-compliance)) then you should select the gold OA route, and we will direct you to the compliant route where possible. Because authors warrant under our subscription licensing terms that they haven't committed to licensing any version of their article under a licence inconsistent with the terms of our agreement – including the applicable embargo period – publication under the subscription model isn't suitable for authors whose funders require no embargo.

If you are active on Twitter/X or Bluesky, please e-mail me your and your coauthors' handles so that we may tag you when the paper is published.

Please feel free to contact me if you have questions about any of these points. Thank you very much for publishing your paper at Nature Methods!

Best regards,

Lin Tang, PhD
Senior Editor
Nature Methods

** Visit the Springer Nature Editorial and Publishing website at www.springernature.com/editorial-and-publishing-jobs for more information about our career opportunities. If you have any questions please click here.**

Open Access This Peer Review File is licensed under a Creative Commons Attribution 4.0 International License, which permits use, sharing, adaptation, distribution and reproduction in any medium or format, as long as you give appropriate credit to the original author(s) and the source, provide a link to the Creative Commons license, and indicate if changes were made. In cases where reviewers are anonymous, credit should be given to 'Anonymous Referee' and the source.

Response to reviewers:

Nicheformer: a foundation model for single-cell and spatial omics [NMETH-A58401]

Original reviewer comments (black), point-by-point responses (green), sections of the text (*blue italic*)

Dear Lin,

Thanks a lot for fast and constructive review of our Nicheformer manuscript. Based on these reviews we have identified the key issues with the study as lack of comparative analysis to delineate the effect of different modalities and tissues, lack of robustness in the attention analysis, some to-dos in the PCA comparison analysis and insufficient code in the Github repository.

To address these concerns we have made the following major changes:

- We conducted an **extensive series of computational experiments**, where Nicheformer models, with the same number of parameters and compute, were trained on different subsets of data, each containing the same number of cells. By varying the data subsets to isolate specific modalities and organisms, we demonstrated that only models trained on a diverse corpus encompassing all these covariates consistently achieved strong performance, demonstrating both robustness of the final model as well as the importance of comprehensive data diversity. We additionally show that not even increasing by 3X the amount of dissociated data, a dissociated-only model can keep up with the performance of a model trained on spatial data, showing the **need for the spatial modality** as asked by the reviewers.
- We have significantly deepened our **attention analysis ie interpretation of the trained models**, uncovering a more nuanced behavior across different layers of Nicheformer. Specifically, we found that later layers tend to focus primarily on metadata tokens, while middle layers direct sharp attention to specific genes, and early layers exhibit more scattered attention. Furthermore, these findings are robust across multiple organisms, as demonstrated by our new experiments with lung and liver data, showing that the attention patterns hold consistently across biological contexts. Additionally, our analysis confirms that this behavior remains consistent across modalities, including dissociated data. We also identify attention heads with very specific behaviours that are consistent across organisms and data modalities, as well as attention heads that adapt to the modality at hand.

- We have now **benchmarked against more existing foundation models such as Universal Cell Embeddings (UCE)** and also against **CellPLM**, a somewhat more targeted spatial foundation model. We find that Nicheformer majorly outperforms both of them in the more complex spatial tasks. We have also increased our benchmark against PCA, as demanded by the authors, finding again the need for model complexity of Nicheformer. This is important since for more simpler disassociated tasks, current papers critically discuss power of foundation models such as Geneformer or scGPT.
- We have **overhauled the Github repository making Nicheformer an easily installable package** and provided tutorial notebooks to tokenize data, use the model for downstream tasks and even continue the pretraining in in-house datasets. We also support all the pipeline directly from h5ad files making all the tokenization under the hood to ease its use.

In the following, we present our response to the reviewer's comments. We include **reviewer comments (black)**, **point-by-point responses (green)** and in parts copy *sections of the text (blue italic)* or specific figure panels.

Thank you and the reviewers again for the important contributions.

Regards,
Fabian

Reviewer #1 (Remarks to the Author):

Comment on “Nicheformer: a foundation model for single-cell and spatial omics, by Schaar et al.

Summary:

The authors present Nicheformer, a transformer-based foundation model designed for spatial single-cell and spatial transcriptomics data. Nicheformer bridges the gap between dissociated and spatial transcriptomics by leveraging a large dataset, a SpatialCorpus-110M, which includes over 110 million cells (57 million dissociated and 53 million spatially resolved cells) from humans and mice across 73 tissues. A brief summary of results: Nicheformer can

1) perform spatial composition prediction and spatial label prediction.

2) existing foundation models trained on dissociated single-cell data alone – the authors compare with scGPT, Geneformer, scVI (and PCA) – concluding that those models are not capable of recapitulating the spatial complexity of cells in their microenvironments

3) transfer of spatial information to scRNA-seq datasets.

We thank the reviewer for the accurate summary of our work. We are addressing individual points raised below in our point-by-point response.

The study is carefully executed in its technical design. The aim of developing a foundation model incorporating both single-cell RNAseq data and spatial transcriptomics data is a natural and appreciated “next step” in the genomics community. The manuscript is well written, the introduction is clear, and setting up the context and problem is appropriate. However, there are a number of details missing, the GitHub needs work, and there are conceptual and practical issues with the authors logic of their benchmark (comparative) analysis.

Furthermore, it is difficult to be thrilled about the results and conclusions. What I mean is, for example, result (2) above -- that, indeed, we would expect a model incorporating data modalities A and B to perform “better” than a model using only A. Now, surprisingly, that information from B can be transferred / leveraged to A (i.e., result (3) above) is to be expected. Since Nicheformer works with spatial transcriptomics data (B), the observation (1) above is not revolutionary.

Thanks for appreciating the work and technical design, and agreeing that this is a natural next step. We agree that you would expect the model to perform better on more complex tasks. The potentially really important point next to the model and the significant data collection in our opinion is that we came up with tasks that work on disassociated samples - ie could be in principle performed by cell-only focused models - that are actually *hard*. As has been shown in many recent papers, cell-only foundation models are not really all that good i.e. are often not outperforming simpler, more specific models such as scVI or even PCA. But for niche labeling, compositional or density inference,

this is clearly not the case. So in addition to the actual model, we expect these tasks to help benchmark and leverage more of the expected but often not really demonstrated power of LLM-like approaches in single cell biology. Please see below for significantly updated benchmarks, and thanks again for the clear and honest statement to which we agree.

This assessment may sound more negative than intended, but overall, while the manuscript is a natural next conceptual step towards a foundation model, the results read and give the impression of an incremental achievement rather than a significant step forward. Let me provide more specific critiques and suggestions below.

Major points:

1. The Comparative Analysis. The authors benchmark nicheformer against scGPT, Geneformer, scVI. Now, there are a couple of principal concerns.

(a) Why is it meaningful to compare against models that are trained only on scRNAseq data? What do we learn from this? Models using modalities A + B outperforms models using modality A. It is not surprising. Can the authors elaborate on the specific insights we obtain from such an analysis? Can the authors motivate or justify the reasons and potential power of such an analysis.

First, we of course agree, see above. But it is still incredibly useful to *use* that additional modality, make all the work of aggregating sufficient such data and then building such a model that does more than the existing ones - maybe a bit like early NLP LLMs then adding computer vision, natural, not astonishing that they could do vision-related tasks better but of course useful.

So indeed, this comparison is the most natural to make, as most existing models are trained solely on scRNA-seq data, and our novelty lies in incorporating spatial transcriptomics to enable joint training to prove that it is needed to “close the gap” between modalities. We agree that the comparison against foundation models trained on only scRNA-seq is not sufficiently highlighted. Specifically, we observed that existing scRNA-seq foundational models do not perform well on tasks designed for spatially defined downstream tasks. To us, the specific insight one can obtain from this analysis is that models trained on one modality and one scale alone are not yet capable of fully understanding the biological complexity of cellular microenvironments. To deepen further in this point, we have trained several models in different subsets of the data to study how different modalities perform in different downstream tasks. The results are detailed in the next points.

(b) More technically, Genformer is trained on 30 million cells, whereas Nicheformer uses 57 million (dissociated) cells. Is the performance difference due to (i) the extra 27 million cells, (ii) the existence of spatial transcriptomics, or (iii) just that the sheer number of cells (110 million) is larger?

We very much agree that understanding and quantifying the impact of different data scales is crucial and thank the reviewer for that suggestion. However, we want to stress that fitting large-scale

models on millions of cells requires large computational resources (order of week on 16+ GPUs per run) and selecting extensive benchmarks on different data scales should be carefully considered.

We therefore decided to be mindful in our scaling analysis and quantified the performance differences between only-dissociated versus only-spatial in a 1% subset of our SpatialCorpus-110M across different data corpus sizes. Additionally, we also trained an only-dissociated model in a 3% subset, to evaluate whether 3x more dissociated data than spatial data can improve performance and perform as well as a model trained on spatial data. As a summary, we trained models in the following splits.

1. Random dissociated split containing 1% of the data
2. Random dissociated split containing 3% of the data
3. Stratified dissociated split containing 1% of the data (1/4 blood, 1/4 colon, 1/4 liver, 1/4 intestine, 1/4 lung, 1/4 brain)
4. Random spatial split containing 1% of the data

We evaluated different downstream tasks such as niche classification in the CosMx human lung and liver datasets and cell type classification and niche regression in the MERFISH mouse brain dataset. To perform a controlled experiment that clarifies the cause of the performance difference, we trained Nicheformer models, with the same number of parameters, and with the same computation power - 3 days in an entire node with 4 A100 40GB GPUs. We evaluate all models in the linear probing scenario, to highlight more the quality of the representations learnt during pretraining. We evaluate 3 random seeds.

The results clearly show that the fundamental differential aspect is the use of spatial data.

In all cases, the model trained on spatial data very significantly outperformed the models trained in the different subsets of dissociated data. The results show clearly that there is a fundamental difference between dissociated and spatial data. Not only the model trained in spatial data (random spatial split containing 1% of the data) outperforms models trained with the same number of cells but dissociated, but also it clearly outperforms the model trained in 3 times more dissociated data.

This is a strong result that showcases that there is no amount of dissociated data that can outperform models trained with spatial data evaluated in spatial tasks. Hence, the performance difference does not come from the sheer number of cells, the fundamental difference is the inclusion of spatial data. Training with both dissociated and spatial data enables performing well in both tasks and transferring labels from one modality to the other.

Furthermore, more important insights can be extracted out of the results.

In the first place, there exists differences in the performance between the models trained on a 1% random dissociated subset and a 3% dissociated subset. The model trained on less amount of data is slightly better than the model trained on more data. The reason is that the amount of computation spent in both models is the same (3 days in 4 A100 40GB GPUs). Hence, in the model with more data, less compute power is used per each observation, i.e. the model gets fewer updates per sample, which leads to slightly worse results. This result is indeed very interesting and aligned with previous findings in LLMs ¹.

In the second place, the stratified model is only better than the random model in the case of the liver. This is an interesting case that can be explained by looking at the proportion of each tissue found in the random subset. Specifically, in the random subset of data, 19% belong to blood, 19% of cells belong to the brain, 9% belong to the lung, and the rest of tissues contribute much less. Hence, in the random subset, brain cells are even more present than in the stratified split, and lung cells, while they are not that numerous, are the third majoritarian tissue, which would explain why small differences in performance.

We added the figure above as a new supplementary figure (Suppl. Fig. 3) and have expanded the manuscript to explain these results. In the first place, we have expanded our section on model design and training:

We confirmed technology-dependent biases between spatial and dissociated transcriptomics data through extensive experiments consisting on pretraining across multiple data splits and evaluating on downstream tasks (Methods). Specifically, we trained Nicheformer models, with the same number of parameters and FLOPs (floating point operations, a measure of computational cost), on the same number of cells. We consistently found that training exclusively on dissociated single-cell data resulted in lower performance across all downstream tasks (Suppl. Fig. 3), even if the amount of dissociated single-cell data is 3x more than the spatial data. This indicates that dissociated data alone can not resolve the sources of variation observed in spatial transcriptomics data. Similarly, we also conducted experiments, with the same setup (same number of cells, same number of parameters, same

compute spent) to evaluate the role of training with mouse and human data (Methods). Results show that training on only human or only mouse data worsens performance in the missing organism (Suppl. Fig. 4). In general, both results support Nicheformer's chosen training strategy, suggesting that a training broad coverage distribution is preferred to achieve good performance in a large variety of contexts.

We have also expanded our methods section to explain how the models were evaluated.

To analyse the need of training a model on a diverse train dataset, we conducted controlled experiments in which we pretrained Nicheformer models and tested them in different downstream tasks and tissues. Specifically, we pretrained Nicheformer models of 49.3M parameters using the same compute budget - 3 days in an entire node containing 4 A100 GPUs. Due to the large compute needed to retrain Nicheformer models using the entire SpatialCorpus-110M, we subset it for the experiments, so each model is pretraining in a 1% of that dataset (~1.1M cells).

In particular, we pretrained models in the following data splits: 1.1M randomly sampled spatial cells, 1.1M randomly sampled dissociated cells and 3.3M randomly sampled dissociated cells (to assess whether a large amount of dissociated cells can account for the lack of spatial information). Additionally, we also pretrained a model in 1.1M dissociated cells sampled in such a way that there is the same amount of cells from blood, colon, intestine, lung, liver and brain, to assess the effect of the tissue variability of the dataset. To assess the importance of multi-species datasets, we also pretrained models on 1.1M spatial cells sampled only from human and 1.1M spatial cells sampled only from mouse.

We evaluated the pretrained models on the following downstream tasks: niche prediction in the human liver and lung CosMX datasets, and cell type classification and niche regression in the mouse brain MERFISH dataset. In all cases, the models were evaluated in the linear probing scenario running 3 seeds.

We also provide a general remark on this point after point 3 (comparison between species), which we believe is related to this one.

(c) Let's assume that such a comparison is indeed warranted. Then why not do the benchmarking with additional scRNAseq-based models, where some are suggested to outperform Geneformer, just to name one possibility? Candidate models to assess against Nicheformer include CellFM1 (trained on 100 million cells, outperforms scGPT and Geneformer on zero-shot tasks), scFoundation2 (50 million cells, 100 tissue types), GeneCompass3 (120 million human and mouse cells), UCE – Universal Cell Embeddings4 (multiple species and tissues, outperforms scGPT and Geneformer on clustering, batch effects, ...)

We have expanded our comparison beyond GeneFormer and scGPT, now using CellPLM and also Universal Cell Embeddings (UCE). From both of them we use the latest versions provided in their official repositories. We want to highlight that we tried to use the other proposed models but they were not usable since their Githubs provide no tutorial notebooks or those are not reproducible, which is also shown by recent other comparisons on disassociated samples only.

We find that the performance of both of those models is worse than the performance of Nicheformer in all the downstream tasks tested, confirming what we see in (b) about modality extension for UCE, and missing robustness/scale/model detail for CellPLM. We have modified the text to mention both models in the comparison and added the figures with the comparisons as supplementary material.

We have added the following text in the Methods section:

To get UCE embeddings we used the latest version from the original repository and followed the tutorials to obtain the cell embeddings. The fraction of overlapping genes compared to the gene context used in scGPT was for the MERFISH mouse brain data 472/483 genes, for the CosMx human liver dataset 990/999 genes, and for the CosMx human lung dataset 954/960 genes.

For the comparison against CellPLM, we employed the latest official version of the repository. For the MERFISH mouse dataset we first mapped the mouse genes to human genes using BioMart² through the official Ensembl releases³. The fraction of overlapping genes compared to the gene context used in scGPT was for the MERFISH mouse brain data 473/483 genes, for the CosMx human liver dataset 997/999 genes, and for the CosMx human lung dataset 958/960 genes. The cell embeddings were obtained by following the notebook tutorials.

We have also expanded the test regarding the comparisons:

[...] Finally, we also evaluate the linear probing scenario for foundation models trained on dissociated data alone: GeneFormer, scGPT and UCE; and for a foundation model trained on spatial data: CellPLM.

We found that both of our approaches outperform both traditional embedding methods, scVI and PCA, no matter which was the training set, as well as GeneFormer, scGPT, UCE and CellPLM, in terms of macro F1 score, which effectively balances precision and recall across all classes (Fig. 4B, Suppl. Fig. 15).

[...]

We found that fine-tuned Nicheformer systematically outperforms the linear probing models trained on Nicheformer embedding, Geneformer, scGPT, scVI and PCA - independently of the number of principal components used (Suppl. Fig. 16) - for this task on all three organs in terms of mean absolute error. Likewise, for UCE and CellPLM, which we evaluated by training a linear layer on their embeddings, we also found that linear probing with Nicheformer outperforms both across all three datasets (Suppl. Fig. 15-17).

(d) A “proper” comparison would be to benchmark against a model using both scRNAseq and spatial transcriptomics data. While the authors are (almost) the first to develop such a model, CELLPLM is another such model. The authors note, “Notably, to date, none of these models account for spatial relationships of cells during training, with the exception of CellPLM, which, however, is trained on a limited dataset of 9 million dissociated and 2 million spatial transcriptomics cells and not fine-tuned on spatial tasks beyond gene imputation” [third paragraph introduction]. The authors reference – the

bioarxiv 2023 – while the CellPLM has been accepted at ICLR 2024 (i) even though cellplm (2023 version) is “small,” a comparative analysis would be informative for readers. Which differences between the two are due to scaling versus data modalities. This may also shed some light on the comparative analysis w.r.t. scRNAseq (only) models. (ii) Here, in the ICLR contribution, while pretrained on 9 and 2 million cells – as stated above – CellPLM is tested on zero-shot clustering, scRNA-seq denoising, spatial transcriptomic imputation, cell type annotation, and perturbation prediction. Thus, not only gene imputation, as stated by the authors. Hence, a comparative analysis using the latest CellPLM version is needed.

We thank the reviewer for raising this point and the not-updated citation for CellPLM. We updated the citation accordingly. As mentioned in the previous point, we have extended our comparison to CellPLM and put the comparison in the supplementary material.

For the benchmark, we have not taken the checkpoint of the initial CellPLM model of the preprint but the most updated one provided by the authors in the official Github repository. We have tested CellPLM in niche classification in the liver and lung datasets, as well as for niche regression in the brain datasets. Nicheformer greatly outperforms CellPLM in all tasks without need of fine tuning. Both models were compared in the linear probing scenario. We have added the results to the supplementary section, as indicated in the previous response.

2. The GitHub, code, and reproducibility. I asked one of the senior computer scientists on the team to take a look. He is very experienced in constructing and checking all our codes in research and GitHub pages related to our publications. There were a number of issues. In summary:

(a) Downstream tasks were missing. Only tokenization was demonstrated.

(b) There is no guide for utilizing in-house/new datasets, making the process challenging to follow and reuse. For example, instructions on updating configuration files could be clearer.

(c) Detailed explanations of commands, such as those for get embeddings, training, and finetuning are missing.

Hence, The original results are difficult to replicate, and the tool is challenging to reuse with new datasets. So, while he was able to install the code, the absence/lack of examples, documentation, or guidance made it impossible to replicate the results and even more so when trying to use it with new datasets.

Please update accordingly and provide the necessary annotation. Considering the use of GitHub by a less experienced user, you may need to add more than what is mentioned here.

We agree with the reviewer that the previously released GitHub repository was not yet entirely tailored for applying Nicheformer to new datasets and full exploration of all features of our work. We should have done this better already, but quite frankly find as many labs dealing with models at this scale initially a bit daunting. Since then we have significantly improved in our codebase setup, and therefore now released an updated version of the Nicheformer repository which specifically includes the following:

- Tutorial for preprocessing in-house/new datasets to be used by the model
- Tutorials on how to get embeddings, training and fine-tuning.
- Tutorials on how to continue the pretraining on in-house/new datasets. This also includes guidelines on the minimal computational resources required to re-train Nicheformer on a new dataset.

Nicheformer can now be easily installed using the following commands:

```
mamba create -n nicheformer_env python=3.10
mamba activate nicheformer_env
git clone https://github.com/theislab/nicheformer.git
cd nicheformer/
pip install -e .
```

Importantly, we identified the tokenization process as the biggest hurdle, so the new repo also now supports skipping the tokenization process since it is done automatically within the dataloading, to make it easier to use for the user.

We strongly believe that the provided updates enable researchers to use Nicheformer for the analysis of their datasets and testing of all released features.

We thank the reviewer again for so clearly pushing on the reproducibility point, which we find very important (in particular in the current foundation model hype).

3. Is it possible to provide any guidelines w.r.t.

(a) How to balance the amount of data, and number of cells – when considering data modalities – how much-dissociated scRNAseq is needed versus spatial transcriptomics data. Would more spatial data compensate “better” when having less dissociated data – or – could a vast amount of dissociated data make up most of the signal, whereas the spatial makes a minor (smaller) contribution? See supp Fig 3. Would more scRNAseq data reduce the difference between the curves? Different considerations for different downstream tasks ?

(b) How about guidelines for balancing mouse and human data? In the specific analysis presented in Nicheformer – to what extent are the mouse data important? Is it just a number game, more data, or does the mouse data bring something qualitatively specific?

Thank you for raising this point. We acknowledge this is not properly explained in the initial manuscript. As explained in point 1, to evaluate the role of the different modalities and organisms in the data, we have trained Nicheformer models in reduced subsets of the data.

For the first point (dissociated vs. spatial), we refer to our response in 1b. Models trained on spatial data clearly outperform models trained in 3x more dissociated data. We believe that is a strong proof

that both modalities have their own nuances and it is not possible to close the gap between them without training jointly.

For the second point (human vs. mouse) we have trained additional Nicheformer models under the same settings as the ones in point 1 (same number of cells, same number of parameters, same compute power):

1. Random spatial split with only human cells containing 1% of the data
2. Random spatial split with only mouse cells containing 1% of the data
3. Random spatial split with mouse and human cells containing 1% of the data.

Again, the results revealed that a broad coverage distribution is needed to achieve the best results in a large variety of scenarios. The model trained only on human underperformed on mouse, the model trained only on mouse underperformed on human, and the model trained on par performance with the models trained on the organism at hand. We have added the result as a supplementary figure (Supp. Fig. 5) and modified the text as follows:

[...]Similarly, we also conducted experiments, with the same setup (same number of cells, same number of parameters, same compute spent) to evaluate the role of training with mouse and human data (Methods). Results show that training on only human or only mouse data worsens performance in the missing organism (Suppl. Fig. 5). Models trained on one organism outperforms models trained on only the opposite organism, while a model trained on both achieves best performance across the organisms. Importantly, this result is not influenced by the sheer number of cells since all models are trained with the same number of cells, the only difference is the diversity of the data.

A general remark to both this point and point 1. First of all we want to thank the reviewer for prompting us to run these tests and analysis, since we reckon they have expanded our understanding of the model and the large model training paradigm in general. Secondly, we believe that both the results in this point and the ones in point 1 very much aligned and led to a conclusion: the more broad coverage in the train distribution, the better the model will perform. This might be obvious but it is indeed the result we get out of the analysis and it is very aligned with what has been observed in the LLM research field¹⁴. Probably, the next step, which requires a specific publication for it, is how to assess the quality and goodness of the data, which is also currently an active field of research^{5,6}.

4. The authors use orthologous genes, which makes sense as a first approximation. Yet, there are some potential inherent challenges. These include the fact that the function of the orthologue genes in mice and humans may be different, and there might be expression differences between such mouse and human orthologue genes. Could the “spatial context” between two orthologous genes be different? Did the authors investigate whether (i) those potential challenges turned up in their data and analysis, (ii) and, if so, what (size) effects they had on different downstream tasks?

We thank the reviewer for this point. The fact that orthologous genes might have different functions and spatial contexts was one of the reasons that motivated the introduction of the contextual token <ORGANISM>. That way, while the model sees the same gene token, it is aware of which is the organism being modelled, which allows it to adapt the computations accordingly. Basically, given enough data, enough computation, and enough parameters, the model should be able to adapt the computation given the existing context, which includes a strong signal (the <ORGANISM> token), and also the signal of the rest of genes present. After all, Nicheformer has 12 transformer blocks stacked to go from the learnt gene embedding to the final cell representation.

To explore whether there is any difference in the use of orthologue genes in terms of downstream tasks, we changed our vocabulary and train two Nicheformer models in a small subset of the data (0.5%) and using only spatial data during 2 days in a single node with 4 GPU A100. In particular, we compared a model with a vocabulary of 7407 genes (including orthologue) against a model that does not incorporate the orthologue genes, hence having a large vocabulary size: 9026. Then, we evaluated niche regression in CosMx human liver and lung and niche prediction in the MERFISH mouse brain dataset. We observed difference between both models only in the latter case.

Additionally, we also compared the similarity of genes in the model trained without orthologue genes. We used the official Ensembl releases to create a mapping between mouse and human genes and study whether the genes of those genes are more similar between them than to random genes. In other words, we examined whether, in a model without any ortholog prior mapping, the gene space cluster together the ortholog genes. To measure similarity we used cosine similarity.

We found that ortholog genes are less similar between them than random genes, i.e. human genes that have an equivalent in the mouse genome are less similar to that equivalent than to random genes (including both human and mouse ones). We tested the statistical significance of this result with a Mann-Whitney U Test. This can be explained by the fact that those tokens are never seen

together, since they belong to different species and hence they never take part of the same cellular context.

We have added the following supplementary figures (Suppl. Fig 25 & 26).

Supplementary Figure 24 | Orthologs versus non orthologs comparison. A) Venn diagram showing the number of genes of the non orthologs-trained model (9026) and the orthologs-trained model (7407). The 1619 genes of difference are genes that have a corresponding ortholog but we choose not to use the mapping. B) Niche regression in the MERFISH mouse brain dataset is the only downstream task - among the tested ones - in which there is a statistical significant difference (t-test) between both models. C) No statistically significance was found in the case of niche prediction for the CosMX human datasets.

Supplementary Figure 25 | Orthologs genes are less similar between them than to random genes. Boxplots showing the distribution of similarities between tokens measured as cosine similarity. We use the official Ensembl releases to map ortholog genes and assess if they are more similar between them than to random genes and we find that they are actually less similar.

We included also a section in Methods explaining this analysis:

Orthologs genes analysis

We conducted an attention analysis to study deeper the role of ortholog genes in Nicheformer and assess whether there were major differences between using or not using them and how they are related. To do so, we trained small Nicheformer models in a reduced gene space with and without using orthologs. Specifically, we used a gene vocabulary of 9026 genes, that when mapping ortholog is reduced to 7407 (Suppl. Fig. 25A). We compared the performance of both models with 3 different downstream tasks: niche prediction in the CosMX human lung and liver dataset and niche regression in the MERFISH mouse brain dataset. We found that there are just differences in the performance in the latter (Suppl. Fig. 25B-C).

Likewise, we studied, for the model without the ortholog mapping, whether genes with known cross-organism equivalent are more similar to their ortholog equivalent than to any other random gene. To analyse that, we extract the gene embeddings after the pretraining and analyse their cosine similarity. The results indicated that genes are less similar to their ortholog than to random genes, which can be explained by the fact that they are never seen together in any cell and that they might have different functions (Suppl. Fig. 26).

5. Tissues: It appears that Nicheformer is doing its best in brain tissue, but there is less of a clear difference with liver tissue (Fig 5c). Why is this? The liver data comes from the mouse, right (Fig 2b).

Considering that the liver data is spatial, one would expect that this would be very beneficial for nicheformer compared to the pure scRNAseq models, but it appears not to be that boost one would expect to see – under the assumption that spatial data really makes a difference.

We thank the reviewer for this point, this is indeed very interesting. But would like to note that the CosMx dataset highlighted through our manuscript is measured in a human patient diagnosed with Hepatocellular Carcinoma as well as a healthy control patient. More details on the dataset are also available in our method section in “Datasets used for downstream tasks and evaluations - CosMx human liver”:

[...] We collected the CosMx human liver dataset from the publicly available CosMx data resource¹²². The dataset comprises cells from both a normal healthy liver measuring 332,877 cells across 301 fields of views covering one tissue section in a male 35-year-old patient, as well as cells from a Hepatocellular Carcinoma measuring 460,441 cells across 383 fields of view in one tissue section from a 65-year-old female patient.

We still agree that the performance differences in brain versus liver are worth discussing in more detail. We discuss potential reasons for the performance differences in our manuscript.

[...] However, the Nicheformer linear probing model did not outperform linear probing models trained on scVI and PCA models - trained on the training set of the liver dataset (Suppl. Fig. 9F). We hypothesized that the worse linear probing performance is related to the insufficient model capacity due to limitations in the number of trainable parameters and training time. This prevents the model from accurately learning a nuanced representation of liver cells, which have relatively low overall abundance in the SpatialCorpus-110M (Fig. 2A, B). We therefore asked whether additional pretraining of Nicheformer solely on the liver training set improved performance in this organ. Indeed, we observed an improvement in linear probing performance for the prediction of the liver niche labels on the longer pretrained Nicheformer embedding (Suppl. Fig. 9F). In practice this would have to be done for each organ, so we propose to altogether pretrain for a longer time, to explore lower abundance states in the training corpus.

We consider this quite a bit surprising: the fact that the model performances changes so greatly with a few more targeted pretraining steps in a specific tissue. Actually, this fact is also related with the experiments shown in point 1b. Specifically, the performance in liver across all models is good. We find that this points out that there might be some undertraining when training in SpatialCorpus110M and that there might be delicate tradeoff between number of biological diversity (understood as number of tissues), sheer number of cells and compute time needed during pretraining. This plays interestingly in the current discussion whether model capacity and training time is even needed in foundation models - here clearly it *is*.

We linked the results of the tests we run for point 1 with this point, since there one can see that there is no drop in the performance in liver. This suggests that there is a tradeoff between training dataset size, number of parameters, data variability and compute time. It is indeed a very interesting result that can guide future research. We expanded the text referring this:

[...]. We conclude that longer pretraining can be beneficial for datasets with lower abundance in the training corpus.

Interestingly, the analysis performed to assess the importance of the pretraining data yielded new insights. In particular, we observed that in Nicheformer models trained with just ~1% data, there is no such a drop in performance. Additionally, we observed that the model trained on a smaller dissociated subset (1%) performs slightly better than one trained on a larger subset (3%) because both were trained with the same compute budget, leading to more updates per sample in the smaller dataset - more computation spent per sample. This strongly suggests that computational efficiency plays a large role in the performance of large models, where computational power should be scaled along with the dataset size, which supports the theory that Nicheformer might be undertrained for some tissues, but can be fixed with a small amount of pretraining in the affected tissue. This aligns with findings in Large Language Models (LLMs) and the so-called scaling laws¹, that indicates that compute budget and dataset size (also number of parameters) should be jointly scaled.

6. Hyperparameters. I appreciate the clarity in presenting this – often overlooked – topic (Supp Table 1 &5). I think this part still would benefit from a deeper analysis. In essence, we have – the number of transformer blocks/stacked layers – and – the number of parallel attention mechanisms (heads) in each layer – and – the gene tokens, which would (?) depend on the modality of the dataset – spatial versus dissociated.

We thank the reviewer for their feedback. We address the individual points raised below.

(a) did the authors use the same hyperparameters for all data (spatial and dissociated). I do recognize that the author tried different hyperparameters (good!!) and settled for specific values (Table 1). Were these the same for both mice and humans, dissociated and spatial.

This remark is valid, we indeed used the same hyperparameters for the entire pretraining dataset (spatial and dissociated, mouse and human). We specifically decided to not assess the optimal set of hyperparameters for different species or modalities to reduce any additional design choices during our pretraining. However, we did run ablation studies on a limited set of hyperparameters as for example shown in Suppl. Fig. 2.

Suppl. Figure 2 | MLM loss as a function of the total number of tokens seen by the model. Shown are the loss curves of three different models with varying parameter size, 15.1 million parameters, 40.9 million parameters and 49.3 million parameters, respectively. The larger the model, the lower is the pretraining loss. All the losses are a moving average with a window of 10. All the models were evaluated in the same training set with fixed random seed.

There is just one single Nicheformer model, trained across spatial and dissociated data. Hence, there are just one set of hyperparameters used.

(b) Here, I would suspect that due to differences between dissociated versus spatial data, there would be (should be) differences. For example, in spatial, there is less of a number of genes – note that the authors use image-based spatial transcriptomics data – whereas the dissociated data have more comprehensive expression profiles. Wouldn't you then need more tokens for the dissociated data and vice versa? The size of the data sets would be biased towards more or less the number of attention heads. Since the heads in different layers attend to different “things” (dissociated versus spatial aspects), would you then not expect different constraints with the number of heads in different layers? Fewer attention heads for spatial data ? Would a particular final choice of hyperparameters tilt Nicheformer to take dissociated or spatial data into “more” account or give it a larger relative weight?

We thank the reviewer for this interesting point. At the moment, we simply train one single Nicheformer model across dissociated and special data. Hence, we optimized our model in the joint dataset. It is however possible - as proposed by the reviewer - that modality-specific models require different hyperparameters. This way, the number of attention heads is fixed for both modalities, it cannot be adapted on the fly depending on the modality of the cell that Nicheformer is processing. The reviewer raises an important point about the relationship between the number of tokens and attention heads. While it is true that dissociated cells typically have fewer tokens (representing genes) compared to spatially measured cells, there is no established correlation suggesting that smaller token contexts inherently require fewer attention heads. Furthermore, the attention analysis (shown in the next point) reveals that most of the attention patterns are consistent across modalities, tissues and organisms.

In large language models (LLMs), it has been shown that attention heads often specialize in distinct tasks unrelated to context length. For example, induction heads⁷ and successor heads⁸ demonstrate functional specializations that span the entire context, rather than partitioning the context length into isolated regions for each head - at most, there are attention heads that focus on the most expressed genes, but we did not find ones focusing on the less expressed, or middle expressed, etc.. Our visualization of attention matrices in the supplementary material further supports this: the attention heads perform operations across the whole context length rather than splitting it into disjoint segments. It is not that, for instance, head 1 attends to genes from 1 to 100, head 2 from 100 to 200, and so on.

Furthermore, we want to highlight that there is a tradeoff between the number of attention heads and the capacity of each head. For a fixed token dimensionality, fewer heads imply a higher dimensionality per head, which may limit the ability to model diverse aspects of the data effectively. Conversely, increasing the number of heads allows for greater specialization, albeit with reduced dimensionality per head. State-of-the-art models, such as LLaMA ⁹, adopt a strategy where the

dimensionality of each head is fixed, and the total number of heads scales with model dimensionality, ensuring adequate capacity for diverse tasks.

Finally, we run an analysis to study how robust Nicheformer is to dropouts in the context length. The results were robust, which suggest that by being trained on a joint dataset with different gene panels, the model learns how to handle unequal context lengths.

We evaluated whether Nicheformer is robust to the commonly used gene ranking strategy², in particular when a subset of the genes is not used as input to the model. This evaluation resembles the real-world challenge of probe-set selection in spatial transcriptomics, where only a small set of genes is measured. We randomly shifted, removed or transposed the ranks of individual or groups of genes (Methods). Next we embedded the cells with perturbed ranks with Nicheformer and used integration metrics to assess whether the non-perturbed-rank and perturbed-rank cells are embedded closely by the model. We observed that cell embeddings are stable in both technologies up to a perturbation of 20 % in their gene ranks in terms of silhouette score (Suppl. Fig. 1). This analysis shows that Nicheformer's embeddings are sufficiently robust to input perturbations such as incomplete gene panels.

(c) same as (b) – but instead of contrasting dissociated versus spatial, rather – was there any difference w.r.t. best hyperparameters for mouse vs human?

As Nicheformer is trained across both dissociated and spatial, including mouse and human cells, there are not Nicheformer-Mouse and Nicheformer-Human models. Hence, the hyperparameters chosen were selected training in the joint training set including both organisms. While there can be different hyperparameter combinations for organism-specific models, we only investigated the hyperparameter performance in the joint scenario.

A general remark to this point of hyperparameters. The results of our experiments pointed out that what seems to be the most important factor is the total number of parameters, the allocation of them does not seem to be that determinant in the final performance of the model. In Suppl. Table 1, it can be seen that out of all hyperparameters tested, the ones that gave the best results were always the ones that increased the most the number of parameters of the model. This is completely in line with results obtained in other fields in which scaling laws and different parameterizations of models were studied, which showcase that the total number of parameters, rather than their specific allocation, is the most important factor^{1,10,11}.

7. The attention analysis. I find this part very interesting, but it does not really come through in the ms. The early layers find gene-gene interactions, whereas the spatial comes in later layers and metadata in the last layers.

We thank the reviewer for their feedback. The attention mechanism is indeed very interesting and during the course of this revision we have improved our understanding of how it operates. We provide more comments point-by-point below.

(a) These observations – as far as I can tell from the ms – are made in the brain data, which is the premier showcase for nicheformer w.r.t. performance. Do these observations generalize to other tissues? Is this a generic scheme that is reflected here in the brain tissue?

We agree with the reviewer that investigating the consistency of this schema across different tissues can provide valuable insights. We therefore additionally performed the attention analysis in the CosMx human liver and lung datasets. Specifically, we analysed the attention patterns of liver cells (2000 disease cells and 2000 healthy cells) and lung cells (2000 CD4 cells and 2000 CD8 cells). Additionally, we have also analysed 2000 cells from the dissociated brain dataset measured in the motor cortex. This way, we have human cells (lung and liver dataset) and mouse cells (brain datasets), as well as cells from different tissues and different technologies.

As the reviewer points out, it seems that Nicheformer transformer blocks pay attention to different aspects of the cells. In the new analysis we have deepened further in this phenomena. What we have found is that - interestingly - the patterns are consistent across tissues.

In the first place, the last layers of Nicheformer pay very high attention to the metadata tokens. This pattern is consistent across datasets. We consider this point is related with the last point of this response, which explains that the L2 norm of the metadata output tokens is very high. Notice that while the magnitude of attention paid to metadata tokens in the dissociated case changes, the pattern is still consistent. We run statistical analysis to validate the statistical significance of this pattern (Suppl. Fig. 7).

Next, Nicheformer consistently demonstrates a hierarchical attention pattern across datasets. The early layers distribute attention more uniformly across all tokens, while the middle layers exhibit a

sharper focus, likely attending to more specific biological factors. This pattern appears robust across tissues, suggesting that Nicheformer’s hierarchical attention specialization is a generic property rather than being specific to brain tissue. Specifically, we measure the maximum attention paid to every gene token in each of the attention heads of each attention block. The results show that Nicheformer middle layers sharply focus on specific tokens. We run statistical analysis to ensure the statistical significance of these results, which we added as supplementary figures (Suppl. Fig. 8).

The trend is clearly similar: early Nicheformer layers scatter their attention across all genes; middle layers sharply focus on specific biological features; and later layers focus strongly on metadata tokens. This is a meaningful strategy and it is interesting that the network has learned this. Furthermore, notice that there exists a difference in absolute values between brain dissociated cells

and brain spatial, liver and lung cells. This difference is caused due to the difference in the selection process of the genes whose attention is studied. In the case of the brain cells, we sampled 100 random genes not related with sex differences in brain; while in the case of liver and lung cells, we sampled the union of the top 50 most expressed genes per each condition (CD8 vs CD4 and disease vs healthy). This, in fact, denotes that Nicheformer pays special attention to the highest expressed genes, which can be seen, for instance, in the attention heads to which we refer below. We also evaluated statistically the significance of these results.

Finally, we also verify that the behaviour of the attention heads is indeed robust across tissues. While in LLMs the new field of mechanistic interpretability has allowed us to clearly identify circuits in transformers that carry on specific functions (as mentioned before, examples are induction heads⁷ and successor heads⁸), interpretability of large transformer models in cell biology is still a challenge. Still, we identify that there exists attention heads with particular patterns that hold across tissues, which indicate that they are highly specialised.

We have notably expanded the attention section to gather and detail better this analysis, as well as adding several supplementary figures.

[...]. Our analysis suggests a distinct division of labor across Nicheformer's layers. The final layers consistently focus on contextual tokens (Fig. 3A, Suppl. Fig. 6), while middle layers exhibit a sharp attention toward specific genes (Fig. 3B), likely capturing biologically relevant relationships. In contrast, the early layers distribute their attention more broadly, with no clear prioritization of individual tokens. This structured pattern of attention specialization is robust across all analyzed tissues and modalities, indicating that Nicheformer learns a hierarchical representation that generalizes beyond a single dataset. To assess the statistical significance of those findings, we run, independently per each dataset, Mann-Whitney U test to compare the attention distributions between gene and contextual tokens, and between different layers of Nicheformer. We adjusted the p-values with the Benjamini-Hochberg procedure to control the false discovery rate (FDR). The results were found statistically significant (Suppl. Fig. 7-8).

Beyond this layer-wise organization, we found that attention heads maintain consistent functional roles across tissues and modalities. For example, certain heads consistently prioritize highly expressed genes, regardless of whether the dataset originates from brain, liver, lung, or dissociated cells (Suppl. Fig. 9). Nevertheless, some other heads showed behaviours that change depending on the modality of the cells processed, which indicates that Nicheformer contains internal mechanisms that differ per each modality (Suppl. Fig 10).

In general, we found a large variety of patterns across the attention heads. Some heads exhibit strong self-attention patterns (visualize as strong diagonal attention scores), while some show off-diagonal patterns that indicate that the attention is distributed across genes with similar normalized expression (and therefore closer in the input sequence) (Suppl. Fig. 11). These findings highlight the diverse range of attention behaviors that Nicheformer develops when processing complex biological data. Interestingly, recent work in large language models (LLMs) has demonstrated that specific attention heads acquire well-defined functions, such as induction heads that detect repeated patterns in sequences⁷ or successor heads that track sequential dependencies⁸. While mechanistic interpretability in biological foundation models is still in its early stages, our results suggest that Nicheformer exhibits a similar specialization, with certain heads consistently attending to biologically relevant features across datasets.

The Methods section is also updated:

We conduct an attention analysis to explore the attention patterns in Nicheformer and how it differentiates between male and female cells by focusing on sex-specific gene variations. We sample 2,000 CD8 and 2,000 CD4 cells from the lung; 2,000 healthy and 2,000 cancer cells from the liver; 2,000 male and 2,000 female cells from the MERFISH mouse brain datasets and 2,000 random cells from the primary motor cortex scRNA-seq dataset to ensure sufficient diversity. In all cases, except in the MERFISH mouse brain dataset, we study the attention paid to the top 50 most expressed genes on average. For the MERFISH mouse brain cells, we use two gene sets: a prior-knowledge set of SDGs, known for exhibiting sex differences, and a randomly sampled control set of 97 genes. We feed all cells into the model and extract attention matrices from all 16 attention heads across the 12 transformer blocks.

Finally, we also mentioned our findings in the discussion:

To further understand how Nicheformer processes information, we analyzed its attention mechanism, finding that different layers attend to distinct features. We identified specific attention heads that remain robust across modalities and tissues, as well as others that adapt to these variations. We also explored how Nicheformer captures biological conditions through its attention patterns.

(b) Why do we see this – a division of labor – between the layers? Is this an effect of finely tuned hyperparameters where we have large enough data of different modalities, allowing the authors to have a sufficient number of layers and attention heads to disentangle such a fine-graining of the representations? Smaller data sets, less spatial, and so forth... would (potentially ?) mask this scheme.

We agree that the differentiation between the different transformer layers is a crucial aspect of our work and worth to be addressed also in a dissociated scRNAseq dataset. We refer to our response in the previous point (7 (a)) to show the results of the attention analysis in the dissociated dataset and how it compares with the spatial datasets. In general, the patterns are consistent.

On top of this, recent research, in LLMs, demonstrates that different layers in transformers exhibit functional specialization, with different layers performing different operations¹². Importantly, we want to clarify again that the attention results we inspect are the ones generated during pretraining, we do not fine tune the model anyway to obtain this results. Hence, this behaviour appears due to the self-supervised pretraining and not by means of any downstream task-specific fine tuning. We clarify this in line 277:

We evaluated whether Nicheformer pretraining and obtained attention matrices capture meaningful biological variations in two MERFISH mouse brain datasets that were obtained from a male and a female mouse respectively¹³ and are part of the SpatialCorpus-110M (Methods)

(c) Do the authors observe this in the lung and liver tissues ?

We refer the reviewer to our response in 7 (a).

8. A sample of technical details that are missing:

We thank the reviewer for their feedback. We provide more comments point-by-point below.

- The model pads missing genes with tokens and sets all sequences to a fixed length of 1,500 tokens. However, spatial technologies often measure far fewer genes than scRNA-seq. This is particularly true for image-based spatial technologies – which the authors rely on here. The paper doesn't explain how the model deals with these substantial differences in sequence lengths or the potential loss of information due to truncation. While the results claim the model is robust to incomplete gene panels, the methods lack details on how missing data affects gene ranking and model performance. High proportions of tokens in spatial data might also impact the attention mechanisms in the model.

According to analysis run to assess the robustness of the representations learnt by Nicheformer (Suppl. Fig.1) we observed no major changes in the representations due to dropout or permutations of the genes in the context length.

A detail we consider worth mentioning and that can help to explain this behaviour (which is common among BERT-like models) is that not all tokens masked by Nicheformer are changed by a <MASK> token. In general, 15% of the tokens are masked. Of those, 80% are substituted by the <MASK> tokens, so the model learns to complete the “sentence” of cells using the unmasked tokens. Another 10% of tokens are left unchanged, so the model learns to identify the real and actual information in the context length. Finally, the last 10% of tokens is changed by another random token, so the model learns to correct the wrong information in its context. This way, the model is encouraged to adapt itself to perturbations and wrong tokens in its context.

- There is minimal information on how scVI and PCA embeddings were generated (e.g., choice of latent dimensions, hyperparameters). Without consistent configurations, differences in performance may arise from suboptimal baseline settings rather than a genuine superiority of Nicheformer. Detailed baseline configurations are essential for a fair comparison.

We acknowledge that there was a lack of details about how the scVI and PCA embeddings were generated. This is also pointed out by Reviewer 2. We have updated accordingly the Methods section providing more details as well as adding into supplementary material the plots of the variance explained by the PCA, to better identify how the number of principal components corresponds with the explained variance in the train set (Suppl. Fig 21). We have also computed extended benchmarks against different principal components used (Suppl. Fig 16).

Next we use scanpy to log1p-transform the data matrix to ensure the data is centered before using it as input to the PCA implementation. We use the sklearn implementation and evaluate the cumulative explained variance ratio in the train dataset (Suppl. Fig. 27). Finally, we evaluate the model for a diverse set of principal components to have a fair comparison (Suppl. Fig 18-20).

Notice as well that we have expanded our comparisons with PCA to test multiple dimensions instead of just a couple, as was also pointed out by Reviewer 2.

- The paper does not mention any techniques to address class imbalance.

We do not use class imbalance techniques for three main reasons. First of all, during the pretraining stage, no labels are used, hence no class imbalance technique can be used. Secondly, during the downstream tasks that involve classification, in which labels are used for training, we train for only one single epoch during downstream tasks to ensure that the model relies on its learned representations rather than overfitting to the labels. This approach aligns with our goal of demonstrating the generalizability and robustness of the representations learned by Nicheformer during the unsupervised pretraining phase. Finally, even though it sounds counterintuitive, previous works have shown how the use of class imbalances can even hurt performance¹⁴. We have expanded the Methods section to mention this:

We use no techniques to address class imbalances for two reasons. First, to evaluate the robustness of the representations learnt by Nicheformer. Secondly, it has been shown that using class imbalances techniques can even hurt performance in cases such as cell type classification

- The train-test split varies between datasets (e.g., random field of view hold-out vs. random cell hold-out). Different splitting strategies may introduce biases, such as including correlated cells in both sets or not accounting for donor variability.

We agree with the reviewer about the importance of adequate splits. Also, we want to highlight the difficulty in selecting the right holdouts. We want to clarify that Nicheformer does not use any spatial information during training that is not used by the other baselines. Hence, if there are biases in the data, those biases affect in an equal manner to Nicheformer than to the baselines. Still, we believe that the current splits mostly prevent those biases.

- Contextual tokens for , <MODALITY>, and <ORGANISM> are prepended to input sequences but are excluded during embedding aggregation for downstream tasks. Excluding contextual tokens during aggregation may remove important modality-specific information that the model has learned to incorporate during pretraining. Reassess the exclusion of contextual tokens during embedding aggregation or justify their omission.

Thank you for raising the point of the exclusion of the contextual tokens. It is indeed an interesting point that we did not properly explain in the initial manuscript. In the beginning, we treated the cell representation as an aggregation of all tokens, including the contextual ones. However, the label

transfer between modalities was not working. It started working only when we excluded the metadata tokens from the computation. The downstream tasks were not affected and there is no difference in using or not the contextual tokens in the embedding aggregation. Raising this point has prompted us to investigate this in more detail and we have found interesting insights.

Specifically, to study this point, we have examined which is the average output token norm. The rationale is that we suspected that the average of the contextual tokens was very large, therefore affecting in a substantial manner the mean computation. Specifically, we have computed the output token average norm across more than 2000 tokens in separated tissues: lung and liver. In all cases, we found that the <MODALITY> token was always the token with highest average norm, with a very notable low variance of the norm. As result, we have added in the supplementary material the following histograms, that illustrate the distribution of average output token norms. The highlighted bin represents the bin in which the <MODALITY> token is found.

Interestingly, this effect of tokens with high output norm has been recently studied in the context of vision models in ¹⁵, in which they find that transformers employ tokens with low input variability to allocate them to perform internal computations. We also believe that this result is related with the attention patterns that show that the attention in the last layers of Nicheformer is mainly biased towards the contextual tokens.

We have expanded the Methods section in which we mentioned the aggregation of the tokens to get the cell representation:

Importantly, the contextual tokens are not used in the aggregation. While we observed no difference between using them or not in the downstream tasks focused on one modality - density prediction, niche classification, etc. -, we observed that transferring labels between spatial and dissociated datasets did not work at all when using the contextual tokens in the aggregation. Further investigation revealed that the output norm of the contextual token of modality was always the highest one, independently of the tissue (Suppl. Fig 24), hence playing a big role in the cell representation and biasing it towards the respective modality. This phenomenon has been reported in vision transformers¹⁵, where some features that contain background information show higher norm as a consequence of the model using them to allocate internal computations. Literature¹⁵ proposes the use of registers that are discarded in the computation of the final representation

Reviewer #2 (Remarks to the Author):

Similar to many existing studies on single-cell foundation model building, the authors proposed to build for their own foundation model. The topic is trendy but well-trying by others as evidenced by the competing methods shown by the authors themselves as well as simple Google searches. The proposed model is mostly based on transformer architecture. The results are mostly numeric for performance comparisons with downstream tasks in spatial transcriptomes. I have few comments:

We greatly appreciate the time spent by the reviewer in reading and judging our manuscript! While indeed transformer based models have become en vogue, one point to raise is that unsimilar to many existing studies, Nicheformer includes spatial context and actually introduces new, hard tasks where this approach outperforms more non-spatial foundation models as well as more classical embeddings.

We provide a detailed response point by point below.

1. One of the most controversial concerns on such a foundation topic is whether it is necessary to have deep learning foundation model as there is already a study arguing that PCA was already enough [1]. The authors are clearly aware of it, based on the extensive PCA comparisons in the study. However, the current comparisons with PCA are still not enough. The most concerning part to me is that the PCA comparison is based on 30 components as stated by the authors: "We use the sklearn implementation with the number of components being set to 30. ". Other number of PCA components should be examined for fairness as it is targeted for foundation modelling but not single-task model.

Thank you for this point. It is true that there has been an increase in the number of papers presenting foundational models for single-cell transcriptomics lately. And we agree that in most cases, mainly for perturbation prediction tasks, the results are sobering since linear models often outperform those large deep learning models¹⁶⁻¹⁸. We indeed agree that the evaluation against solid baselines helps to put in perspective the effectiveness of novel methods, that is why we compare against different foundational models - and according to the suggestion we have now majorly increased the range of PCA comparisons. We have expanded the comparison against PCA and reported how the performance varies when increasing the number of PCs (10, 32, 64, 128, 256 and 512) and also added plots to show the explained variance. We have updated the Methods section to point this out and we have included several supplementary figures showing how the number of principal components scales with the variance explained in the train set (Suppl. Fig.27), and how the performance in different downstream tasks evolve (Suppl. Fig.18-20) as a function of the number of components employed. As potentially implied by the reviewer, we indeed see an increase in downstream performance for a higher number of PCs but still lower than Nicheformer, indicating the necessity of nonlinear models in this setting. (This has been shown before in the case of cell type classification for complex cross organ classifiers¹⁴)

We modified the text to mention this comparisons:

We found that both of our approaches outperform both traditional embedding methods, scVI and PCA, no matter which was the training set, as well as GeneFormer, scGPT, UCE and CellPLM, in terms of macro F1 score, which effectively balances precision and recall across all classes (Fig. 4B, Suppl. Fig. 15). Further comparisons using PCA embeddings with an increasing number of principal components also confirm Nicheformer's superior performance. While, specially, in the case of the brain dataset, using PCA with a large number of components offers a good performance practically on par with using a linear probe on top of Nicheformer's representations, or even surpassing it in the case of region prediction - but being worse than fine tuning Nicheformer (Suppl. Fig. 18). The differences between Nicheformer and competitors were found statistically significant running t-tests between Nicheformer and the best performing comparison method (Suppl. Fig. 15). Interestingly, this comparison of linear versus non-linear approaches has also been studied in the context of cell type classification, where it has been proven that non-linear approaches eventually outperform linear ones⁹¹ even though the former is still competitive.

2. Furthermore, how much variance the PCA has explained? It is very important for PCA tuning to such an extension that the majority of data variance can be explained. Indeed, by looking at Figure 4b, it is very clear that PCA could already be enough and it could be embarrassing to get published just for the sake of complicated AI in biology here with the exponentially increased model complexity in both time and space.

As mentioned in the previous point, we have added a supplementary figure (Suppl. Fig. 27) showing the variance ratio for the different datasets employed in the downstream tasks and also a supplementary figure (Suppl. Fig. 18-20) comparing the performance of different principal components against Nicheformer. Notice that while it is true that the first PCA dimensions contribute notably to the performance, the effect naturally fades away with progressively more principal components, since they explain less and less. Notice as well that the explained variance corresponds only to the train set, not to the test set.

3. The word "Corpus" is mostly about text data in the NLP community. I have no idea why the authors adopted such a word for spatial transcriptomics.

We agree with the reviewer that the word "corpus" is often used in the context of language. However, we understand that corpus refers to a collection of data used to train machine learning models. Examples of other papers that refer to datasets as "corpus" are ¹⁹ and ^{19,20}. Additionally, science colleagues and collaborators have already asked for the dataset and cite it, so we decided that changing the name now would be more confusing than keeping it

4. Another challenging aspect here is the so-called "foundation model". Isn't it just a feature extraction or dimensionality reduction model? At the end of the day, it is still a model adopted by the related researchers only in a niche area. How would it be broadly adopted by biologists easily? For now, it is mostly released with source code on GitHub. I cannot see any web interfaces or easy-to-use functionalities. It is quite hard to justify for its broad impacts with the word "foundation model".

We thank the reviewer for raising this point. Nicheformer is positioned as a foundation model because it is trained across a wide array of diverse datasets (in fact at the moment one of the largest

single cell models around in terms of data), enabling transfer learning for various downstream tasks. Similar terminology has been applied to other models in single-cell genomics, such as scBERT, Geneformer, scGPT, CellPLM, or scFoundation, which aim to generalize across multiple datasets and tasks. While we agree that the nomenclature might be used too boldly, we also consider that it is important to mention it to place the model among related models. Finally, this term has been broadly taken up in reviews and eg by Nature as one of the methods to watch²¹.

Regarding the adoption by biologists, we have refactored the GitHub repository to make it more user-friendly, with improved documentation and workflows to ensure researchers can integrate Nicheformer into their analyses effectively. We believe these efforts support its potential for broad impact as a foundation model in omics research. We have already received positive feedback from collaborating labs that are actively using the model in their studies.

The new Nicheformer is provided as an installable package with notebook tutorials for different functionalities such as getting embeddings, solving downstream tasks, fine tuning or continuing pretraining in in-house datasets. It can be installed using the following commands:

```
mamba create -n nicheformer_env python=3.10
mamba activate nicheformer_env
git clone https://github.com/theislab/nicheformer.git
cd nicheformer/
pip install -e .
```

5. As the submitting journal is “Nature Methods”, wet-lab validation is missing here. It is not clear if the model can reveal truly novel insights based on existing well-known data crawled from the web, although training and testing are separated.

We agree with the reviewer that wet-lab experiments are the cornerstone of biological research and this work is luckily based on a large corpus of datasets generated with the effort of many labs around the world. However, we consider our study to be a methodological contribution aimed at advancing computational approaches for analyzing these datasets. By benchmarking against multiple state-of-the-art methods and demonstrating strong predictive performance across diverse datasets, we provide compelling evidence of Nicheformer's utility. This is common practice for a broad series of papers in Nature Methods and related journals. We thank the reviewer for kind understanding here.

6. Different transcriptomic sequencing technologies / platforms have different data biases. It is not clear if the existing data normalization steps are sufficient and integrated into a single “corpus”.

We thank the reviewer for raising this point. Something important to highlight is that we did not want to create an atlas of integrated data, as many other efforts within the single-cell community. Our intention was to gather a collection of different datasets that can be used to train models at scale. Hence, the data is indeed not integrated - similar to for example Cellxgene, which is not

integrated into a joint latent space as well - it is instead a resource that we hope the community uses to train their large models. We believe that normalization or integrated choices correspond to modeling choices of the eventual models trained.

In our case, we understand that as part of the tokenization, the reason for which we use assay specific median vectors to scale the data and also the reason why we employ <ASSAY>, <MODALITY> and <ORGANISM> tokens. We are aware that the data is not yet integrated, hence the model should learn a powerful representation by being aware of the nuances and differences of the data.

We consider that it might be worth linking this point to point 3 to clarify again that what we wanted to get with SpatialCorpus110M is a large collection of datasets to train models. We did not aim at integrating all datasets into one large atlas and then train models on it. The same way that Large Language Models (LLMs) such as GPT4 or Llama are trained on Internet-size data, SpatialCorpus110M is intended to be a large collection of datasets to train large models.

We added the following text:

Importantly, while we applied these steps to ensure consistency in feature representation, our goal was not to create an integrated reference atlas but rather to compile a large-scale dataset that enables model training across diverse contexts. We deliberately preserved dataset-specific characteristics rather than applying batch correction or integrating datasets into a single latent space. Instead, Spatial-Corpus-110M serves as a diverse collection of datasets that researchers can use to train large-scale models, leaving normalization and integration choices to individual modeling approaches. This aligns with other large-scale repositories, such as Cellxgene, which provide raw data for flexible use rather than enforcing a unified representation. We believe this flexibility makes the corpus more broadly applicable to a range of machine learning models and downstream analyses

7. Nonetheless, I do appreciate the use of orthologs in such a modelling. It is largely missed by the existing models. However, p-values and FDRs are missing in the current study.

We thank the reviewer for appreciating the use of orthologs! Also, we acknowledge the lack of p-values and FDR correction. We have added p-values to our model comparisons, which are shown in the supplementary figures with the extended comparisons (Suppl. Fig. 15-20). We also use p-values to assess the statistical significance of the analysis asked by reviewer 1 (Suppl. Fig. 3 & 5). For the comparisons, we have used either t-test in the case of comparing 2 distributions - when comparing Nicheformer with the best performing competitor, e.g. the case of the PCA with increasing number of principal components - and ANOVA in the case of comparing more than 2 distributions. The p-values provided are always FDR adjusted using the Benjamini-Hochberg procedure. We have extended the text as follows:

[...] Furthermore, the differences between models are found statistically significant, running ANOVA tests and adjusted for false discovery rate (FDR) using the Benjamini-Hochberg procedure (Suppl. Fig. 3, Suppl. Fig. 5).

[...] Statistical tests (t-test) to assess the statistical significance of the results were performed, with positive results (Suppl. Fig. 15-17).

For the attention analysis we have now included p-values to show that the attention patterns we identify are statistically significant. In particular, we run Mann-Whitney U Test, across different tissues and modalities independently, to verify whether the differences in attention between different groups of tokens and layers are significant. For controlling for FDR, we again employed the Benjamini-Hochberg procedure to adjust the p-values. The text has been adapted as follows and figures can be found in the supplementary (Suppl. Fig 7-8).

[...] To assess the statistical significance of those findings, we run, independently per each dataset, Mann-Whitney U test to compare the attention distributions between gene and contextual tokens, and between different layers of Nicheformer. We adjusted the p-values with the Benjamini-Hochberg procedure to control the false discovery rate (FDR). The results were found statistically significant (Suppl. Fig. 7-8).

1. Kaplan, J. *et al.* Scaling Laws for Neural Language Models. (2020).
2. Smedley, D. *et al.* BioMart--biological queries made easy. *BMC Genomics* **10**, 1–12 (2009).
3. Martin, F. J. *et al.* Ensembl 2023. *Nucleic Acids Res.* **51**, D933–D941 (2023).
4. Muennighoff, N. *et al.* Scaling Data-Constrained Language Models. (2023).
5. Albalak, A. *et al.* A Survey on Data Selection for Language Models. (2024).
6. Mitchell, M. *et al.* Measuring Data. (2022).
7. Olsson, C. *et al.* In-context Learning and Induction Heads. (2022).
8. Gould, R., Ong, E., Ogden, G. & Conmy, A. Successor Heads: Recurring, Interpretable Attention Heads In The Wild. (2023).
9. Touvron, H. *et al.* LLaMA: Open and Efficient Foundation Language Models. (2023).
10. Henighan, T. *et al.* Scaling Laws for Autoregressive Generative Modeling. (2020).
11. Hoffmann, J. *et al.* Training Compute-Optimal Large Language Models. (2022).
12. Sun, Q., Pickett, M., Nain, A. K. & Jones, L. Transformer Layers as Painters. (2024).
13. Yao, Z. *et al.* A high-resolution transcriptomic and spatial atlas of cell types in the whole mouse brain. *Nature* **624**, 317–332 (2023).
14. Fischer, F. *et al.* scTab: Scaling cross-tissue single-cell annotation models. *Nature Communications* **15**, 1–15 (2024).
15. Darcet, T., Oquab, M., Mairal, J. & Bojanowski, P. Vision Transformers Need Registers. (2023).
16. Bendidi, I. *et al.* Benchmarking Transcriptomics Foundation Models for Perturbation Analysis : one PCA still rules them all. (2024).
17. Ahlmann-Eltze, C., Huber, W. & Anders, S. Deep learning-based predictions of gene perturbation effects do not yet outperform simple linear baselines. *bioRxiv* 2024.09.16.613342 (2025) doi:10.1101/2024.09.16.613342.
18. Boiarsky, R., Singh, N., Buendia, A., Getz, G. & Sontag, D. A Deep Dive into Single-Cell

RNA Sequencing Foundation Models. *bioRxiv* 2023.10.19.563100 (2023)

doi:10.1101/2023.10.19.563100.

19. Ahdritz, G. *et al.* OpenProteinSet: Training data for structural biology at scale. (2023).
20. Li, Q. *et al.* OmniCorpus: A Unified Multimodal Corpus of 10 Billion-Level Images Interleaved with Text. (2024).
21. Eisenstein, M. Self-driving laboratories, advanced immunotherapies and five more technologies to watch in 2025. *Nature* **637**, 1008–1011 (2025).

Reviewer #1 (Remarks to the Author):

Comments on the revised manuscript "Nicheformer: a foundation model for single-cell and spatial omics, by Schaar et al.

The authors addressed the main concerns, including the Lack of comparative analysis. Here, the authors performed extensive benchmarking using different subsets of data (including species) to evaluate how different modalities (dissociated vs. spatial) impact model performance. This analysis clarified the importance of spatial data, which could not be compensated for by the information in the dissociated cells. Robustness of attention mechanisms. The additional analysis was carefully executed, and it was established that the observed structured hierarchy in attention layers was generalized to all the other tissues beyond the brain.

Additional benchmarking against the other models (Universal Cell Embeddings (UCE) and CellPLM, along with PCA, scVI, scGPT, and Geneformer) as requested. The results confirmed Nicheformer's superior performance.

The GitHub Repository was updated, and we confirmed that it was easier to install and use, including tutorials for tokenization, model training, and downstream tasks.

The authors responded and performed a proper analysis addressing the points. Furthermore, I appreciate that the authors could deepen their understanding of the attention analysis, which now has a more substantial backing in the revised manuscript.

I believe the authors are to be commended for an excellent revision, and I have no further requests.

We thank the reviewer for its feedback. The issues the reviewer raised were certainly justified and its feedback has been invaluable to improve the paper.

My only remaining suggestions are:

1. The authors note that the <MODALITY> and <ORGANISM> tokens have high norms in the last layer of the transformer model, which could overshadow the gene expression signals if included in the final cell embedding. To address this, they excluded these contextual tokens when computing the embedding. However, a potential downside of this "all or nothing" approach is by completely excluding the <MODALITY> and <ORGANISM> tokens from the final embedding, the model may lose valuable information for tasks where modality or organism context is relevant. They may consider this for future research or to briefly discuss it in the manuscript as a possible extension of their approach.

We acknowledge this and we have modified the parts in Methods in which we mention the norm analysis done, as well as in the discussion, saying that this approach can be improved in future works:

Additionally, the current strategy excludes metadata tokens from the final cell representation to avoid bias from their high norm (Methods), which can impede label transfer. However, this may limit model expressivity by discarding these tokens entirely. More refined strategies, such as selective integration, could retain relevant context without allowing it to dominate the embedding.

2. We checked their GitHub and noticed improvements, particularly in the following aspects: (a) Improvement in the installation process, making setup easier and faster. (b) The addition of exemplary notebooks covering various tasks provides better guidance for users and enables them to run and adapt the notebooks to their data.

Another point we noted is that there have been 17 open issues from different users (since last year), highlighting different problems or doubts related to the tool. Most of them remain unresolved, and addressing them would improve the user experience and the tool's performance. (issues:

<https://github.com/theislabs/nicheformer/issues>)

Thank you for pointing this out. We have already closed most of those open issues regarding the previous API and for the remaining ones we are currently iterating on them to solve and improve user perspective.

Reviewer #2 (Remarks to the Author):

Since the handling editor still gave a revision chance for the authors with only 2 reviews without any wet-lab verification on Nature Methods, I do not have any further comment although the authors are so confident in their final acceptance without due respects on reviewers.

We want to thank reviewer 2 for their useful feedback which has helped to substantially improve our manuscript.